

Inter-comparison of multiple two-way coupled meteorology and air quality models
(WRF v4.1.1-CMAQ v5.3.1, WRF-Chem v4.1.1, and WRF v3.7.1-CHIMERE v2020r1)
in eastern China
Chao Gao[1, 2], Xuelei Zhang[1, 2, *]Aijun Xiu[1, 2, *], Qingqing Tong[1, 2], Hongmei Zhao[1, 2], Shichun Zhang[1, 2],
Guangyi Yang[1, 2, 3], Mengduo Zhang[1, 2, 3], and Shengjin Xie[1, 2, 4]
[1]Key Laboratory of Wetland Ecology and Environment, Northeast Institute of Geography and Agroecology, Chinese
Academy of Sciences, Changchun, 130102, China
[2]Key Laboratory of Wetland Ecology and Environment, State Key Laboratory of Black Soils Conservation and
Utilization, Northeast Institute of Geography and Agroecology, Chinese Academy of Sciences, Changchun, 130102,
China
[3]University of Chinese Academy of Sciences, Beijing, 100049, China
[4]School of Environment, Harbin Institute of Technology, 150000, Harbin, China
Correspondence to: X.L. Zhang (zhangxuelei@iga.ac.cn) & A.J. Xiu (xiuaijun@iga.ac.cn)
Abstract
In the eastern China region, two-way coupled meteorology and air quality models
have been applied aiming to more realistically simulate meteorology and air quality by
accounting for the aerosol–radiation–cloud interactions. There have been numerous
related studies being conducted, but the performances of multiple two-way coupled
models simulating meteorology and air quality under equivalent configurations have
not been compared in this region. In this study, we systematically evaluated annual and
seasonal meteorological and air quality variables simulated by three open-source and
widely used two-way coupled models (i.e., WRF-CMAQ, WRF-Chem, and WRF-
CHIMERE) by validating the model results with surface and satellite observations for
eastern China during 2017. Our comprehensive model evaluations showed that all three
two-way coupled models simulated the annual spatiotemporal distributions of
meteorological and air quality variables reasonably well, especially the surface
temperature (with R up to 0.97) and fine particular matter ($PM_{2.5}$) concentrations (with
R up to 0.68). The model results of winter $PM_{2.5}$ and summer ozone compared better
with observations and WRF-CMAQ exhibited the best overall performance. The
aerosol feedbacks affected model results of meteorology and air quality in various ways
and turning on aerosol-radiation interactions made the $PM_{2.5}$ and surface shortwave
radiation simulations better, but worse for T2 and Q2. The impacts of aerosol-cloud
interactions (ACI) on model performances' improvements were limited and several
possible improvements on ACI representations in two-way coupled models are further
discussed and proposed. When sufficient computational resources become available,
two-way coupled models including the aerosol-radiation-cloud interactions should be
applied for more accurate air quality prediction and timely warning of air pollution
events in atmospheric environmental management.





## 1 Introduction

Aerosols in the atmosphere due to anthropogenic and nature emissions not only cause air pollution but also induce climate and meteorological impacts through aerosol-radiation interaction (ARI) and aerosol-cloud interaction (ACI) (Carslaw et al., 2010; Rosenfeld et al., 2014; Fan et al., 2016; IPCC, 2021). The feedbacks of aerosols to meteorology have been widely investigated by two-way coupled meteorology and air quality models in the past two decades (Jacobson, 2002; Grell et al., 2005; Wong et al., 2012; Wang et al., 2014; Zhou et al., 2016; Briant et al., 2017; Feng et al., 2021). In these models, two-way interactions between meteorology and aerosols are enabled by including all the processes involving ARI or/and ACI (Grell and Baklanov, 2011; Wang et al., 2014; Briant et al., 2017; Wang et al., 2021). The fundamental theories, modeling technics, developments, and applications of two-way coupled meteorology and air quality models in North America, Europe and Asia have been systemically reviewed (Zhang, 2008; Baklanov et al., 2014; Gao et al., 2022).

As pointed out by these review papers, the treatments and parameterization schemes of all the physiochemical processes involving ARI and ACI can be very different in two-way coupled models, so that the simulation results from these models could vary in many aspects. At the same time, the configurations of coupled models, such as meteorological and chemical initial and boundary conditions (ICs and BCs), horizontal and vertical resolutions, and emission inventories and processing tools, etc., play important roles in models' simulations. In the past, model inter-comparison projects have been carried out targeting various two-way coupled meteorology and air quality models. For example, the Air Quality Model Evaluation International Initiative Phase II focused on the performance of multiple two-way coupled models and the effects of aerosol feedbacks in Europe and the United States (Brunner et al., 2015; Im et al., 2015a, b; Makar et al., 2015a, b). In Asia, the Model Inter-Comparison Study for Asia Phase III was conducted to evaluate ozone ($O_3$) and other gaseous pollutants, fine particular matter ($PM_{2.5}$), and acid and reactive nitrogen deposition with various models with/out ARI or/and ACI (Li et al., 2019; Chen et al., 2019; Itahashi et al., 2020; Ge et al. al., 2020; Kong et al., 2020). With respect to this project, Gao et al. (2018, 2020) have reviewed in detail the model performance of seven two-way coupled models from different research groups in simulating a heavy air pollution episode during January 2010 in North China Plain and how aerosol feedbacks affected simulations of meteorological variables and $PM_{2.5}$ concentrations. Targeting the heavy polluted India region, Govardhan et al. (2016) compared aerosol optical depth (AOD) and various aerosol species (black carbon, mineral dust, and sea salt) modeled by WRF-Chem (with ARI) and Spectral Radiation-Transport Model for Aerosol Species (with both ARI and ACI), but under different model configurations.

So far, there is no comprehensive comparisons of multiple coupled models under the same model configuration with respect to the high aerosol loading region over eastern China, where has experienced rapid growth of economy, urbanization, population, as well as severe air quality problems in the past decades (He et al., 2002; Wang and Hao, 2012; Gao et al., 2017; Geng et al., 2021). In the eastern China region (ECR), several open-source and proprietary two-way coupled models have been applied



89 to investigate the ARI and/or ACI effects, yet most studies have focused on certain
90 short-term episodes of heavy air pollution without any year-long simulations (Xing et
91 al., 2017; Ding et al., 2019; Ma et al., 2021). The commonly used open-source models
92 in ECR are WRF-Chem and WRF-CMAQ (Grell et al., 2005; Wong et al., 2012), but
93 there is no any application of the two-way coupled WRF-CHIMERE model that has
94 been applied to examine aerosol-radiation-cloud interactions in Europe and Africa
95 (Briant et al., 2017; Tuccella et al., 2019). At the same time, model simulations should
96 be compared not only against surface measurement data but also satellite data (Zhao et
97 al., 2017; Hong et al., 2017; Campbell et al., 2017; Wang et al., 2018). Even though the
98 running time of an individual modeling system (e.g., WRF-CMAQ and WRF-
99 CHIMERE) was evaluated by considering its online and offline versions and under
100 various computing configurations (Wong et al., 2012; Briant et al., 2017), the
101 computational efficiencies of multiple two-way coupled models need to be accessed
102 under the same computing conditions as well.

103 In this paper, a comparative evaluation of three open-sourced two-way coupled
104 meteorology and air quality models (WRF-CMAQ, WRF-Chem and WRF-CHIMERE)
105 in ECR is conducted. The remainder of the paper is organized as follows: Section 2
106 describes the study methods including model configurations and evaluation protocols.
107 Sections 3 and 4 presents the analyses and intercomparisons of simulations from these
108 three two-way coupled models with regard to meteorology and air quality, respectively.
109 The major findings of this work are summarized in Section 5.

110

## 2 Data and methods

### 2.1 Model configurations and data sources

113 One-year long-term simulations in eastern China were examined using the two-
114 way coupled WRF v4.1.1-CMAQ v5.3.1, WRF-Chem v4.1.1, and WRF v3.7.1-
115 CHIMERE v2020r1 models, with and without enabling ARI and/or ACI, and with 27-
116 km horizontal grid spacing (there were 110, 120, and 120 grid cells in the east–west
117 direction, and 150, 160, and 170 in the north–south direction for WRF-CMAQ, WRF-
118 Chem, and WRF-CHIMERE, respectively). The vertical resolution for all simulations
119 consisted of 30 levels from the surface (~20 m) to 100 hPa. The anthropogenic
120 emissions of Multi-resolution Emission Inventory for China (MEIC) (Li et al., 2017)
121 and FINN v1.5 biomass burning emissions were applied in our simulations, and their
122 spatial, temporal, and species allocations were performed using Python language.
123 Biogenic emissions were calculated using the Model of Emissions of Gases and
124 Aerosols from Nature version 3.0 (MEGAN v3.0) (Gao et al., 2019). Dust and sea-salt
125 emissions were both used with calculations of inline modules, as shown in Table 1. The
126 meteorological ICs and BCs were derived from the National Center for Environmental
127 Prediction Final Analysis (NCEP-FNL) datasets (http://rda.ucar.edu/datasets/ds083.2),
128 with a horizontal resolution of 1° × 1° at 6-hour intervals for each of the three coupled
129 models. To improve the long-term accuracy of meteorological variables when using the
130 WRF model, options of observational and grid four-dimensional data assimilation
131 (FDDA) were turned on, and pressure, station height, relative humidity, wind speed,



and wind direction were observed four times per day at 00:00, 06:00, 12:00, and 18:00
UTC from 2168 stations (https://doi.org/10.5281/zenodo.6975602). The chemical
ICs/BCs were downscaled from the Whole Atmosphere Community Climate Model
(WACCM) for WRF-CMAQ and WRF-Chem via the mozart2camx and mozbc tools,
respectively. The options of parameterization schemes of aerosol–radiation–cloud
interactions are summarized in Table 1. It should be noted that ACI processes cannot be
implemented in the official release of WRF-CMAQ.
To demonstrate the capabilities of the three two-way coupled models with/without
feedbacks in simulating meteorology and air quality, we undertook comprehensive
evaluations of the strengths and weaknesses each coupled model, validated against
extensive ground-based and satellite measurements. Ground-based data included 572
hourly ground-based meteorological observations (air temperature (T2) and relative
humidity (RH2) air temperature at 2m above the surface, wind speed at 10m above the
surface (WS10), and precipitation (PREC)) (http://data.cma.cn), 327 hourly national
environmental observations (fine particulate matter ($PM_{2.5}$), ozone ($O_3$), nitrogen
dioxide ($NO_2$), sulfur dioxide ($SO_2$), and carbon monoxide (CO))
(http://106.37.208.233:20035), 109 hourly surface shortwave radiation (SSR)
measurements (Tang et al., 2019) and 74 radiosonde sites retrieved twice per day (Guo
et al., 2019); the locations of these data are depicted in Figure 1. Because there were no
observed water vapor mixing ratio (w) data, this parameter was calculated via the
formula $w = \dfrac{rh}{w_s}$, where rh is the relative humidity and $w_s$ is the saturation mixing ratio
(Wallace and Hobbs, 2006).
Satellite data included the following: monthly average downwelling short-/long-
wave flux at the surface and short-/long-wave flux at the top of the atmosphere (TOA)
from the Clouds and the Earth's Radiant Energy System (CERES)
(https://ceres.larc.nasa.gov); precipitation from the Tropical Rainfall Measuring
Mission (TRMM); cloud fraction, liquid water path (LWP), and aerosol optical depth
(AOD) from the Moderate Resolution Imaging Spectroradiometer (MODIS);
tropospheric $NO_2$ column and $SO_2$ column in the planetary boundary layer (PBL) from
the Ozone Monitoring Instrument (OMI); total CO column from the Measurements of
Pollution in the Troposphere (MOPITT) (https://giovanni.gsfc.nasa.gov/giovanni);
total column ozone (TCO) from the Infrared Atmospheric Sounding Interferometer-
Meteorological Operational Satellite-A (IASI-METOP-A)
(https://cds.climate.copernicus.eu/cdsapp#!/dataset/satellite-ozone?tab=form); and
total ammonia ($NH_3$) column from IASI-METOP-B (https://cds-
espri.ipsl.fr/iasibl3/iasi_nh3/V3.1.0). These data were downloaded and interpolated to
the same horizontal resolution as the model results using Rasterio library (Gillies et al.,
2013), then the model and observed values at each grid point were extracted.

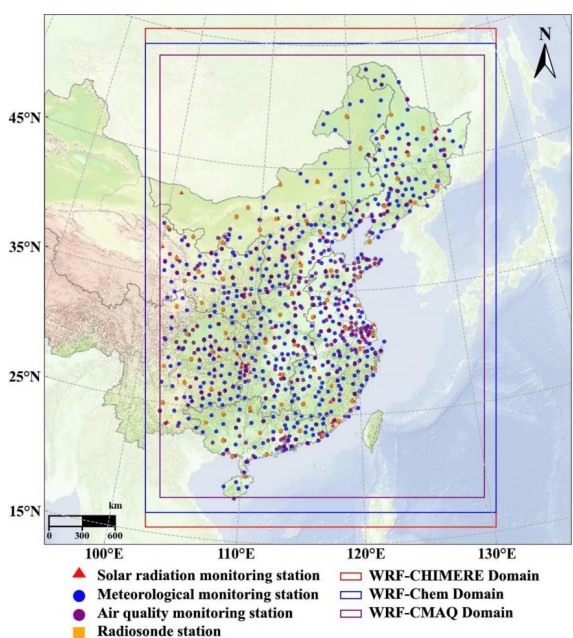


Figure 1. Modeling domains (WRF-CMAQ, WRF-Chem, and WRF-CHIMERE), and solar
radiation, meteorology, air quality, and radiosonde stations.

Table 1. Model configurations and parameterization schemes.

| Configurations | WRF-CMAQ | WRF-Chem | WRF-CHIMERE |
|---|---|---|---|
| Horizontal grid spacing | 27 km (110 × 150) | 27 km (120 × 160) | 27 km (120 × 170) |
| Vertical grid | 30 levels | 30 levels | 30 levels |
| Shortwave radiation | RRTMG | RRTMG | RRTMG |
| Longwave radiation | RRTMG | RRTMG | RRTMG |
| Aerosol mixing state | Core-Shell | Core-Shell | Core-Shell |
| Cloud microphysics | Morrison | Morrison | Thompson |
| PBL | ACM2 | YSU | YSU |
| Cumulus | Kain-Fritsch | Grell-Freitas | Grell-Freitas |
| Surface | Pleim-Xiu | Monin-Obukhov | Monin-Obukhov |
| Land surface | Pleim-Xiu LSM | Noah LSM | Noah LSM |
| Gas-phase chemistry | CB6 | CBMZ | MELCHIOR2 |
| Photolysis | Fast-JX | Fast-JX | Fast-JX |
| Aerosol mechanism | AERO6 | MOSAIC 4BIN | SAM 10BIN |
| Anthropogenic emission | MEIC 2017 | MEIC 2017 | MEIC 2017 |
| Biogenic emission | MEGAN v3.0 | MEGAN v3.0 | MEGAN v3.0 |
| Biomass burning emission | FINN v1.5 | FINN v1.5 | FINN v1.5 |
| Dust emission | Foroutan | GOCART | Menut |
| Sea-salt emission | Gong | Gong | Monahan |
| Meteorological ICs and BCs | FNL | FNL | FNL |
| Chemical ICs and BCs | MOZART | MOZART | LMDZ-INCA |





## 2.2 Scenario set up

Eight sets of hindcast WRF-CMAQ, WRF-Chem, and WRF-CHIMERE simulations with/without aerosol feedbacks were carried out to investigate the performance of each coupled model over eastern China during 2017, as presented in Table 2. It should be noted that the officially released WRF-Chem and WRF-CHIMERE are capable of simulating ARI and ACI, but WRF-CMAQ is not. In all of the simulations performed in this study, a month of spin-up time was set up to reduce the influence of the initial conditions. We calculated multiple model evaluation metrics between each scenario simulation and relevant observations to assess the model performance; these included the correlation coefficient (R), mean bias (MB), normalized mean bias (NMB), and root mean square error (RMSE). The mathematical definitions of these metrics are provided in Supplement S1. We comprehensively analyzed the annual and seasonal statistical metrics of meteorological and air quality variables including simulations by all three two-way coupled models with/without enabling ARI and/or ACI effects. We then quantified the respective contributions of the ARI and ACI effects to model performance.

Table 2. Summary of scenarios setting in three coupled models.

| Model | Scenario | Configuration option | Description |
|---|---|---|---|
| WRF-CMAQ | (1) WRF-CMAQ_NO | DO_SW_CAL=F | Without aerosol feedbacks |
| | (2) WRF-CMAQ_ARI | DO_SW_CAL=T | ARI |
| WRF-Chem | (3) WRF-Chem_NO | aer_ra_feedback=0 | Without aerosol feedbacks |
| | | wetscav_onoff=0 | |
| | | cldchem_onoff=0 | |
| | (4) WRF-Chem_ARI | aer_ra_feedback=1 | ARI |
| | | wetscav_onoff=0 | |
| | | cldchem_onoff=0 | |
| | (5) WRF-Chem_BOTH | aer_ra_feedback=1 | ARI and ACI |
| | | wetscav_onoff=1 | |
| | | cldchem_onoff=1 | |
| WRF-CHIMERE | (6) WRF-CHIMERE_NO | direct_feed_chimere=0 | Without aerosol feedbacks |
| | | indirect_feed_chimere=0 | |
| | (7) WRF-CHIMERE_ARI | direct_feed_chimere=1 | ARI |
| | | indirect_feed_chimere=0 | |
| | (8) WRF-CHIMERE_BOTH | direct_feed_chimere=1 | ARI and ACI |
| | | indirect_feed_chimere=1 | |

## 3 Meteorological evaluations and intercomparisons

This section presents annual and seasonal (March–April–May, Spring; June–July–August, Summer; September–October–November, Autumn; and December–January–February, Winter) statistical metrics of simulated meteorological variables and air quality when compared with ground-based and satellite observations, as well as a





discussion of the running times of the eight scenario simulations.
3.1 Ground-based observations
Figures 2 and S1–S7 illustrate comparisons of the spatial distributions of R, MB,
and RMSE for hourly SSR, T2, Q2, RH2, WS10, PREC, PBLH00, and PBLH12 from
WRF-CMAQ, WRF-Chem, and WRF-CHIMERE with/without turning on aerosol
feedbacks against ground-based observations from each site across the whole of 2017.
The calculated annual model evaluation metrics for all sites in eastern China are
summarized in Table S1, and the related seasonal R and MB values are presented in
Figure 3.
The accuracy of radiation predication is of great significance in evaluating ARI.
Yearly and seasonal average simulated SSR data were explicitly compared with ground-
based observations (Figure 3 and Table S1); the SSR over eastern China was simulated
reasonably well by all models with R values in the range of 0.61–0.78. The overall
model performances of WRF-CMAQ and WRF-Chem were better than that of WRF-
CHIMERE, while all simulated results were overestimated at both annual and seasonal
scales (MBs in spring and summer were larger than those in autumn and winter). The
overestimations of annual SSR were 19.98, 14.48, and 9.24 W m$^{-2}$ for WRF-CMAQ,
WRF-Chem, and WRF-CHIMERE, respectively. Overestimations of SSR by most two-
way coupled models were also reported for Europe and North America in the
comparative study conducted by Brunner et al. (2015). Such overestimations could be
explained by multiple factors, namely, the uncertainties in cloud development owing to
PBL and convection parameterizations (Alapaty et al., 2012), and the diversity in
treatment of land surface processes (Brunner et al., 2015), which appear to play more
important roles than does the enabling of two-way aerosol feedbacks on SSR through
ARI and ACI effects in the models. When the three models considered ARI effects, they
effectively improved the simulation accuracy of SSR, over both the whole year and in
the four seasons, but the enabling of ACI effects resulted in relatively limited
improvement. In addition, the MB variations of WRF-CMAQ and WRF-Chem
simulations were higher in spring and winter than those in summer and autumn, while
the MB of WRF-CHIMERE simulations showed a maximum in summer (−10.33 W
m$^{-2}$) and minimum in autumn (−7.64 W m$^{-2}$). Both the annual and seasonal reductions
in SSR simulated by WRF-Chem and WRF-CHIMERE with ACI effects enabled were
much smaller than those with ARI effects enabled.
In general, the simulated magnitudes and temporal variations of air temperature
and water vapor mixing ratio at 2 m above the ground showed a high order of
consistency with observations (R = 0.88–0.97). Looking at annual and seasonal T2,
models tended to have a negative (cool) bias, and T2 underestimations in spring and
winter were greater than those in summer and autumn. As pointed out by Makar et al.
(2015a), WRF-CHEM and GEM-MACH gave negative MBs in summer and positive
MBs in winter when both ACI and ARI effects were enabled (BOTH), and WRF-
CMAQ with only ARI effects enabled also produced negative MBs in summer over
North America during 2010; note that the Makar et al (2015a) study lacked evaluations
of meteorology in winter using WRF-CMAQ. The comparison results of MBs indicated
that WRF-CHIMERE > WRF-CMAQ > WRF-Chem. The annual and seasonal MBs of



WRF-CMAQ and WRF-Chem were approximately −1 ℃, while those of WRF-
CHIMERE ranged from −2 to −1 ℃. The RMSEs were approximately equal for WRF-
CMAQ (2.71–3.05 ℃) and WRF-Chem (2.82–3.27 ℃), and larger for WRF-
CHIMERE (3.39–4.53 ℃), at both annual and seasonal scales. It is noteworthy that
underestimations of annual and seasonal T2 were mitigated in eastern China in the three
coupled models when ARI effects were enabled. When ACI effects were enabled, the
MBs for T2 simulated by WRF-Chem_BOTH showed no significant changes compared
with those of WRF-Chem_NO; WRF-CHIMERE_BOTH further enhanced the
underestimations of T2 in the full year (−1.30 ℃), spring (−0.12 ℃), and winter
(−0.40 ℃) compared with WRF-CHIMERE_NO.
For Q2, WRF-CMAQ showed the best performance, followed by WRF-Chem, and
WRF-CHIMERE (Table S1 and Figure S2), all with RMSEs of less than 3 g kg$^{-1}$. Most
models tended to underestimate annual and seasonal Q2 (−0.57 to −0.18 g kg$^{-1}$ and
−1.16 to +0.20 g kg$^{-1}$, respectively), and the underestimations were most significant in
summer. However, multiple two-way coupled models produced slightly positive values
for Q2 during January 2010 over the North China Plain in the MICS-Asia III project
(Gao et al., 2018). Compared with simulations that did not have aerosol feedbacks
enabled, WRF-CMAQ_ARI and WRF-CHIMERE_ARI increased the negative biases
of annual and seasonal Q2, with the former being more significant (Figure 3 and Table
S1). The changes in annual, summer, and autumn MBs for WRF-Chem_ARI were
consistent with the trend of WRF-CMAQ_ARI, except for spring and winter.
Looking at RH2, annual and seasonal simulations using WRF-CMAQ had the
highest correlation with the observed values, followed by WRF-Chem, and WRF-
CHIMERE, and the smallest correlation coefficients for all three models occurred in
autumn (~ 0.5). The spatial MBs between simulations by the three models and
observations showed a general converse trend compared with T2 (i.e., RH2 was
overestimated where T2 was underestimated, and vice versa). This can be explained by
the calculation of RH2 being based on T2 in the models (Wang et al., 2021). The annual
and seasonal MBs were approximately 0.65%–71.03% and −21.30% to 60.00%,
respectively (Figure 3 and Table S1), and only WRF-Chem produced negative MBs in
summer. The magnitude of RMSE showed an inverse pattern compared with R for all
three models, with maximum (28.48%–29.52%) and minimum (12.57%–16.07%)
values shown in autumn and summer, respectively. As shown in Figure 3 and Table S1,
WRF-CMAQ_ARI further reduced the overestimations of annual and seasonal RH2 in
eastern China, while WRF-Chem_ARI (except for summer) and WRF-CHIMERE_ARI
showed the opposite trend. Moreover, variations in annual and seasonal RH2 MBs
simulated by WRF-Chem_BOTH and WRF-CHIMERE_BOTH were further reduced
compared with WRF-Chem_ARI (except for summer) and WRF-CHIMERE_ARI,
respectively.
Similar analyses were also performed for WS10, and revealed that WRF-CMAQ
performed better in capturing WS10 patterns compared with WRF-Chem and WRF-
CHIMERE. The R values for all three models ranged from 0.47 to 0.60; WRF-CMAQ
and WRF-Chem overestimated wind speed by approximately 0.5 m s$^{-1}$, while WRF-
CHIMERE overestimated it by approximately 1.0 m s$^{-1}$. The overestimation of WS10



under real-world low wind conditions is a common phenomenon of current weather
models, which is mainly caused by outdated geographic data, coarse model resolution,
and a lack of a good physical representation of the urban canopy (Gao et al., 2015; Gao
et al., 2018). All three models presented lower correlations (0.31–0.54) and MBs (0.20–
0.86 m s$^{-1}$) in summer compared with other seasons, and the RMSEs were
approximately 2.0 m s$^{-1}$. When ARI effects were enabled, the overestimations of the
three models were alleviated, especially for WRF-CMAQ_ARI.
The annual and seasonal correlation coefficients of precipitation were 0.56–0.69,
0.46–0.63, and 0.25–0.55 for WRF-CMAQ, WRF-Chem, and WRF-CHIMERE,
respectively (Table S1 and Figure S5). All simulated results had the highest correlations
in winter and the lowest in summer, because the convective activity was enhanced in
summer and the models struggle to effectively capture this. WRF-CMAQ and WRF-
CHIMERE (WRF-Chem except for autumn) underestimated and overestimated annual
and seasonal precipitation, respectively. At the annual and seasonal scales, WRF-Chem
and WRF-CHIMERE overestimated the magnitude of daily precipitation by more than
1 mm day$^{-1}$, while WRF-CMAQ underestimated it by approximately 0.5 mm day$^{-1}$. A
similar picture emerged for North America during 2010, whereby the magnitude of
precipitation MBs was higher in WRF-Chem than in WRF-CMAQ (see figure 11 in
Makar et al., 2015a). The largest precipitation MBs simulated by the three models
occurred in summer and ranged from −0.70 to + 1.39 mm day$^{-1}$. The RMSE was highest
in WRF-CHIMERE, followed by WRF-Chem, and WRF-CMAQ, and all models had
the largest (> 10 mm day$^{-1}$) and smallest (approximately 2.5 mm day$^{-1}$) values in
summer and winter, respectively. When ARI effects were considered, WRF-
CMAQ_ARI simulations increased the annual and seasonal precipitation
underestimations in eastern China, while WRF-Chem_ARI (except for autumn) and
WRF-CHIMERE_ARI simulations reduced the precipitation overestimations. The
effects of ARI on summer MBs using all three coupled models were significant
compared with other seasons. When ACI effects were further included, WRF-
Chem_BOTH showed very limited improvement in the overestimation of precipitation
compared with WRF-Chem_NO, while WRF-CHIMERE_BOTH enhanced the
overestimation of precipitation, especially in summer.
Overall, PBLH data were not well reproduced by any of the three coupled models,
which may be a result of the low resolution of the sounding data (Brunner et al., 2015)
and the different settings of Richardson number thresholds in calculating PBLH (Guo
et al., 2016). At 8:00 and 20:00 local time (LT), annual and seasonal PBLH simulated
by WRF-CMAQ had the highest correlations (R = 0.21–0.40) and largest negative MBs
(ranging from −400 to −133 m). The poor performance was mainly caused by: 1)
different configurations of the PBL scheme in this study, namely, WRF-CMAQ adopted
the ACM2 scheme with hybrid local–nonlocal closure, while WRF-Chem and WRF-
CHIMERE adopted the YSU scheme with non-local closure (Table 1); 2) the settings
of the Richardson number threshold varied owing to the unstable atmospheric
conditions, i.e., the YSU and ACM2 schemes used thresholds of 0 and 0.25,
respectively (Xie et al., 2012); 3) the entrainment layer was further considered in the
ACM2 scheme for PBLH calculations (Xie et al., 2012).





Meanwhile, all correlations of the three models at 20:00 LT (R = 0.3–0.4) were
better than those at 8:00 LT (R = 0.1–0.2), because the gradient of the rapid increase in
PBLH in the morning was larger than that of the gradual decrease in PBLH at night,
and hence more difficult to accurately simulate. In addition, the RMSEs of PBLH in
autumn (369.89–388.79 m) and winter (347.48–392.38 m) were smaller than those in
spring (405.61–622.37 m) and summer (348.80–570.16 m) for all three models. As
shown in Figure 3 and Table S1, the effects of aerosol feedbacks on MB and RMSE
were larger than that on R. Considering that the MBs of PBLH are important for the
simulation of air quality, the MBs were further analyzed here. For WRF-CMAQ, ARI
effects induced an increase (−1.93 m) and decrease (+6.66 m) in the annual
underestimations at 8:00 and 20:00 LT, respectively (Table S1). The negative MBs for
WRF-Chem_ARI and WRF-Chem_BOTH showed an increase (8:00 LT: −25.25 m,
20:00 LT: −25.60 m) and decrease (8:00 LT: +19.65 m, 20:00 LT: +14.09 m) compared
with those for WRF-Chem_NO and WRF-Chem_ARI, respectively. The ARI (−6.17
and −3.34 m) and ACI (−0.65 and −1.11 m) effects both further underestimated annual
PBLH at 8:00 and 20:00 LT for WRF-CHIMERE. The variations in MBs induced by
aerosol feedbacks for the three coupled models at the annual scale were similar to those
at the seasonal scale.

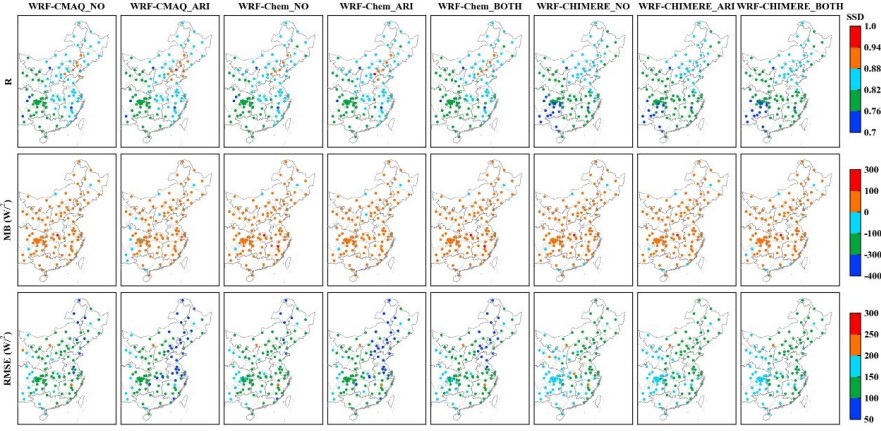


Figure 2. Statistical metrics (R, MB, and RMSE) between annual simulations and observations of
surface shortwave radiation in eastern China.



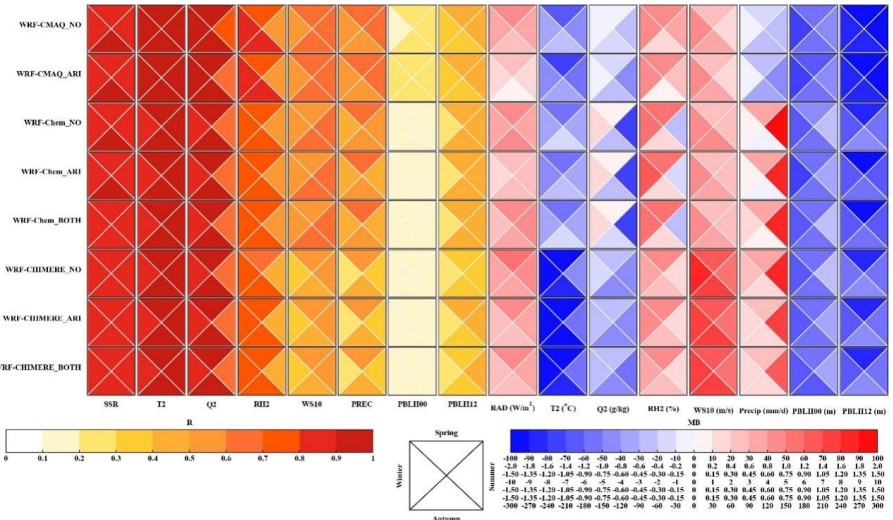

Figure 3. Portrait plots of statistical indices (R and MB) between seasonal simulations and surface observations of meteorological variables (SSR, T2, Q2, RH2, WS10, PREC, and PBLH at LT 08:00 and 20:00) in eastern China.

To identify and quantify how well our results compare with previous studies using two-way coupled models, we here discuss comparisons between our work and earlier research in terms of the evaluation results of meteorology and air quality; meteorology is discussed in this section and air quality is discussed in Section 4.1. Box-and-whisker plots were used and the 5th, 25th, 75th, and 95th percentiles were used as statistical indicators. In the plots, the dashed lines in the boxes are the mean values, and the circles represent outliers. Previous studies mainly used WRF-Chem and WRF-CMAQ to evaluate meteorology and air quality, while applications of WRF-NAQPMS and GRAPES-CUACE were scarce. As mentioned in Section 1, investigations of meteorology and air quality using WRF-CHIMERE with/without aerosol feedbacks have not previously been conducted in eastern China. Therefore, only evaluation results involving WRF-Chem and WRF-CMAQ to study aerosol feedbacks are analyzed herein.

Figure 4 illustrates the statistical metrics of T2, RH2, Q2, and WS10 in this study compared with the evaluation results of previous studies. According to the number of samples (NS) in the statistical metrics of each meteorological variable, most previous studies mainly involved the simulation and evaluation of T2, WS10, and RH2, with relatively few studies focusing on Q2. Compared with the evaluation results of previous studies, the ranges of statistical metrics in our study were roughly similar, but there were some important differences. The R values of the WRF-CMAQ and WRF-Chem models in our study were higher than those of previous studies; the MBs of T2 simulated via WRF-CMAQ were smaller, but those of T2 simulated via WRF-Chem were larger; and the RMSEs of the WRF-CMAQ simulation were larger, but those of the WRF-Chem simulation were smaller. For RH2, the R values for WRF-CMAQ and WRF-Chem in this study were all larger than the average level of previous studies, while the



MBs and RMSEs for WRF-CMAQ were larger, and those for WRF-Chem were smaller
than the average of previous studies. For Q2, the model performance of WRF-CMAQ
in this study was generally better than the average level of previous studies, but the R
between WRF-Chem simulation results and observed values was higher (and MB and
RMSE were lower) than the average level of previous studies. We also conclude that
the simulation results of WRF-CMAQ and WRF-Chem in our study better reproduced
variations in WS10 compared with previous studies (Fig. 4d).

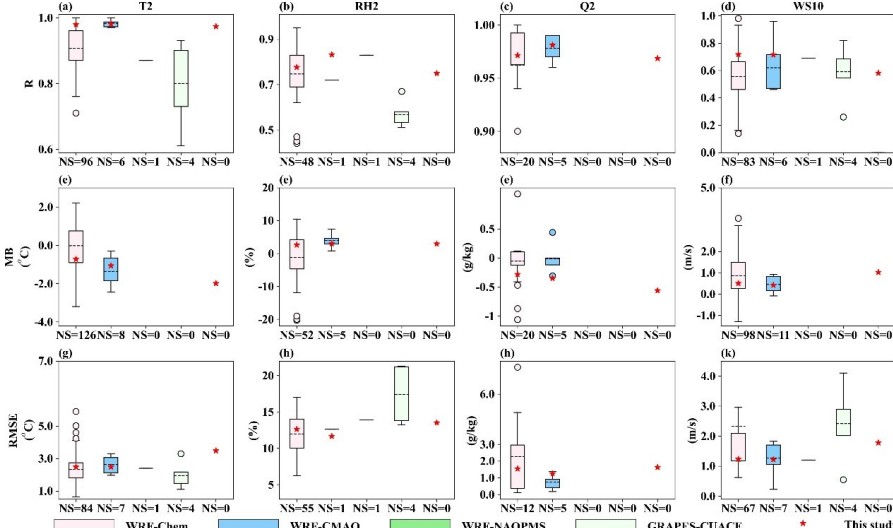

Figure 4. Comparisons of model capacities between our study (red stars) and previous literature
(box plots) in terms of the surface T2, RH2, Q2, and WS10 in eastern China. Note that red stars in
the fifth column of each subgraph represent the statistical metrics of WRF-CHIMERE in this study.

3.2 Satellite-borne observations
To further evaluate the wider spatial performance of WRF-CMAQ, WRF-Chem,
and WRF-CHIMERE, we analyzed the annual and seasonal statistical metrics of short-
and long-wave radiation at the surface and top of the atmosphere (TOA), precipitation,
cloud cover, and liquid water path simulated by the three coupled models with and
without aerosol feedbacks, via comparisons between simulations and satellite-borne
observations (Table 3; Figures 5, S8–S13).
As listed in Table 3, the three coupled models predicted the longwave radiation
variables at the surface (SLR) and top of the atmosphere (TLR) well (R values of 0.74
to 0.99), with annual domain-average MBs of −9.97 to −6.05 W m$^{-2}$ and −2.14 to 0.66
W m$^{-2}$, respectively. The annual SLRs were underestimated by all three models, and
the MBs of WRF-CMAQ (−6.46 to −6.05 W m$^{-2}$) were smaller than those of WRF-
CHIMERE (−9.66 to −8.39 W m$^{-2}$) and WRF-Chem (−9.97 to −9.34 W m$^{-2}$). For
annual TLR, the simulation results of WRF-CHIMERE (−0.96 to +0.05 W m$^{-2}$) and
WRF-CMAQ (−2.14 to −1.42 W m$^{-2}$) provided underestimations, but WRF-Chem
(0.28–0.71 W m$^{-2}$) gave overestimations. Significant seasonal differences in simulated



longwave radiation were also present among the three coupled models; all WRF-CMAQ and WRF-CHIMERE scenarios gave underestimations, with maximum and minimum values of SLR in winter and summer, respectively, while the maximum underestimations of WRF-Chem occurred in autumn, especially for WRF-Chem_BOTH (Figure S8). For seasonal TLR, the WRF-CMAQ and WRF-Chem model performances were better than that of WRF-CHIMERE for all seasons except autumn (Figure S9).

Compared with longwave radiation, the three coupled models showed poorer performance for the shortwave radiation variables at the surface (SSR) and top of the atmosphere (TSR) with annual MBs of 8.21–30.74 W m$^{-2}$ and −4.40 to +5.42 W m$^{-2}$, respectively, and correlations ranging from 0.61 to 0.92 for both variables. A similar poor performance for shortwave radiation compared with longwave radiation was also reported in the USA using the coupled WRF-CMAQ and offline WRF models (Wang et al., 2021). The overall seasonal characteristics of SSR were successfully reproduced by the three coupled models (Figure S10). Meanwhile, no matter whether aerosol feedbacks were enabled or not, all three models overestimated seasonal SSR (except for WRF-Chem_ARI in winter), and showed higher MBs in spring and summer than in autumn and winter. The seasonal SSR overestimations may be a direct result of the underestimation of calculated AOD when considering ARI effects (Wang et al., 2021). Seasonal TSR was also successfully simulated by all three models, especially in winter (Figure S11). No matter whether ARI and/or ACI effects were enabled or not, WRF-CMAQ had negative MBs in all seasons, WRF-CHIMERE had negative MBs in all seasons except for spring, and WRF-Chem produced underestimations and overestimations of TSR in spring–summer and autumn–winter, respectively.

As all three coupled models adopted the same grid resolution (27 × 27 km) and short- and long-wave radiation schemes (RRTMG), the above analysis demonstrates that the configurations of different aerosol/gas chemical mechanisms contributed to the diversity of seasonal MBs. Moreover, the three two-way coupled models with ARI feedbacks enabled effectively improved the performances of annual and seasonal SSR; however, for SLR, TLR, and TSR, performance improvements were much more variable across the three coupled models and across different scenarios with and without ARI and/or ACI feedbacks enabled. Further details on this can be found in Table S2.

From IPCC 2007 to IPCC 2021, the effects of aerosol feedbacks (especially for ACI effects) on precipitation and cloud processes remain under debate. Here, we further assessed annual and seasonal simulated precipitation, cloud cover, and liquid water pathways in eastern China with high aerosol loadings against satellite observations (Table 3 and Figures S11–S13), and attempted to provide new insights from a yearly perspective into enabling online feedbacks in two-way coupled modeling simulations.

The results illustrated that correlations of precipitation via WRF-CMAQ (0.51–0.89) were larger than those of WRF-Chem (0.61–0.73) and WRF-CHIMERE (0.54–0.70). WRF-CMAQ had the best correlation in winter, while WRF-Chem and WRF-CHIMERE had the best correlation in spring; all three models showed their worst correlation in summer. The reason for this is that numerical models struggle to effectively capture enhanced convective activity in summer. Huang and Gao (2018)



also pointed out that accurate representations of lateral boundaries are crucial in improving precipitation simulations during summer over China. WRF-CMAQ underestimated annual precipitation, with MBs of −76.49 to −51.93 mm, while WRF-Chem and WRF-CHIMERE produced large precipitation overestimations ranging from +108.04 to +207.05 mm (Table 3), especially in regions of southern China (Figure S11). WRF-CMAQ also produced negative biases (−27.89 to +42.08 mm) of seasonal precipitation, excluding WRF-CMAQ_ARI in winter. WRF-Chem and WRF-CHIMERE only underestimated seasonal precipitation in autumn (−31.39 to −26.89 mm) and winter (−7.12 to −4.43 mm), respectively (Figure S11). The variations in annual and seasonal MBs of precipitation were consistent with changes in cloud fraction and LWP (Zhang et al., 2016), which will be discussed in more detail below.

When aerosol feedbacks were considered, the ARI-induced reductions in the annual MBs of precipitation for WRF-CMAQ, WRF-Chem, and WRF-CHIMERE were 24.56, 12.11, and 4.70 mm, respectively. WRF-Chem_BOTH (24.9 mm) and WRF-CHIMERE_BOTH (3.41 mm) enhanced the overestimation of annual precipitation compared with WRF-Chem_ARI and WRF-CHIMERE_ARI, respectively. Significant increases (+53.15 mm) and decreases (−6.3 to −3.41 mm) in MBs in winter and summer, respectively, were produced by WRF-CMAQ and the other two models with ARI effects enabled compared with no feedbacks. WRF-Chem and WRF-CHIMERE with both ARI and ACI effects enabled led to larger enhancements of MBs (+3.54 to +7.46) at the seasonal scale (Figure S11). It must be noted that the discrepancies in simulated precipitation could mainly be attributed to the selection of different microphysics and cumulus schemes in WRF-CMAQ (Morrison and Kain-Fritsch), WRF-Chem (Morrison and Grell-Freitas), and WRF-CHIMERE (Thompson and Grell-Freitas).

Cloud fraction (CF) and LWP can significantly influence the spatiotemporal distributions of precipitation; our simulated results of annual and seasonal CF over eastern China are presented in Table 3 and Figure S12. Overall, WRF-CMAQ performed best in simulating CF. The R values for WRF-Chem during summer (0.69) and winter (0.70) were larger than those of WRF-CMAQ (0.59 and 0.64) and WRF-CHIMERE (0.56 and 0.66), while WRF-CMAQ and WRF-CHIMERE showed better simulation results in winter and autumn with correlations of up to 0.89 and 0.67, respectively. All three coupled models underestimated annual and seasonal CF with MBs that ranged from −16.83% to −6.18% and −21.13% to −4.13%, respectively; these were consistent with previous two-way coupled modeling studies using WRF-CMAQ (−19.7%) and WRF-Chem (−32% to −9%) in China (Hong et al., 2017; Zhao et al., 2017).

All models reasonably simulated annual LWP in eastern China, with R values above 0.55 and negative biases varying from −57.36 to −31.29 g m$^{-2}$. The underestimations were closely related to missing cloud homogeneity (Wang et al., 2015; Dionne et al., 2020) and excessive conversion of liquid to ice in all selected cloud microphysics schemes (Klein et al., 2009). As shown in Figure S13, all models showed their best performance in simulating LWP in spring (R = 0.51–0.79) and exhibited the largest underestimations in winter (MBs of −54.82 to −40.89 g m$^{-2}$), except for WRF-Chem, which had its maximum bias in autumn.





In terms of quantitatively determining the functions of aerosol feedbacks on CF
and LWP, all simulated scenarios revealed that WRF-CMAQ_ARI overwhelmingly
decreased annual and seasonal underestimations of CF (0.48%–1.05%) and LWP (3.03–
4.29 g m$^{-2}$), while there were slightly increased underestimations (CF: 0.02%–0.39%;
LWP: 0.03–0.58 g m$^{-2}$) in WRF-Chem_ARI and WRF-CHIMERE_ARI. Larger
variations in annual and seasonal MBs of CF (0.23%–0.93%) and LWP ($-2.96$ g m$^{-2}$ to
7.38 g m$^{-2}$) were produced by WRF-CHIMERE_BOTH compared with WRF-
CHIMERE_ARI. WRF-Chem_BOTH showed equivalent variations (CF: 0.03%–
0.71%; LWP: 0.02–2.89 g m$^{-2}$) to those of WRF-Chem_ARI. Although we have
obtained preliminary quantitative results of the ACI effects on regional precipitation,
CF, and LWP, it should be kept in mind that several limitations in representing ACI
effects still exist in state-of-the-art two-way coupled models; these include a lack of
consideration of the responses of convective clouds to ACI (Tuccella et al., 2019), and
a lack of numerical descriptions of giant cloud condensation nuclei (Wang et al., 2021)
and heterogeneous ice nuclei (Keita et al., 2020).
Table 3. Statistical metrics (R, MB, NMB, and RMSE) between annual simulations and
satellite retrievals of surface shortwave and longwave radiation, TOA shortwave and
longwave radiation, precipitation, cloud fraction, and liquid water path in eastern China.
The best results are in bold, while mean simulations and observations are in italics.

| Variables | Statistics | WRF-CMAQ_NO | WRF-CMAQ_ARI | WRF-Chem_NO | WRF-Chem_ARI | WRF-Chem_BOTH | WRF-CHIMERE_NO | WRF-CHIMERE_ARI | WRF-CHIMERE_BOTH |
|---|---|---|---|---|---|---|---|---|---|
| Surface shortwave radiation (*172.74* W m$^{-2}$) | Mean_sim | *197.15* | *180.94* | *203.48* | *194.52* | *201.45* | *197.39* | *191.34* | *195.58* |
| | R | 0.76 | 0.75 | 0.73 | **0.78** | 0.75 | 0.61 | 0.64 | 0.66 |
| | MB | 24.41 | 8.21 | 30.74 | 21.78 | 28.71 | 24.75 | 18.71 | 22.94 |
| | NMB (%) | 14.13 | **4.75** | 17.79 | 12.61 | 16.62 | 14.34 | 10.84 | 13.29 |
| | RMSE | 30.25 | **20.37** | 35.34 | 26.88 | 32.80 | 34.70 | 29.60 | 31.45 |
| Surface longwave radiation (*322.3* W m$^{-2}$) | Mean_sim | *316.25* | *315.83* | *312.96* | *312.60* | *312.32* | *313.33* | *314.60* | *314.47* |
| | R | 0.98 | 0.98 | 0.98 | 0.98 | 0.98 | **0.99** | **0.99** | **0.99** |
| | MB | -6.05 | -6.46 | -9.34 | -9.70 | -9.97 | -9.66 | -8.39 | -8.53 |
| | NMB (%) | **-1.88** | -2.00 | -2.90 | -3.01 | -3.09 | -2.99 | -2.60 | -2.64 |
| | RMSE | **13.65** | 14.13 | 14.81 | 14.97 | 15.17 | 15.47 | 14.52 | 14.72 |
| TOA shortwave radiation (*111.56* W m$^{-2}$) | Mean_sim | *107.76* | *112.68* | *110.38* | *110.95* | *107.16* | *114.33* | *116.62* | *113.09* |
| | R | **0.81** | 0.79 | 0.69 | 0.68 | 0.62 | 0.65 | 0.65 | 0.65 |
| | MB | -3.80 | 1.13 | -1.18 | -0.61 | -4.40 | 3.12 | 5.42 | 1.89 |
| | NMB (%) | -3.40 | **1.01** | -1.05 | -0.55 | -3.94 | 2.81 | 4.87 | 1.70 |
| | RMSE | **15.75** | 16.04 | 17.07 | 16.10 | 17.21 | 20.85 | 20.67 | 18.96 |
| TOA longwave radiation (*233.68* W m$^{-2}$) | Mean_sim | *231.54* | *232.26* | *234.34* | *233.96* | *234.39* | *232.52* | *232.17* | *233.18* |
| | R | 0.88 | 0.90 | 0.91 | 0.91 | **0.92** | 0.74 | 0.74 | 0.76 |
| | MB | -2.14 | -1.42 | 0.66 | 0.28 | 0.71 | -0.61 | -0.96 | 0.05 |
| | NMB (%) | -0.92 | -0.61 | 0.28 | **0.12** | 0.30 | -0.26 | -0.41 | 0.02 |
| | RMSE | 6.94 | 6.20 | 6.00 | 5.94 | **5.86** | 10.10 | 10.07 | 9.70 |
| Precipitation (*948.91* mm y$^{-1}$) | Mean_sim | *872.42* | *896.98* | *1069.06* | *1056.95* | *1081.84* | *1165.06* | *1160.35* | *1163.77* |
| | R | **0.71** | **0.71** | **0.71** | **0.71** | 0.70 | 0.69 | 0.69 | 0.69 |
| | MB | -76.49 | -51.93 | 120.15 | 108.04 | 132.94 | 207.05 | 202.35 | 205.76 |


| | | | | | | | | |
|---|---|---|---|---|---|---|---|---|
| | NMB (%) | -9.23 | **-8.40** | 12.66 | 11.39 | 14.01 | 21.61 | 21.12 | 21.48 |
| | RMSE | **573.14** | 595.76 | 675.91 | 668.92 | 693.74 | 776.60 | 786.36 | 790.73 |
| Cloud cover (64.09 %) | Mean_sim | *52.51* | *53.32* | *48.18* | *47.80* | *47.46* | *58.12* | *57.98* | *58.55* |
| | R | 0.68 | 0.68 | **0.69** | **0.69** | 0.68 | 0.66 | 0.66 | 0.64 |
| | MB | -11.58 | -10.77 | -16.12 | -16.50 | -16.83 | -6.60 | -6.74 | -6.18 |
| | NMB (%) | -18.07 | -16.80 | -25.07 | -25.66 | -26.18 | -10.20 | -10.41 | **-9.54** |
| | RMSE | 16.47 | 16.28 | 20.17 | 20.48 | 20.73 | 15.28 | 15.33 | 15.34 |
| liquid water path (88.44 g m⁻²) | Mean_sim | *53.50* | *57.15* | *32.29* | *31.87* | *31.08* | *56.23* | *56.21* | *54.00* |
| | R | **0.61** | 0.58 | 0.47 | 0.46 | 0.28 | 0.55 | 0.55 | 0.51 |
| | MB | -34.94 | -31.29 | -56.16 | -56.58 | -57.36 | -32.37 | -32.40 | -34.61 |
| | NMB (%) | -39.51 | **-35.38** | -63.49 | -63.97 | -64.86 | -36.54 | -36.56 | -39.06 |
| | RMSE | 54.35 | 54.31 | 63.54 | 63.92 | 67.21 | **53.39** | 53.42 | 55.86 |


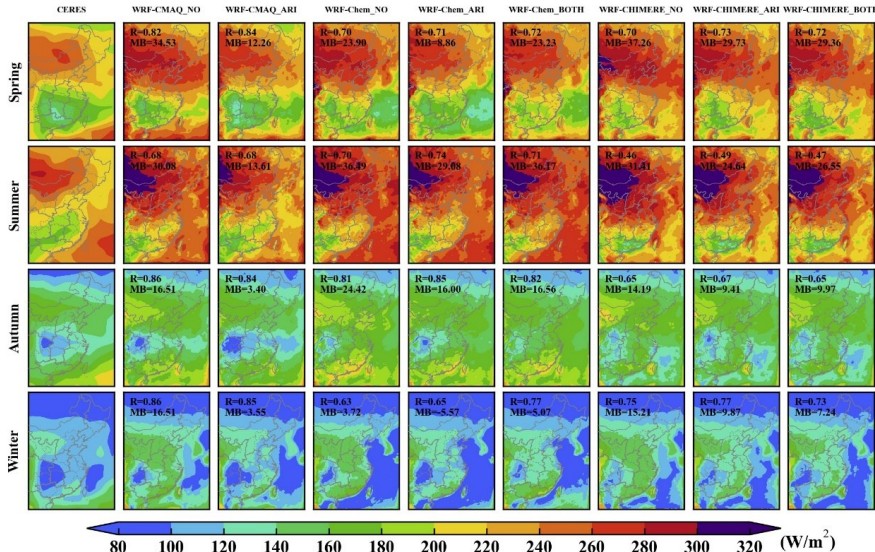


Figure 5. Spatial distributions of seasonal SSR between CERES observations and simulations from WRF-CMAQ, WRF-Chem, and WRF-CHIMERE with and without aerosol feedbacks in eastern China.

## 4 Air quality evaluations and intercomparisons

In a similar way to meteorology, to further determine the quantitative effects of enabling aerosol feedbacks on the simulation accuracy of air quality variables in eastern China, ground-based and satellite-borne observations were adopted as comparisons in the following evaluation analysis. The usage status of computing resources during each simulation process is also assessed in Section 4.3.



### 4.1 Ground-based observations

Table 4 and Figure 6 present the statistical metrics of annual and seasonal air pollutant concentrations ($PM_{2.5}$, $O_3$, $NO_2$, $SO_2$, and CO) simulated by each of the three coupled models. The R values of annual $PM_{2.5}$ concentrations for WRF-CMAQ (0.68) were the highest, followed by WRF-Chem (0.65–0.68), and WRF-CHIMERE (0.52–0.53). All three models showed higher correlations in winter compared with those in other seasons (Figure 7). WRF-CMAQ underestimated annual and seasonal (except for autumn) $PM_{2.5}$ concentrations with NMBs ranging from −9.78% to −6.39% and −17.68% to +5.17%, respectively. WRF-Chem generated both overestimations and underestimations of $PM_{2.5}$ at the annual and seasonal scales, with related NMBs varying from −39.11% to +24.72%. Meanwhile, WRF-CHIMERE excessively overestimated annual and seasonal $PM_{2.5}$ concentrations (NMB: +19.51% to +75.47%). These biases were produced by the configurations of different aerosol and gas phase mechanisms, online dust emission schemes, and chemical ICs and BCs in the two-way coupled models. Based on the differences in NMBs between simulations with ARI and those with no aerosol feedbacks, ARI-induced annual and seasonal NMB variations of WRF-CMAQ_ARI and WRF-Chem_ARI ranged from +3.01% to +4.21% and +3.07% to +5.02%, respectively, indicating that the enabling of ARI feedbacks slightly reduced annual and seasonal (except for autumn) underestimations of $PM_{2.5}$ concentrations. Note that WRF-CHIMERE_ARI further overestimated the annual and seasonal $PM_{2.5}$, with an increase in NMB of up to 10.04%. The increases in $PM_{2.5}$ concentrations caused by ARI effects can be attributed to synergetic decreases in SSR, T2, WS10, and PBLH, and increases in RH2. With ACI feedbacks further enabled, WRF-Chem_BOTH largely underestimated the annual and seasonal $PM_{2.5}$, with NMBs varying from −24.15% to −14.44% compared with WRF-Chem_ARI. WRF-CHIMERE_BOTH tended to decrease (−2.1% to −0.51%) annual and autumn–winter NMBs, and increase (+0.35% to +3.04%) spring–summer NMBs. Further comparison between ARI- and ACI-induced NMB variations demonstrates the key point that ARI-induced variations in $PM_{2.5}$ concentrations were smaller than those induced by ACI in WRF-Chem, but this pattern was reversed in WRF-CHIMERE. This may be explained by WRF-CHIMERE incorporating the process of dust aerosols serving as IN, which was not included in WRF-Chem in this study.

For $O_3$, WRF-CHIMERE (R = 0.62) exhibited the best model performance, followed by WRF-CMAQ (R = 0.55), and WRF-Chem (R = 0.45) (Table 4 and Figure S15). WRF-CMAQ slightly underestimated annual $O_3$, with NMBs of −7.83% to −11.52%, but WRF-Chem and WRF-CHIMERE both significantly overestimated it, with NMBs of 47.82%–48.10% and 29.46%–29.75%, respectively. The seasonal results of statistical metrics showed patterns that were consistent with annual simulations, and summer $O_3$ pollution levels were better simulated than those in other seasons (Figure 6). All models with ARI feedbacks enabled resulted in slight decreases in annual and seasonal $O_3$ NMBs, ranging from −3.02% to +0.85% (the only positive value of +0.85% was produced by WRF-CMAQ in summer). Meanwhile, for ACI effects, WRF-Chem and WRF-CHIMERE had increased annual $O_3$ NMBs of +0.12% and +0.65%, respectively. ACI-induced seasonal NMB variations were different for WRF-Chem



compared with WRF-CHIMERE; WRF-Chem increased in spring–summer and decreased in autumn–winter, while WRF-CHIMERE increased in all seasons except for winter (Figure 6). Such diversity in NMB variation can be explained by configuration differences in gas-phase chemistry mechanisms, which involve various photolytic reactions (a more detailed explanation can be found in Section 4.2).

A comprehensive assessment of the effects of seven gas-phase chemical mechanisms (RADM2, RADMKA, RACM-ESRL, CB05Clx, CB05-TUCL, CBMZ, and MOZART-4) on $O_3$ simulations via three two-way coupled models (WRF-Chem, WRF-CMAQ, and COSMO-ART) was conducted by Knote et al. (2015); they concluded that the $O_3$ concentrations simulated via WRF-Chem with the CBMZ mechanism were closest to the mean values of multiple models over North America and Europe in spring and summer. However, in contrast to North America and Europe, the two-way coupled WRF-Chem with CBMZ had the poorest performance during spring in eastern China. In addition, ARI and/or ACI effects contribute to atmospheric dynamics and stability (as mentioned in the PBLH evaluation part of Section 3.1), as well as photochemistry and heterogeneous reactions, and, in turn, they will eventually influence $O_3$ formation (Xing et al., 2017; Qu et al., 2021; Zhu et al., 2021).

According to the annual statistical results (Table 4 and Figure S16), the $NO_2$ simulated by all three models had comparable correlations (0.50–0.60) with ground-based observations. WRF-CMAQ slightly overestimated $NO_2$ (MBs of +2.74 to +3.26 µg m$^{-3}$, and NMBs of +8.77% to +10.44%), but WRF-Chem (MBs of −10.03 to −9.22 µg m$^{-3}$, and NMBs of −32.14% to −29.55%) and WRF-CHIMERE (MBs of −9.35 to −8.96 µg m$^{-3}$, and NMBs of −29.96% to −28.73%) tended to largely underestimate $NO_2$ in eastern China. For seasonal variations (Figure 6), WRF-CMAQ had the best performance in winter, and generally overestimated $NO_2$ in all seasons (NMBs of −2.21% to 34.34%). Both WRF-Chem and WRF-CHIMERE had maximum R and NMB values (0.42–0.50 and −13.09% to −3.23%, respectively) in winter, and minimum values (0.57–0.62 and −41.57% to −38.05%, respectively) in summer. The annual and seasonal positive biases of WRF-CMAQ are partially caused by not incorporating the heterogeneous reactions of $NO_2$ on ground and aerosol surfaces (Spataro et al., 2013; Li et al., 2018; Liu et al., 2019). These gaps had been filled by Zhang et al. (2021) in CMAQ v5.3 but not incorporated into the official released versions. For WRF-Chem and WRF-CHIMERE, underestimations of $NO_2$ were consistent with overestimations of $O_3$, because $NO_x$ depletions were dominated by $O_3$ titrations. In addition, subtle differences existed in the default settings of reaction rate constants for specific chemical reactions in WRF-CMAQ, WRF-Chem, and WRF-CHIMERE; more detailed information can be found in the source code files of mech_cb6r3_ae6_aq.def, module_cbmz.F, and rates.F, respectively. With ARI feedbacks enabled, the annual and seasonal R values of $NO_2$ simulated by WRF-CMAQ improved, but the NMB got worse; both WRF-Chem and WRF-CHIMERE were improved. Our results show that ARI effects tended to enhance $NO_2$ overestimations in WRF-CMAQ, and mitigate underestimations in WRF-Chem and WRF-CHIMERE. This can be explained by the ARI-induced $NO_2$ reductions being attributed to slower photochemical reactions, and strengthened atmospheric stability and $O_3$ titration, and vice versa. When ACI effects



were further enabled in WRF-Chem and WRF-CHIMERE, the improvements in model
performances were relatively limited.

All models performed most poorly for annual and seasonal $SO_2$ and CO
simulations over eastern China (Table 4 and Figure 6). For $SO_2$, annual correlations
were equivalent for all models, and ranged from 0.39 to 0.41. All three models
underestimated $SO_2$; WRF-CMAQ showed the smallest MB ($-4.31$ µg m$^{-3}$), and WRF-
Chem the largest ($-10.30$ µg m$^{-3}$). Gao et al. (2018) also showed that all two-way
coupled models, except the WRF-Chem version from the University of Iowa modeling
group, tended to underestimate $SO_2$ ($-54.77$ to $4.50$ µg m$^{-3}$) over the North China Plain
during January, 2013. The R values for all models were highest in autumn and winter
(0.31–0.46), and lowest in spring and summer (0.16–0.38), but NMBs showed the
opposite trend. As concluded by Liu et al. (2010), larger underestimations of seasonal
$SO_2$ concentrations were caused by the weaker solar radiation and lesser amount of
precipitation in winter compared with summer, which slowed down the photochemical
conversion of $SO_2$ to $SO_4^{2-}$, wet scavenging, and aqueous-phase oxidation rates of $SO_2$.
For CO (Table 4), WRF-CHIMERE (0.47–0.48) had higher correlation
coefficients than those of WRF-CMAQ (0.23–0.24) and WRF-Chem (0.21–0.22). All
three models underestimated CO concentrations, with MBs ranging from $-0.52$ to
$-0.39$ mg m$^{-3}$. These underestimations were partly caused by uncertainties in the
vertical allocation of CO emissions (He et al., 2017). WRF-CMAQ and WRF-Chem
both produced spring-minimum (0.15) and winter-maximum (0.36) seasonal cycles of
R values (Figure 6), while WRF-CHIMERE had high (0.47) and low (0.26) correlations
in winter and summer, respectively. Negative seasonal NMBs ($-56.94\%$ to $-33.18\%$)
were present in all coupled models. When ARI effects were considered, annual and
seasonal $SO_2$ and CO model performances in all three models were slightly improved
(R increased by approximately 0.01, and NMB increased by 0.98%–1.71%). Moreover,
improvements in the simulation accuracies of $SO_2$ and CO for two-way coupled WRF-
Chem and WRF-CHIMERE were dominated by ARI effects rather than ACI effects.
Table 4. Statistical metrics (R, MB, NMB, and RMSE) between annual simulations and
observations of surface PM$_{2.5}$, O$_3$, NO$_2$, SO$_2$, and CO in eastern China. The best results
are in bold, while mean simulations and observations are in italics.

| Variables | Statistics | WRF-CMAQ_NO | WRF-CMAQ_ARI | WRF-Chem_NO | WRF-Chem_ARI | WRF-Chem_BOTH | WRF-CHIMERE_NO | WRF-CHIMERE_ARI | WRF-CHIMERE_BOTH |
|---|---|---|---|---|---|---|---|---|---|
| PM$_{2.5}$ (*44.99* µg/m$^3$) | Mean_sim | *40.59* | *42.12* | *44.45* | *46.65* | *38.33* | *62.17* | *65.36* | *65.13* |
| | R | **0.68** | **0.68** | 0.65 | 0.65 | 0.69 | 0.52 | 0.53 | 0.53 |
| | MB | -4.40 | -2.87 | -0.54 | 1.66 | -6.66 | 17.18 | 20.37 | 20.14 |
| | NMB (%) | -9.78 | -6.39 | **-1.21** | 3.69 | -14.81 | 38.19 | 45.27 | 44.76 |
| | RMSE | **27.62** | 27.69 | 32.58 | 34.64 | 32.48 | 55.13 | 60.25 | 59.41 |
| O$_3$ (*62.23* µg/m$^3$) | Mean_sim | *55.06* | *54.41* | *88.53* | *87.81* | *87.89* | *76.92* | *76.48* | *76.89* |
| | R | 0.54 | 0.55 | 0.46 | 0.45 | 0.45 | **0.62** | **0.62** | **0.62** |
| | MB | -7.17 | -7.83 | 26.30 | 25.58 | 25.65 | 14.69 | 14.25 | 14.66 |
| | NMB (%) | **-11.52** | -12.57 | 42.26 | 41.10 | 41.22 | 23.60 | 22.90 | 23.55 |
| | RMSE | **28.32** | 28.68 | 48.10 | 47.99 | 47.82 | 29.65 | 29.46 | 29.75 |
| NO$_2$ | Mean_sim | *33.94* | *34.46* | *21.17* | *21.98* | *21.40* | *21.85* | *22.20* | *22.24* |



| | | | | | | | | |
|---|---|---|---|---|---|---|---|---|
| (31.2 µg/m³) | R | 0.59 | **0.60** | 0.50 | 0.50 | 0.50 | 0.55 | 0.56 | 0.56 |
| | MB | 2.74 | 3.26 | -10.03 | -9.22 | -9.80 | -9.35 | -9.00 | -8.96 |
| | NMB (%) | **8.77** | 10.44 | -32.14 | -29.55 | -31.40 | -29.96 | -28.84 | -28.73 |
| | RMSE | **19.14** | 19.48 | 21.23 | 21.21 | 21.21 | 18.72 | 18.68 | 18.70 |
| SO₂ | Mean_sim | *14.02* | *14.39* | *8.22* | *8.56* | *7.85* | *8.88* | *9.18* | *9.19* |
| (18.51 µg/m³) | R | 0.40 | 0.40 | 0.44 | 0.44 | **0.46** | 0.40 | 0.41 | 0.41 |
| | MB | -4.49 | -4.12 | -10.29 | -9.95 | -10.66 | -9.63 | -9.33 | -9.32 |
| | NMB (%) | -24.25 | **-22.24** | -55.61 | -53.76 | -57.57 | -52.02 | -50.39 | -50.34 |
| | RMSE | 21.11 | 21.30 | 20.13 | **20.02** | 20.20 | 22.07 | 22.17 | 22.18 |
| CO | Mean_sim | *0.44* | *0.45* | *0.53* | *0.54* | *0.53* | *0.56* | *0.58* | *0.57* |
| (0.96 mg/m³) | R | 0.23 | 0.24 | 0.21 | 0.22 | 0.22 | 0.47 | **0.48** | 0.47 |
| | MB | -0.52 | -0.51 | -0.43 | -0.42 | -0.43 | -0.40 | -0.39 | -0.39 |
| | NMB (%) | -53.97 | -52.99 | -45.10 | -43.94 | -44.68 | -41.82 | **-40.11** | -40.28 |
| | RMSE | 0.90 | 0.90 | 0.82 | 0.83 | 0.83 | **0.62** | **0.62** | **0.62** |


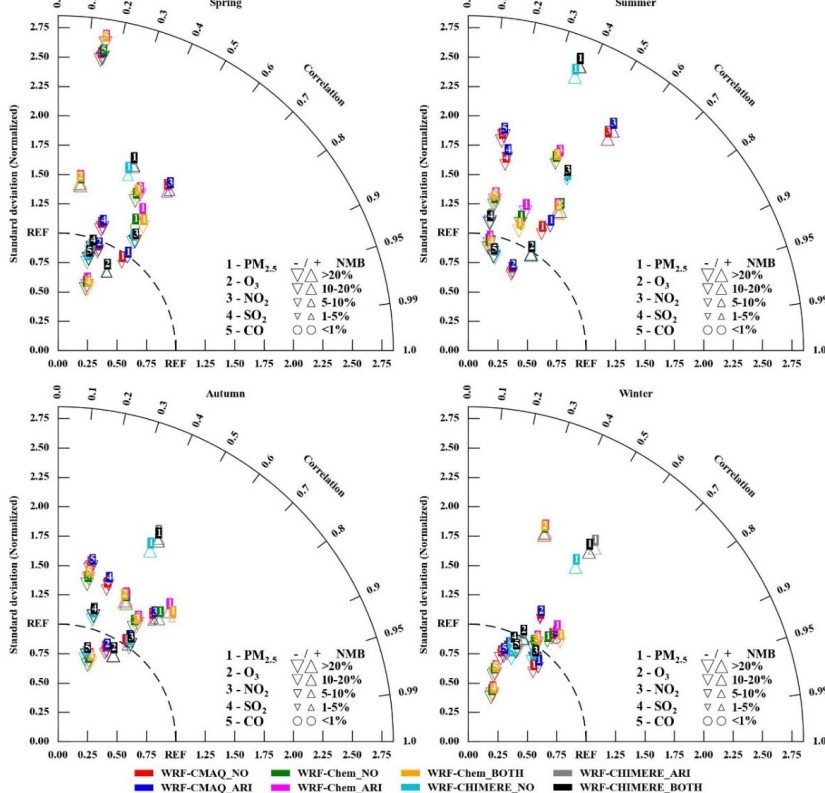


Figure 6. Taylor diagrams (R, normalized standard deviation, and NMB) of seasonal
PM₂.₅, O₃, NO₂, SO₂, and CO via three two-way coupled models (WRF-CMAQ, WRF-
Chem, and WRF-CHIMERE) with/without ARI and/or ACI effects in eastern China
compared with surface observations.





In a similar manner to the meteorological variables presented above, we aimed to
conduct quality assurance for the statistical metrics by making further comparisons with
$PM_{2.5}$ and $O_3$ results from previous model evaluations (summarized in Figure 7). The
performances of WRF-CMAQ and WRF-Chem in simulating $PM_{2.5}$ in this study were
better than the average levels of previous studies from eastern China. For $O_3$, WRF-
Chem simulations performed worse than the average level of previous studies.
Although the R values of $O_3$ simulated by WRF-CMAQ in this study were lower than
the average level of previous studies, the RMSEs in this study were smaller.

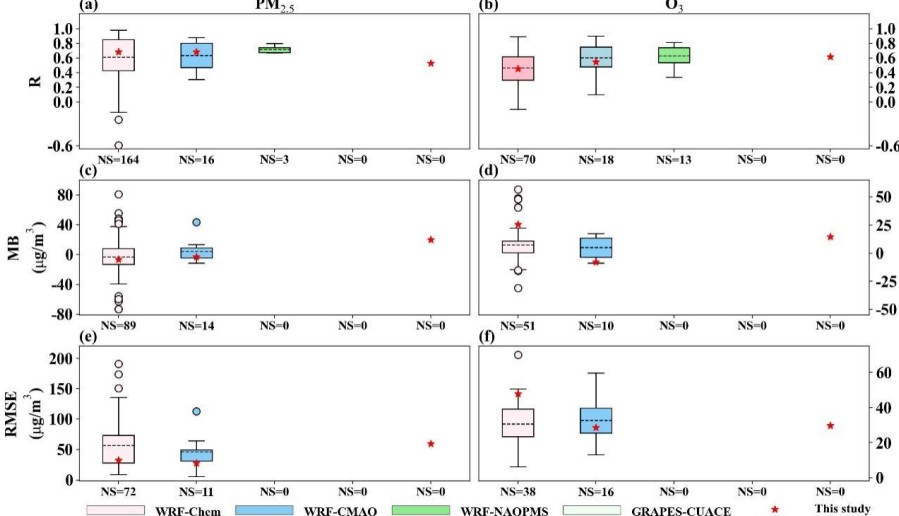

Figure 7. Comparisons of model capacities between our study (red stars) and previous literature
(box plots) in terms of surface $PM_{2.5}$ and $O_3$ concentrations in eastern China. Note that red stars in
the fifth column of each subgraph represent the statistical metrics of WRF-CHIMERE in this study.

4.2 Satellite-borne observations
In this section, we further investigate the discrepancies among different models in
terms of the calculated AOD and column concentrations of gases ($O_3$, $NO_2$, $SO_2$, CO,
and $NH_3$), and compare them with various satellite observations. For $NH_3$, owing to not
setting the output of simulated $NH_3$ concentrations in WRF-CHIMERE, the discussion
here only includes the results from WRF-CMAQ and WRF-Chem.
As shown in Table 5, annual AOD at 550 nm, TCO, $NO_2$, and CO simulated by all
three models agreed most closely with satellite observations, with correlation
coefficients of 0.80–0.98; these were followed by $NH_3$ (0.75–0.76), and $SO_2$ (0.50–
0.53). WRF-CMAQ presented negative biases for annual AOD (−0.01), TCO (−5.92
Dobson Units (DU)), $SO_2$ (−0.03 to −0.02 DU), CO (−1.25 × $10^{17}$ molecules cm$^{-2}$), and
$NH_3$ (−2.95 × $10^{15}$ molecules cm$^{-2}$), but a positive bias for $NO_2$ (1.09–1.21
petamolecules cm$^{-2}$). For AOD, WRF-Chem and WRF-CHIMERE produced positive
and negative MBs of +0.09 and −0.06, respectively. Both WRF-Chem and WRF-
CHIMERE overestimated $NO_2$ (0.28–0.63 petamolecules cm$^{-2}$) and CO (0.93–1.21 ×
$10^{17}$ molecules cm$^{-2}$), and underestimated $O_3$ (-10.99 to -3.63 DU) and $SO_2$ (-0.03 to -



0.02 DU). Similar to WRF-CMAQ, WRF-Chem also underestimated NH$_3$ by
approximately $-3.14 \times 10^{15}$ molecules cm$^{-2}$.
For seasonal variations, relatively high correlation relationships (0.71–0.88) of
AOD were present in autumn, with lower values (0.53–0.84) in other seasons (Figure
9). WRF-CMAQ and WRF-Chem tended to underestimate AOD in summer (MBs of
$-0.1$ to $-0.4$) and overestimate it in other seasons (MBs of 0.01–0.05). WRF-
CHIMERE had positive biases (0.03–0.04) in winter and negative biases ($-0.10$ to
$-0.01$) in other seasons.
For TCO (Figure S19), the model performances of WRF-CMAQ and WRF-Chem
in spring and winter were slightly better than those in summer and autumn, but all
seasonal R values were greater than 0.89. Both WRF-CMAQ ($-9.53$ to $-0.72$ DU) and
WRF-Chem ($-24.62$ to $+10.57$ DU) had negative biases in all seasons (note: WRF-
Chem except for autumn). WRF-CHIMERE was better at capturing TCO in spring and
summer (overestimations of $+9.19$ to $+29.20$ DU) than in autumn and winter
(underestimations of $-33.75$ to $-19.40$ DU).
The R values of NO$_2$ columns for all three models were slightly higher in autumn
and winter (0.82–0.91) than in spring and summer (0.76–0.84). The simulation
accuracies of NO$_2$ columns via WRF-CHIMERE were significantly better than those
using WRF-CMAQ or WRF-Chem in all seasons except for winter (Figure S20). All
models overestimated SO$_2$ column concentrations in winter (by approximately 0.01–
0.03 DU) but underestimated them in other seasons ($-0.05$ to $-0.001$ DU) (Figure S21).
For NH$_3$, the only primary alkaline gas in the atmosphere, better model
performances of WRF-CMAQ and WRF-Chem occurred in summer (R: 0.81–0.87; MB:
$-3.42$ to $2.07 \times 10^{15}$ molecules cm$^{-2}$) (Figure S22). Ammonia emissions from fertilizer
and livestock have been substantially underestimated in China (Zhang et al., 2017), and
peak values occur in spring and summer (Huang et al., 2012). In addition, bidirectional
exchanges of fertilizer-induced NH$_3$ were not considered in our simulations.
WRF-CMAQ, WRF-Chem, and WRF-CHIMERE showed relatively poor
performances (R: 0.68–0.79) in simulating CO columns during spring, summer, and
autumn, respectively, compared with other seasons (Figure S23). WRF-CMAQ and
WRF-CHIMERE respectively underestimated and overestimated CO columns in other
seasons except for summer and spring, with MBs of $-3.29$ to $0.31 \times 10^{17}$ and $-0.62$ to
$2.09 \times 10^{17}$ molecules cm$^{-2}$, respectively. WRF-Chem had positive MBs in summer and
autumn ($4.03$–$5.12 \times 10^{17}$ molecules cm$^{-2}$) and negative MBs in spring and winter
($-3.15$ to $-2.10 \times 10^{17}$ molecules cm$^{-2}$).
Moreover, after comparing the performance results for each pollutant between
sections 4.1 and 4.2, the only disparity found between evaluations with ground-based
observations compared with those with satellite-borne observations was for CO. The
formation of CO via the oxidation of methane, an important source of CO emissions
(Stein et al., 2014), is not considered in the three coupled models, and methane
emissions are not included in the MEIC inventory. In addition, the contribution of CO
to atmospheric oxidation capacity (OH radicals) was non-negligible (e.g., values were
approximately 20.54%–38.97% in Beijing (Liu et al., 2021), and 26%–31% in Shanghai
(Zhu et al., 2020).





These discrepancies in the model performances for simulating AOD and column
concentrations of gases can be explained by differences in the representations of aerosol
species groups, Fast-JX photolysis mechanism, and gas-phase mechanisms in the three
coupled models.
When all three models enabled just ARI effects, improvements in annual AOD and
$NO_2$ columns simulated by these models were relatively limited. The AOD simulations
improved in spring and summer, but worsened in autumn and winter (Table 4 and Figure
9). Larger variations in seasonal MBs of $NO_2$ columns induced by ARI effects occurred
in WRF-CMAQ ($-0.18$ to $0.13$ petamolecules cm$^{-2}$) compared with WRF-Chem and
WRF-CHIMERE ($0$–$0.01$ petamolecules cm$^{-2}$). When both ARI and ACI effects were
enabled in WRF-Chem, the model performance for seasonal AOD simulations
worsened considerably. The annual and seasonal $NO_2$ simulations via WRF-Chem
became slightly worse, while those using WRF-CHIMERE became slightly better. In
contrast to AOD and $NO_2$ column concentrations, improvements in annual and seasonal
column simulations of total ozone, PBL $SO_2$, and $NH_3$ via all two-way coupled models
were limited when one or both of ARI and ACI were enabled.
Table 5. Statistical metrics (R, MB, NMB, and RMSE) of simulated and satellite-
retrieved AOD, total column ozone, tropospheric column $NO_2$, PBL column $SO_2$, total
column CO, and total column density of $NH_3$ in eastern China. The best results are in
bold, while annual mean simulations and observations are in italics.

| Variables | Statistics | WRF-CMAQ_NO | WRF-CMAQ_ARI | WRF-Chem_NO | WRF-Chem_ARI | WRF-Chem_BOTH | WRF-CHIMERE_NO | WRF-CHIMERE_ARI | WRF-CHIMERE_BOTH |
|---|---|---|---|---|---|---|---|---|---|
| AOD (*0.27*) | Mean_sim | *0.26* | *0.27* | *0.35* | *0.36* | *0.25* | *0.21* | *0.22* | *0.22* |
| | R | 0.80 | 0.80 | 0.80 | 0.80 | 0.75 | **0.87** | **0.87** | 0.86 |
| | MB | -0.01 | -0.01 | 0.09 | 0.09 | -0.01 | -0.05 | -0.05 | -0.04 |
| | NMB (%) | -3.99 | **-2.93** | 34.14 | 35.03 | -4.92 | -18.72 | -17.37 | -16.22 |
| | RMSE | **0.09** | **0.09** | 0.15 | 0.15 | 0.10 | **0.09** | **0.09** | 0.10 |
| $O_3$ VCDs (*312.07* DU) | Mean_sim | *306.15* | *306.15* | *300.77* | *300.73* | *300.46* | *307.69* | *307.47* | *307.75* |
| | R | **0.98** | **0.98** | 0.97 | 0.97 | 0.97 | 0.65 | 0.65 | 0.65 |
| | MB | -5.92 | -5.92 | -10.68 | -10.72 | -10.99 | -3.69 | -3.91 | -3.63 |
| | NMB (%) | -1.90 | -1.90 | -3.43 | -3.44 | -3.53 | -1.19 | -1.26 | **-1.17** |
| | RMSE | **8.91** | **8.91** | 83.72 | 83.73 | 83.94 | 39.88 | 39.71 | 39.73 |
| Tropospheric $NO_2$ VCDs (*$2.71 \times 10^{15}$ molecules cm$^{-2}$*) | Mean_sim | *3.80* | *3.91* | *3.07* | *3.08* | *3.06* | *2.62* | *2.63* | *2.63* |
| | R | 0.85 | 0.85 | **0.87** | **0.87** | **0.87** | **0.87** | **0.87** | **0.87** |
| | MB | 1.09 | 1.21 | 0.62 | 0.63 | 0.61 | 0.28 | 0.29 | 0.29 |
| | NMB (%) | 40.35 | 44.64 | 25.27 | 25.52 | 24.89 | **12.03** | 12.47 | 12.42 |
| | RMSE | 3.18 | 3.33 | 2.27 | 2.27 | 2.27 | **1.65** | 1.67 | 1.68 |
| PBL $SO_2$ VCDs (*0.09* DU) | Mean_sim | *0.07* | *0.07* | *0.09* | *0.09* | *0.06* | *0.06* | *0.06* | *0.06* |
| | R | 0.53 | 0.53 | **0.56** | **0.56** | 0.54 | 0.50 | 0.50 | 0.50 |
| | MB | -0.03 | -0.02 | -0.03 | -0.02 | -0.03 | -0.03 | -0.02 | -0.02 |
| | NMB (%) | -27.32 | -25.48 | -32.50 | **-21.50** | -35.08 | -28.64 | -27.31 | -27.51 |
| | RMSE | **0.07** | **0.07** | 0.08 | 0.08 | **0.07** | **0.07** | **0.07** | **0.07** |
| Total CO VCDs | Mean_sim | *20.34* | *20.35* | *22.20* | *22.20* | *22.21* | *22.34* | *22.36* | *22.35* |
| | R | 0.83 | 0.83 | **0.87** | **0.87** | **0.87** | 0.86 | 0.86 | 0.86 |





|  |  |  |  |  |  |  |  |  |  |
|---|---|---|---|---|---|---|---|---|---|
| (21.60×10¹⁷ molecules cm⁻²) | MB | -1.26 | -1.24 | 0.93 | 0.93 | 0.94 | 1.19 | 1.21 | 1.19 |
|  | NMB (%) | -5.83 | -5.75 | **4.35** | 4.37 | 4.44 | 5.64 | 5.70 | 5.65 |
|  | RMSE | **2.54** | **2.54** | 2.69 | 2.68 | 2.69 | 2.57 | 2.58 | 2.58 |
| Total NH₃ VCDs (16.05×10¹⁵ molecules cm⁻²) | Mean_sim | *13.06* | *13.15* | *12.31* | *12.27* | *8.63* | NA | NA | NA |
|  | R | **0.76** | **0.76** | 0.73 | 0.73 | 0.76 | NA | NA | NA |
|  | MB | -3.00 | -2.90 | -3.27 | -3.32 | -3.34 | NA | NA | NA |
|  | NMB (%) | -18.66 | **-18.08** | -21.01 | -21.28 | -21.41 | NA | NA | NA |
|  | RMSE | **9.26** | 9.47 | 9.48 | 9.46 | 9.61 | NA | NA | NA |

NA indicates that outputs of NH₃ column concentrations were not extracted from WRF-CHIMERE
with/without aerosol feedback simulations.

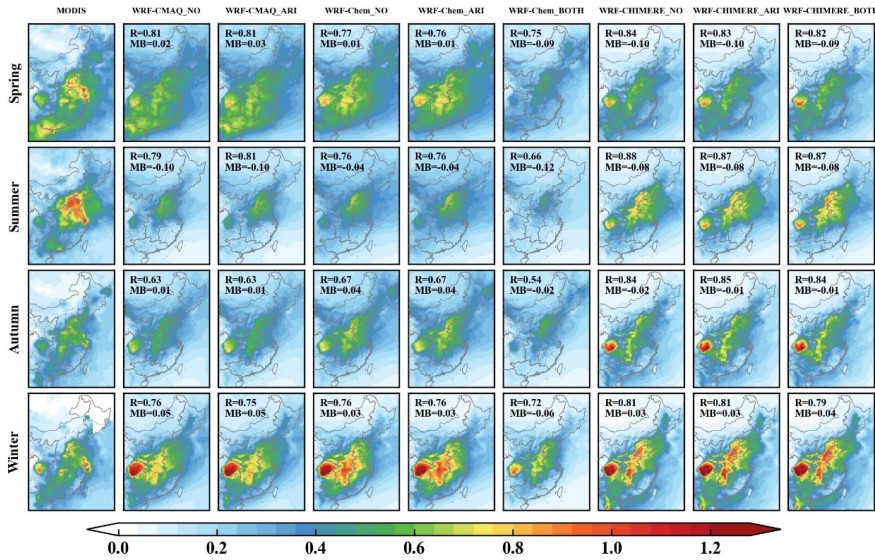

Figure 8. Spatial distributions of seasonal AOD between MODIS observations and
simulations from WRF-CMAQ, WRF-Chem, and WRF-CHIMERE with and without
aerosol feedbacks in eastern China.
**4.3 Computational performance**
Table 5 summarizes the comparative results of central processing unit (CPU) time
consumption for one day simulations via WRF-CMAQ, WRF-Chem, and WRF-
CHIMERE with and without aerosol feedbacks in 2017. The results show that
regardless of whether aerosol feedbacks were enabled, the CPU time consumed by
WRF-CMAQ simulating one-day meteorology and air quality was shortest, followed
by WRF-CHIMERE, and WRF-Chem. Compared with simulations without aerosol
feedbacks, the processing time of WRF-CMAQ with ARI enabled increased by 0.22–
0.34 hours per day, while increases in the running time of WRF-Chem and WRF-
CHIMERE were not significant (0.02–0.03 hours per day). The CPU time for both
WRF-Chem and WRF-CHIMERE with both ARI and ACI effects enabled was slightly



increased, and the increase in CPU time for the former (0.25 hours per day) was larger
than that for the latter (0.11 hours per day). Compared with WRF-CMAQ and WRF-
Chem, the CPU time of WRF-CHIMERE showed obvious seasonal differences, with
the time in winter and spring being significantly longer than that in summer and autumn.
These differences can be partially explained by the choice of main configurations,
including model resolution, model version, and parametrization schemes (cloud
microphysics, PBL, cumulus, surface layer, land surface, gas-phase chemistry, and
aerosol mechanisms).

Table 5. Summary of running time for different coupled models.

| Month | WRF-CMAQ (hour) | | WRF-Chem (hour) | | | WRF-CHIMERE (hour) | | |
|---|---|---|---|---|---|---|---|---|
| | NO | ARI | NO | ARI | BOTH | NO | ARI | BOTH |
| Jan. | 0.37 | 0.59 | 0.69 | 0.71 | 0.96 | 0.67 | 0.70 | 0.77 |
| Feb. | 0.35 | 0.60 | 0.68 | 0.70 | 0.93 | 0.64 | 0.67 | 0.73 |
| Mar. | 0.39 | 0.65 | 0.70 | 0.72 | 1.00 | 0.59 | 0.62 | 0.72 |
| Apr. | 0.37 | 0.67 | 0.67 | 0.69 | 0.92 | 0.54 | 0.57 | 0.65 |
| May | 0.39 | 0.71 | 0.61 | 0.66 | 0.86 | 0.52 | 0.55 | 0.62 |
| June | 0.40 | 0.74 | 0.66 | 0.67 | 0.95 | 0.48 | 0.51 | 0.63 |
| July | 0.36 | 0.69 | 0.65 | 0.67 | 0.86 | 0.49 | 0.50 | 0.58 |
| Aug. | 0.38 | 0.68 | 0.66 | 0.68 | 0.90 | 0.49 | 0.52 | 0.61 |
| Sept. | 0.37 | 0.63 | 0.64 | 0.65 | 0.89 | 0.48 | 0.52 | 0.63 |
| Oct. | 0.38 | 0.62 | 0.66 | 0.68 | 0.94 | 0.53 | 0.56 | 0.69 |
| Nov. | 0.36 | 0.58 | 0.68 | 0.70 | 0.91 | 0.64 | 0.67 | 0.72 |
| Dec. | 0.35 | 0.57 | 0.63 | 0.66 | 0.87 | 0.67 | 0.70 | 0.74 |


5 Conclusions

In this study, we comprehensively evaluated the annual hindcast simulations for
2017 by the two-way coupled WRF-CMAQ, WRF-Chem, and WRF-CHIMERE
models with/without aerosol feedbacks and explored the impacts of ARI and/or ACI on
model and computational performances in eastern China. All three two-way coupled
models effectively reproduced the spatiotemporal distributions of meteorology and air
quality, but some variables (SSR and $PM_{2.5}$) in specific regions showed significant
discrepancies. Among meteorological variables at the annual scale, T2 and Q2 were
better simulated by the three models than SSR, RH2, WS10, PBLH, and PREP. The
SSR, RH2, and WS10 were overestimated with MBs around 15.91–42.65 W m$^{-2}$, 2.53–
3.55% and 0.42–1.04 m s$^{-1}$, respectively, while T2 and Q2 were underestimated with
MBs ranged from -0.57 to -0.18 g kg$^{-1}$ and -2.00 to 0.68 ℃, respectively. For PREP, the
WRF-CMAQ's underestimation was 0.5 mm day$^{-1}$, but WRF-Chem and WRF-
CHIMERE overestimated PREP about 1 mm day$^{-1}$. The seasonal variations of
simulated meteorological variables in eastern China were also well matched with
observations. Overall, the MBs of every meteorological variable simulated by the three
models in spring and winter were significantly smaller than those in summer and
autumn. In terms of air quality, all three models presented generally acceptable





performance for annual surface $PM_{2.5}$, $O_3$, and $NO_2$ concentrations, but not for $SO_2$ and
CO. The overall performances of WRF-CMAQ were best, followed by WRF-Chem,
and WRF-CHIMERE. The WRF-CMAQ and WRF-Chem simulations had positive
biases for $NO_2$ (2.74–3.26 $\mu g\,m^{-3}$) and $O_3$ (25.58–26.30 $\mu g\,m^{-3}$), but negative biases for
other pollutants, while WRF-CHIMERE simulations had positive biases for $PM_{2.5}$
(17.18–20.37 $\mu g\,m^{-3}$) and $O_3$ (14.25–14.69 $\mu g\,m^{-3}$). The seasonal simulations of surface
air quality variables showed better correlations of $PM_{2.5}$, $NO_2$, $SO_2$, and CO in winter,
and $O_3$ in summer than those in other seasons. Further compared with satellite
observations, all coupled models well captured radiation, precipitation, cloud fraction,
AOD, and column concentrations of $O_3$, $NO_2$, CO, and $NH_3$ both at annual and seasonal
scales, but not for LWP and $SO_2$ concentrations.
Our evaluations showed that the effects of aerosol feedbacks on model
performances varied depending on the two-way coupled models,variables, and time
scales. In general, all three two-way coupled models enabling ARI improved the
simulation accuracy of annual and seasonal SSR. However, simulation accuracy of SSR
was reduced in WRF-Chem and WRF-CHIMERE with only considering ACI, with
slightly improved results after enabling both ARI and ACI. Aerosol feedbacks induced
various changes of MB for different variables. For example, MBs decreased for SSR
from -19.98 $W\,m^{-2}$ to -9.24 $W\,m^{-2}$, T2 from -0.20 °C to -0.15 °C, Q2 from -0.17 $g\,kg^{-1}$
to -0.02 $g\,kg^{-1}$, WS10 from -0.03 $m\,s^{-1}$ to -0.01 $m\,s^{-1}$ and PBLH from -25.25 m to -1.93
m. MBs increased for $PM_{2.5}$ from 1.53 to 3.19 $\mu g\,m^{-3}$ and other gaseous pollutants ($NO_2$,
$SO_2$ and CO) as well. In addition, there were computational costs (around 20%–70%
increase) involved with turning on aerosol-radiation-cloud effects in two-way coupled
models.
Although many progresses in the developments and enhancements of two-way
coupled models have been made and these models are widely applied worldwide,
several limitations still exist. As comparison studies of offline models' performances
affected by various chemical mechanisms were conducted (Kim et al., 2011; Balzarini
et al., 2015; Zheng et al., 2015), relevant assessments targeting two-way coupled
models are still lacking. Recently, Wu et al. (2018) and Womack et al. (2021)
demonstrated that the non-spherical morphology of BC particles could significantly
enhance light absorption and the spherical core–shell mixing assumptions used in the
most applied coupled models (WRF-CMAQ, WRF-Chem, and WRF-CHIMERE) may
overestimate the ARI effects of BC aerosols. Therefore, numerical representations of
non-spherical aerosol optical properties need to be implemented in two-way coupled
models to reduce uncertainties in the ARI calculations. Previous observational and
modeling studies revealed that there are still large uncertainties in the impacts of ACI
on cloud and precipitation (Seinfeld et al., 2016; IPCC, 2021; Gao et al., 2022), and
more researchers have focused on these gaps and gained some remarkable
developments on aerosol water uptake and in-/below-cloud scavenging in recent years
(Xu et al., 2019; Brüggemann et al., 2020; Kärcher and Marcolli, 2021; Cantrell et al.,
2022; Ryu and Min, 2022; Hogrefe et al., 2023). The latest observational investigations
for coefficient modifications and newly developed parameterizations/schemes need to
be incorporated into the two-way coupled models, such as WRF-CMAQ, WRF-Chem



and WRF-CHIMERE, and further reassessments of uncertainties of the ACI effects in
these models should be carried out in the future.

Code availability
The source codes of the two-way coupled WRF v4.1.1-CMAQ v5.3.1, WRF-
Chem v4.1.1, and WRF v3.7.1-CHIMERE v2020r1 models are obtained from
https://github.com/USEPA/CMAQ, https://github.com/wrf-model/WRF, and
https://www.lmd.polytechnique.fr/chimere, respectively (last access: November 2020).

Data availability
The model inputs and outputs in this study for WRF-CMAQ, WRF-Chem and
WRF-CHIMERE with/without enabling ARI or/and ACI effects are available upon
request. All simulation and observational data of ground-based/satellite-retrieved
meteorological and air quality for computing statistical metrics are available from
https://zenodo.org/record/7750907 (last access: 20 March, 2023).

Author contributions
CG, ZX, AX performed the majority of the source code configuration of WRF-
CMAQ, WRF-Chem and WRF-CHIMERE, designed the numerical simulations to
carry them out, related analysis, figure plotting, and paper writing. QT, HZ, SZ, GY,
MZ and XS were involved with the original research plan and made suggestions for the
paper writing.

Competing interests
The contact author has declared that neither they nor their co-authors have any
competing interests.

Acknowledgements
The authors are very grateful to David Wong, Chun Zhao and Laurent Menut who
provided detailed information on the two-way coupled WRF-CMAQ, WRF-Chem and
WRF-CHIMERE models, respectively.

Financial support
This study was financially sponsored by the Youth Innovation Promotion
Association of Chinese Academy of Sciences, China (grant nos. 2022230), the National
Key Research and Development Program of China (grant nos. 2017YFC0212304 &
2019YFE0194500), the Talent Program of Chinese Academy of Sciences
(Y8H1021001), and the National Natural Science Foundation of China (grant nos.
42171142 & 41771071).





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
