# Peer review of "Inter-comparison of multiple two-way coupled meteorology and air quality models (WRF"

_Geoscientific Model Development, 2023_

## Author Comment (AC1)

Dear Dr. **Juan A. Añel**

We really appreciate your valuable suggestion on our manuscript entitled "***Intercomparison of multiple two-way coupled meteorology and air quality models (WRF v4.1.1-CMAQ v5.3.1, WRF-Chem v4.1.1 and WRF v3.7.1-CHIMERE v2020r1) in eastern China***" (**Manuscript ID: GMD-2023-21**). To comply with the GMD's policy, we have uploaded the source codes of WRF-CMAQ, WRF-Chem, WRF-CHIMERE used in our simulations at https://doi.org/10.5281/zenodo.7901682 (Gao et al., 2023a; link: https://zenodo.org/record/7901682). The meteorological initial and boundary conditions (ICs and BCs) can be obtained at https://doi.org/10.5281/zenodo.7925012 (Gao et al., 2023b; link: https://zenodo.org/record/7925012). The chemical ICs and BCs used for WRF-CMAQ, WRF-Chem and WRF-CHIMERE are available at https://doi.org/10.5281/zenodo.7932390 (Gao et al., 2023; link: https://zenodo.org/record/7932390), https://doi.org/10.5281/zenodo.7932936 (Gao et al., 2023d; link: https://zenodo.org/record/7932936), and https://doi.org/10.5281/zenodo.7933641 (Gao et al., 2023e; link: https://zenodo.org/record/7933641), respectively. The emission data used for WRF-CMAQ, WRF-Chem and WRF-CHIMERE can be downloaded from https://doi.org/10.5281/zenodo.7932430 (Gao et al., 2023f; link: https://zenodo.org/record/7932430), https://doi.org/10.5281/zenodo.7932734 (Gao et al., 2023g; link: https://zenodo.org/record/zenodo.7932734), and https://doi.org/10.5281/zenodo.7931614 (Gao et al., 2023h; link: https://zenodo.org/record/zenodo.7931614), respectively. Please note that due to the size of output data files and the size limit (**50 GB**) for data files on Zenodo, an output file of annual simulation from each model under each scenario needs to be divided into 3-4 parts. The DOIs and links for all the outputs uploaded on Zenodo are added in the revised reference list as shown below.

We revised the manuscript accordingly and all the relevant changes (in blue and italic) made in the revised manuscript are as follows:

1. The description of code availability has been revised to the revised manuscript:

"*Code availability*

*The original source codes of two-way coupled WRF v4.1.1-CMAQ v5.3.1, WRF-Chem v4.1.1, and WRF v3.7.1-CHIMERE v2020r1 models are obtained from*

*https://github.com/USEPA/CMAQ, https://github.com/wrf-model/WRF, and https://www.lmd.polytechnique.fr/chimere, respectively (last access: November 2020). The related source codes, configuration information, namelist files and automated run scripts of these three two-way coupled models are archived at Zenodo with the associated DOI: https://doi.org/10.5281/zenodo.7901682 (Gao et al., 2023a; link: https://zenodo.org/record/7901682).".*

2. The description of data availability has been revised to the revised manuscript:

*"Data availability*

*The meteorological ICs and BCs used for three coupled models can be obtained at https://doi.org/10.5281/zenodo.7925012 (Gao et al., 2023b; link: https://zenodo.org/record/7925012). The Chemical ICs and BCs used for WRF-CMAQ, WRF-Chem and WRF-CHIMERE are available at https://doi.org/10.5281/zenodo.7932390 (Gao et al., 2023c; link: https://zenodo.org/record/7932390), https://doi.org/10.5281/zenodo.7932936 (Gao et al., 2023d; link: https://zenodo.org/record/7932936), and https://doi.org/10.5281/zenodo.7933641 (Gao et al., 2023e; link: https://zenodo.org/record/7933641), respectively. The emission data used for WRF-CMAQ, WRF-Chem and WRF-CHIMERE can be downloaded from https://doi.org/10.5281/zenodo.7932430 (Gao et al., 2023f; link: https://zenodo.org/record/7932430), https://doi.org/10.5281/zenodo.7932734 (Gao et al., 2023g; link: https://zenodo.org/record/7932734), and https://doi.org/10.5281/zenodo.7931614 (Gao et al., 2023h; link: https://zenodo.org/record/7931614), respectively. The DOIs and links regarding the output data of each simulation scenario are presented in Table 6. All data used to create figures and tables in this study are provided in an open repository on Zenodo (https://doi.org/10.5281/zenodo.7750907, Gao et al., 2023q; link: https://zenodo.org/record/7750907).".*

*Table 6 Summary of download information on model output of each simulation scenario*

| Scenario | DOI | Link | Reference |
|---|---|---|---|
| WRF-CMAQ_NO | https://doi.org/10.5281/zenodo.7951404 | https://zenodo.org/record/7951404 | Gao et al., 2023i_part1 |
| | https://doi.org/10.5281/zenodo.7951467 | https://zenodo.org/record/7951467 | Gao et al., 2023i_part2 |
| | https://doi.org/10.5281/zenodo.7951475 | https://zenodo.org/record/7951475 | Gao et al., 2023i_part3 |

| | | | |
|---|---|---|---|
| *WRF-CMAQ_ARI* | *https://doi.org/10.5281/zenodo.7949895* | *https://zenodo.org/record/7949895* | *Gao et al., 2023j_part1* |
| | *https://doi.org/10.5281/zenodo.7950644* | *https://zenodo.org/record/7950644* | *Gao et al., 2023j_part2* |
| | *https://doi.org/10.5281/zenodo.7950830* | *https://zenodo.org/record/7950830* | *Gao et al., 2023j_part3* |
| *WRF-Chem_NO* | *https://doi.org/10.5281/zenodo.7943804* | *https://zenodo.org/record/7943804* | *Gao et al., 2023k_part1* |
| | *https://doi.org/10.5281/zenodo.7945383* | *https://zenodo.org/record/7945383* | *Gao et al., 2023k_part2* |
| | *https://doi.org/10.5281/zenodo.7946944* | *https://zenodo.org/record/7946944* | *Gao et al., 2023k_part3* |
| | *https://doi.org/10.5281/zenodo.7947169* | *https://zenodo.org/record/7947169* | *Gao et al., 2023k_part4* |
| *WRF-Chem_ARI* | *https://doi.org/10.5281/zenodo.7947050* | *https://zenodo.org/record/7947050* | *Gao et al., 2023l_part1* |
| | *https://doi.org/10.5281/zenodo.7948216* | *https://zenodo.org/record/7948216* | *Gao et al., 2023l_part2* |
| | *https://doi.org/10.5281/zenodo.7949410* | *https://zenodo.org/record/7949410* | *Gao et al., 2023l_part3* |
| | *https://doi.org/10.5281/zenodo.7949561* | *https://zenodo.org/record/7949561* | *Gao et al., 2023l_part4* |
| *WRF-Chem_BOTH* | *https://doi.org/10.5281/zenodo.7939221* | *https://zenodo.org/record/7939221* | *Gao et al. 2023m_part1* |
| | *https://doi.org/10.5281/zenodo.7943002* | *https://zenodo.org/record/7943002* | *Gao et al. 2023m_part2* |
| | *https://doi.org/10.5281/zenodo.7943079* | *https://zenodo.org/record/7943079* | *Gao et al. 2023m_part3* |
| | *https://doi.org/10.5281/zenodo.7943323* | *https://zenodo.org/record/7943323* | *Gao et al. 2023m_part4* |
| *WRF-CHIMERE_NO* | *https://doi.org/10.5281/zenodo.7951775* | *https://zenodo.org/record/7951775* | *Gao et al. 2023n_part1* |
| | *https://doi.org/10.5281/zenodo.7951779* | *https://zenodo.org/record/7951779* | *Gao et al. 2023n_part2* |
| | *https://doi.org/10.5281/zenodo.7951791* | *https://zenodo.org/record/7951791* | *Gao et al. 2023n_part3* |
| | *https://doi.org/10.5281/zenodo.7951793* | *https://zenodo.org/record/7951793* | *Gao et al. 2023n_part4* |
| *WRF-CHIMERE_ARI* | *https://doi.org/10.5281/zenodo.7952838* | *https://zenodo.org/record/7952838* | *Gao et al. 2023o_part1* |
| | *https://doi.org/10.5281/zenodo.7952840* | *https://zenodo.org/record/7952840* | *Gao et al. 2023o_part2* |
| | *https://doi.org/10.5281/zenodo.7952842* | *https://zenodo.org/record/7952842* | *Gao et al. 2023o_part3* |
| | *https://doi.org/10.5281/zenodo.7952844* | *https://zenodo.org/record/7952844* | *Gao et al. 2023o_part4* |
| *WRF-CHIMERE_BOTH* | *https://doi.org/10.5281/zenodo.7952859* | *https://zenodo.org/record/7952859* | *Gao et al. 2023p_part1* |
| | *https://doi.org/10.5281/zenodo.7952863* | *https://zenodo.org/record/7952863* | *Gao et al. 2023p_part2* |
| | *https://doi.org/10.5281/zenodo.7952865* | *https://zenodo.org/record/7952865* | *Gao et al. 2023p_part3* |
| | *https://doi.org/10.5281/zenodo.7952867* | *https://zenodo.org/record/7952867* | *Gao et al. 2023p_part4* |

*Reference*

*Chao Gao, Xuelei Zhang, Aijun Xiu, Qingqing Tong, Hongmei Zhao, Guangyi Yang, Mengduo Zhang, Shengjin Xie: Source codes of WRF v4.1.1-CMAQ v5.3.1, WRF-Chem v4.1.1 and WRF v3.7.1-CHIMERE v2020r1, Zenodo [software]. https://doi.org/10.5281/zenodo.7901682, 2023a.*

*Chao Gao, Xuelei Zhang, Aijun Xiu, Qingqing Tong, Hongmei Zhao, Shichun Zhang, Guangyi Yang, Mengduo Zhang, Shengjin Xie: FNL data used for producing meteorological ICs/BCs of WRF v4.1.1-CMAQ v5.3.1, WRF-Chem v4.1.1 and WRF v3.7.1-CHIMERE v2020r1, Zenodo [data set], https://doi.org/10.5281/zenodo.7925012, 2023b.*

*Chao Gao, Xuelei Zhang, Aijun Xiu, Qingqing Tong, Hongmei Zhao, Shichun Zhang, Guangyi Yang, Mengduo Zhang, Shengjin Xie: Chemical initial and boundary conditions for WRF-CMAQ, Zenodo [data set], https://doi.org/10.5281/zenodo.7932390, 2023c.*

*Chao Gao, Xuelei Zhang, Aijun Xiu, Qingqing Tong, Hongmei Zhao, Shichun Zhang, Guangyi Yang, Mengduo Zhang, Shengjin Xie: Chemical initial and boundary conditions for WRF-Chem. Zenodo [data set], https://doi.org/10.5281/zenodo.7932936, 2023d.*

*Chao Gao, Xuelei Zhang, Aijun Xiu, Qingqing Tong, Hongmei Zhao, Shichun Zhang, Guangyi Yang, Mengduo Zhang, Shengjin Xie: Chemical initial and boundary conditions for WRF-CHIMERE, Zenodo [data set], https://doi.org/10.5281/zenodo.7933641, 2023e.*

Chao Gao, Xuelei Zhang, Aijun Xiu, Qingqing Tong, Hongmei Zhao, Shichun Zhang, Guangyi Yang, Mengduo Zhang, Shengjin Xie: Emission input data for WRF-CMAQ, Zenodo [data set], https://doi.org/10.5281/zenodo.7932430, 2023f.

Chao Gao, Xuelei Zhang, Aijun Xiu, Qingqing Tong, Hongmei Zhao, Shichun Zhang, Guangyi Yang, Mengduo Zhang, Shengjin Xie: Emission input data for WRF-Chem, Zenodo [data set], https://doi.org/10.5281/zenodo.7932734, 2023g.

Chao Gao, Xuelei Zhang, Aijun Xiu, Qingqing Tong, Hongmei Zhao, Shichun Zhang, Guangyi Yang, Mengduo Zhang, Shengjin Xie: Emission input data for WRF-CHMIERE, Zenodo [data set], https://doi.org/10.5281/zenodo.7931614, 2023h.

Chao Gao, Xuelei Zhang, Aijun Xiu, Qingqing Tong, Hongmei Zhao, Shichun Zhang, Guangyi Yang, Mengduo Zhang, Shengjin Xie: Simulation results from WRF-CMAQ without aerosol feedbacks in eastern China for January-April 2017, Zenodo [data set], https://doi.org/10.5281/zenodo.7951404, 2023i_part1.

Chao Gao, Xuelei Zhang, Aijun Xiu, Qingqing Tong, Hongmei Zhao, Shichun Zhang, Guangyi Yang, Mengduo Zhang, Shengjin Xie: Simulation results from WRF-CMAQ without aerosol feedbacks in eastern China for May-August 2017, Zenodo [data set], https://doi.org/10.5281/zenodo.7951467, 2023i_part2.

Chao Gao, Xuelei Zhang, Aijun Xiu, Qingqing Tong, Hongmei Zhao, Shichun Zhang, Guangyi Yang, Mengduo Zhang, Shengjin Xie: Simulation results from WRF-CMAQ without aerosol feedbacks in eastern China for September-December 2017, Zenodo [data set], https://doi.org/10.5281/zenodo.7951475, 2023i_part3.

Chao Gao, Xuelei Zhang, Aijun Xiu, Qingqing Tong, Hongmei Zhao, Shichun Zhang, Guangyi Yang, Mengduo Zhang, Shengjin Xie: Simulation results from WRF-CMAQ with enabling aerosol-radiation interactions in eastern China for January-April 2017, Zenodo [data set], https://doi.org/10.5281/zenodo.7949895, 2023j_part1.

Chao Gao, Xuelei Zhang, Aijun Xiu, Qingqing Tong, Hongmei Zhao, Shichun Zhang, Guangyi Yang, Mengduo Zhang, Shengjin Xie: Simulation results from WRF-CMAQ with enabling aerosol-radiation interactions in eastern China for May-August 2017, Zenodo [data set], https://doi.org/10.5281/zenodo.7950644, 2023j_part2.

Chao Gao, Xuelei Zhang, Aijun Xiu, Qingqing Tong, Hongmei Zhao, Shichun Zhang, Guangyi Yang, Mengduo Zhang, Shengjin Xie: Simulation results from WRF-CMAQ with enabling aerosol-radiation interactions in eastern China for September-December 2017, Zenodo [data set], https://doi.org/10.5281/zenodo.7950830, 2023j_part3.

Chao Gao, Xuelei Zhang, Aijun Xiu, Qingqing Tong, Hongmei Zhao, Shichun Zhang, Guangyi Yang, Mengduo Zhang, Shengjin Xie: Simulation results from WRF-Chem without aerosol feedbacks in eastern China for January-March 2017, Zenodo [data set], https://doi.org/10.5281/zenodo.7943804, 2023k_part1.

Chao Gao, Xuelei Zhang, Aijun Xiu, Qingqing Tong, Hongmei Zhao, Shichun Zhang, Guangyi Yang, Mengduo Zhang, Shengjin Xie: Simulation results from WRF-Chem without aerosol feedbacks in eastern China for April-June 2017, Zenodo [data set], https://doi.org/10.5281/zenodo.7945383, 2023k_part2.

Chao Gao, Xuelei Zhang, Aijun Xiu, Qingqing Tong, Hongmei Zhao, Shichun Zhang, Guangyi Yang, Mengduo Zhang, Shengjin Xie: Simulation results from WRF-Chem without aerosol feedbacks in eastern China for July-September 2017, Zenodo [data set], https://doi.org/10.5281/zenodo.7946944, 2023k_part3.

Chao Gao, Xuelei Zhang, Aijun Xiu, Qingqing Tong, Hongmei Zhao, Shichun Zhang, Guangyi Yang, Mengduo Zhang, Shengjin Xie: Simulation results from WRF-Chem

*without aerosol feedbacks in eastern China for October-December 2017, Zenodo [data set], https://doi.org/10.5281/zenodo.7947169, 2023k_part4.*

*Chao Gao, Xuelei Zhang, Aijun Xiu, Qingqing Tong, Hongmei Zhao, Shichun Zhang, Guangyi Yang, Mengduo Zhang, Shengjin Xie: Simulation results from WRF-Chem with enabling aerosol-radiation interactions in eastern China for January-March 2017, Zenodo [data set], https://doi.org/10.5281/zenodo.7947050, 2023l_part1.*

*Chao Gao, Xuelei Zhang, Aijun Xiu, Qingqing Tong, Hongmei Zhao, Shichun Zhang, Guangyi Yang, Mengduo Zhang, Shengjin Xie: Simulation results from WRF-Chem with enabling aerosol-radiation interactions in eastern China for April-June 2017, Zenodo [data set], https://doi.org/10.5281/zenodo.7948216, 2023l_part2.*

*Chao Gao, Xuelei Zhang, Aijun Xiu, Qingqing Tong, Hongmei Zhao, Shichun Zhang, Guangyi Yang, Mengduo Zhang, Shengjin Xie: Simulation results from WRF-Chem with enabling aerosol-radiation interactions in eastern China for July-September 2017, Zenodo [data set], https://doi.org/10.5281/zenodo.7949410, 2023l_part3.*

*Chao Gao, Xuelei Zhang, Aijun Xiu, Qingqing Tong, Hongmei Zhao, Shichun Zhang, Guangyi Yang, Mengduo Zhang, Shengjin Xie: Simulation results from WRF-Chem with enabling aerosol-radiation interactions in eastern China for October-December 2017, Zenodo [data set], https://doi.org/10.5281/zenodo.7949561, 2023l_part4.*

*Chao Gao, Xuelei Zhang, Aijun Xiu, Qingqing Tong, Hongmei Zhao, Shichun Zhang, Guangyi Yang, Mengduo Zhang, Shengjin Xie: Simulation results from WRF-Chem with enabling aerosol-radiation interactions and aerosol-cloud interactions in eastern China for January-March 2017, Zenodo [data set], https://doi.org/10.5281/zenodo.7939221, 2023m_part1.*

*Chao Gao, Xuelei Zhang, Aijun Xiu, Qingqing Tong, Hongmei Zhao, Shichun Zhang, Guangyi Yang, Mengduo Zhang, Shengjin Xie: Simulation results from WRF-Chem with enabling aerosol-radiation interactions and aerosol-cloud interactions in eastern China for April-June 2017, Zenodo [data set], https://doi.org/10.5281/zenodo.7943002, 2023m_part2.*

*Chao Gao, Xuelei Zhang, Aijun Xiu, Qingqing Tong, Hongmei Zhao, Shichun Zhang, Guangyi Yang, Mengduo Zhang, Shengjin Xie: Simulation results from WRF-Chem with enabling aerosol-radiation interactions and aerosol-cloud interactions in eastern China for July-September 2017, Zenodo [data set], https://doi.org/10.5281/zenodo.7943079, 2023m_part3.*

*Chao Gao, Xuelei Zhang, Aijun Xiu, Qingqing Tong, Hongmei Zhao, Shichun Zhang, Guangyi Yang, Mengduo Zhang, Shengjin Xie: Simulation results from WRF-Chem with enabling aerosol-radiation interactions and aerosol-cloud interactions in eastern China for October-December 2017, Zenodo [data set], https://doi.org/10.5281/zenodo.7943323, 2023m_part4.*

*Chao Gao, Xuelei Zhang, Aijun Xiu, Qingqing Tong, Hongmei Zhao, Shichun Zhang, Guangyi Yang, Mengduo Zhang, Shengjin Xie: Simulation results from WRF-CHIMERE without aerosol feedbacks in eastern China for January-March 2017, Zenodo [data set], https://doi.org/10.5281/zenodo.7951775, 2023n_part1.*

*Chao Gao, Xuelei Zhang, Aijun Xiu, Qingqing Tong, Hongmei Zhao, Shichun Zhang, Guangyi Yang, Mengduo Zhang, Shengjin Xie: Simulation results from WRF-CHIMERE without aerosol feedbacks in eastern China for April-June 2017, Zenodo [data set], https://doi.org/10.5281/zenodo.7951779, 2023n_part2.*

*Chao Gao, Xuelei Zhang, Aijun Xiu, Qingqing Tong, Hongmei Zhao, Shichun Zhang, Guangyi Yang, Mengduo Zhang, Shengjin Xie: Simulation results from WRF-*

CHIMERE without aerosol feedbacks in eastern China for July-September 2017, Zenodo [data set], https://doi.org/10.5281/zenodo.7951791, 2023n_part3.

Chao Gao, Xuelei Zhang, Aijun Xiu, Qingqing Tong, Hongmei Zhao, Shichun Zhang, Guangyi Yang, Mengduo Zhang, Shengjin Xie: Simulation results from WRF-CHIMERE without aerosol feedbacks in eastern China for October-December 2017, Zenodo [data set], https://doi.org/10.5281/zenodo.7951793, 2023n_part4.

Chao Gao, Xuelei Zhang, Aijun Xiu, Qingqing Tong, Hongmei Zhao, Shichun Zhang, Guangyi Yang, Mengduo Zhang, Shengjin Xie: Simulation results from WRF-CHIMERE with enabling aerosol-radiation interactions in eastern China for January-March 2017, Zenodo [data set], https://doi.org/10.5281/zenodo.7952838, 2023o_part1.

Chao Gao, Xuelei Zhang, Aijun Xiu, Qingqing Tong, Hongmei Zhao, Shichun Zhang, Guangyi Yang, Mengduo Zhang, Shengjin Xie: Simulation results from WRF-CHIMERE with enabling aerosol-radiation interactions in eastern China for April-June 2017, Zenodo [data set], https://doi.org/10.5281/zenodo.7952840, 2023o_part2.

Chao Gao, Xuelei Zhang, Aijun Xiu, Qingqing Tong, Hongmei Zhao, Shichun Zhang, Guangyi Yang, Mengduo Zhang, Shengjin Xie: Simulation results from WRF-CHIMERE with enabling aerosol-radiation interactions in eastern China for July-September 2017, Zenodo [data set], https://doi.org/10.5281/zenodo.7952842, 2023o_part3.

Chao Gao, Xuelei Zhang, Aijun Xiu, Qingqing Tong, Hongmei Zhao, Shichun Zhang, Guangyi Yang, Mengduo Zhang, Shengjin Xie: Simulation results from WRF-CHIMERE with enabling aerosol-radiation interactions in eastern China for October-December 2017, Zenodo [data set], https://doi.org/10.5281/zenodo.7952844, 2023o_part4.

Chao Gao, Xuelei Zhang, Aijun Xiu, Qingqing Tong, Hongmei Zhao, Shichun Zhang, Guangyi Yang, Mengduo Zhang, Shengjin Xie: Simulation results from WRF-CHIMERE with enabling aerosol-radiation interactions and aerosol-cloud interactions in eastern China for January-March 2017, Zenodo [data set], https://doi.org/10.5281/zenodo.7952859, 2023p_part1.

Chao Gao, Xuelei Zhang, Aijun Xiu, Qingqing Tong, Hongmei Zhao, Shichun Zhang, Guangyi Yang, Mengduo Zhang, Shengjin Xie: Simulation results from WRF-CHIMERE with enabling aerosol-radiation interactions and aerosol-cloud interactions in eastern China for April-June 2017, Zenodo [data set], https://doi.org/10.5281/zenodo.7952863, 2023p_part2.

Chao Gao, Xuelei Zhang, Aijun Xiu, Qingqing Tong, Hongmei Zhao, Shichun Zhang, Guangyi Yang, Mengduo Zhang, Shengjin Xie: Simulation results from WRF-CHIMERE with enabling aerosol-radiation interactions and aerosol-cloud interactions in eastern China for July-September 2017, Zenodo [data set], https://doi.org/10.5281/zenodo.7952865, 2023p_part3.

Chao Gao, Xuelei Zhang, Aijun Xiu, Qingqing Tong, Hongmei Zhao, Shichun Zhang, Guangyi Yang, Mengduo Zhang, Shengjin Xie: Simulation results from WRF-CHIMERE with enabling aerosol-radiation interactions and aerosol-cloud interactions in eastern China for October-December 2017, Zenodo [data set], https://doi.org/10.5281/zenodo.7952867, 2023p_part4.

Chao Gao, Xuelei Zhang, Aijun Xiu, Qingqing Tong, Hongmei Zhao, Shichun Zhang, Guangyi Yang, Mengduo Zhang, Shengjin Xie: Data used to create figures and tables in the GMD manuscript "Inter-comparison of multiple two-way coupled meteorology and air quality models (WRF v4.1.1-CMAQ v5.3.1, WRF-Chem

*v4.1.1 and WRF v3.7.1-CHIMERE v2020r1) in eastern China", Zenodo [data set], https://doi.org/10.5281/zenodo.7750907, 2023q.*

Thank you again for your help and please let me know if we need to make any further revisions to improve the quality of our paper.

Sincerely yours,

Chao Gao, PhD

Assistant Professor, Key Laboratory of Wetland Ecology and Environment

Northeast Institute of Geography and Agroecology, Chinese Academy of Sciences

---

## Author Comment (AC2)

We would like to express our sincere appreciation to the reviewer for the valuable and constructive suggestions, which have helped us improve the quality of this manuscript. We have addressed all these comments carefully and revised the manuscript accordingly. Following the Reviewer' comments in black, please find our point-to-point responses in blue. Hereafter, all new added or modified sentences are marked in blue and italic in this response.

**Anonymous Referee #1**

1. Introduction. "The feedbacks of aerosols to meteorology have been widely investigated by two-way coupled meteorology and air quality models in the past two decades." Two-way coupled meteorological and air quality models have been developed and applied for almost three decades (Jacobson, 1994; 1997; 1998, 2001).

Response: According to this suggestion, the sentence in Introduction has been revised as "*The feedbacks of aerosols to meteorology have been widely investigated by two-way coupled meteorology and air quality models in the past three decades (Jacobson, 1994, 1997, 1998, 2001, 2002; Grell et al., 2005; Wong et al., 2012; Wang et al., 2014; Zhou et al., 2016; Briant et al., 2017; Feng et al., 2021).*" in the revised manuscript.

2. Table 1. what is the vertical resolution of the boundary layer in each model (how many layers in the bottom 1 km and what is the bottom-layer thickness?

Response: All the three coupled models used in this study have 30 levels (i.e., 29 layers) from the surface to 100 hPa. There are 11 layers in the bottom 1 km and the bottom-layer thickness is 23.2 m. The sentence "The vertical resolution for all simulations consisted of 30 levels from the surface (~20 m) to 100 hPa." was revised as "*All the three coupled models used in this study have 30 levels* (i.e., 29 layers) *from the surface to 100 hPa with 11 layers in the bottom 1 km and the bottom-layer thickness being 23.2 m.*". We also revised Table 1 accordingly.

*Table 1. Model setups and inputs for the two-way coupled models (WRF-CMAQ, WRF-Chem and WRF-CHIMERE).*

| | | WRF-CMAQ | WRF-Chem | WRF-CHIMERE |
|---|---|---|---|---|
| *Domain* | Horizontal grid spacing | 27 km (110 × 150) | 27 km (120 × 160) | 27 km (120 × 170) |
| *configuration* | Vertical resolution | *30 levels* | *30 levels* | *30 levels* |
| *Physics* | Shortwave radiation | RRTMG | RRTMG | RRTMG |
| *parameterization* | Longwave radiation | RRTMG | RRTMG | RRTMG |

|  |  |  |  |  |
|---|---|---|---|---|
|  | Cloud microphysics | Morrison | Morrison | Thompson |
|  | PBL | ACM2 | YSU | YSU |
|  | Cumulus | Kain-Fritsch | Grell-Freitas | Grell-Freitas |
|  | Surface | Pleim-Xiu | Monin-Obukhov | Monin-Obukhov |
|  | Land surface | Pleim-Xiu LSM | Noah LSM | Noah LSM |
|  | *Icloud* | *Xu-Randall method* | *Xu-Randall method* | *Xu-Randall method* |
| *Chemistry scheme* | *Aerosol mechanism* | *AERO6* | *MOSAIC* | *SAM* |
|  | *Aerosol size distribution* | *Modal (3 modes)* | *Sectional (4 bins)* | *Sectional (10 bins)* |
|  | Aerosol mixing state | Core-Shell | Core-Shell | Core-Shell |
|  | Gas-phase chemistry | CB6 | CBMZ | MELCHIOR2 |
|  | Photolysis | *Fast-JX with cloud effects* | *Fast-JX with cloud effects* | *Fast-JX with cloud effects* |
| *Emission* | Anthropogenic emission | MEIC 2017 | MEIC 2017 | MEIC 2017 |
|  | Biogenic emission | MEGAN v3.0 | MEGAN v3.0 | MEGAN v3.0 |
|  | Biomass burning emission | FINN v1.5 | FINN v1.5 | FINN v1.5 |
|  | Dust emission | Foroutan | GOCART | Menut |
|  | Sea-salt emission | Gong | Gong | Monahan |
| *Input data* | Meteorological ICs and BCs | FNL | FNL | FNL |
|  | Chemical ICs and BCs | MOZART | MOZART | LMDZ-INCA |

3. Table 1. How many aerosol size bins and components per bin? Do you use a modal or discrete bin approach?

Response: For aerosol size distribution, the modal approach was used in the WRF-CMAQ model (Binkowski and Roselle, 2003) and included Aitken, accumulation and coarse modes with 9 (black carbon (BC), organic carbon (OC), sulfate, nitrate, ammonium, remaining unspeciated particulate matter (PMOTHR), primary non-carbon organic matter (PNCOM), water, metals), 11 (BC, OC, sulfate, nitrate, ammonium, PMOTHR, PNCOM, water, metals, sea salt, dust) and 3 (coarse primary particulate matter (PMC), sea salt, dust) aerosol components, respectively. WRF-Chem and WRF-CHIMERE applied the sectional approach with 4 and 10 size bins covering dry diameters ranging from 0.039 to 10 μm and 0.039 to 40 μm, respectively (Zaveri et al., 2008; Nicholls et al., 2014; Menut et al., 2013, 2016). In WRF-Chem, BC, OC, sulfate, nitrate and sea salt are put in Bins 1–3 and dust, sea salt, and other inorganic matter (OIN) in Bin 4. For WRF-CHIMERE, BC, OC, sulfate and primary particulate matter (PPM) are assigned in Bins 1–5, BC, OC, sulfate, dust and sea salt in Bin 6, dust and sea salt in Bins 7 & 9, BC, OC, PPM, dust and sea salt in Bin 8 and dust in Bin 10. The approaches for aerosol size distributions used in the three coupled models are listed in the revised Table 1, as shown in the reply of Question 2. We also compiled all the components in each mode or bin in Table S2 and added it into the

Supplement of the revised manuscript. In addition, we added the sentence "*As illustrated in Table 1 and Table S2 for aerosol size distribution, we used modal approach with Aitken, accumulation and coarse modes in WRF-CMAQ, and the 4-bin and 10-bin sectional approaches in WRF-Chem and WRF-CHIMERE models, respectively (Binkowski and Roselle, 2003; Zaveri et al., 2008; Nicholls et al., 2014; Menut et al., 2013).*" in the Section 2.1 of the revised manuscript. We revised the sentence "These biases were produced by the configurations of different aerosol and gas phase mechanisms, online dust emission schemes, and chemical ICs and BCs in the two-way coupled models.". In lines 536-538 of the revised manuscript, the sentence "These biases were produced by the configurations of different aerosol and gas phase mechanisms, online dust emission schemes, and chemical ICs and BCs in the two-way coupled models." is revised as "*These biases could be related to different aerosol and gas phase mechanisms, dust and sea salt emission schemes, chemical ICs and BCs, and aerosol size distribution treatments applied in the three two-way coupled models.*".

*Table S2. Summary of the aerosol size distribution treatments and components in each mode or bin for the coupled WRF-CMAQ, WRF-Chem and WRF-CHIMERE models.*

| Model | Aerosol mechanism | Modal approach | | | | | | | | |
|---|---|---|---|---|---|---|---|---|---|---|
| | | Aitken | | Accumulation | | Coarse | | | | |
| WRF-CMAQ | AERO6 | BC, OC, sulfate, nitrate, ammonium, PMOTHR[d], PNCOM[e] water, metals | | BC, OC, sulfate, nitrate, ammonium, PMOTHR, PNCOM, water, metals, sea salt | | PMC[f], sea salt, dust | | | | |
| | | Sectional approach | | | | | | | | |
| WRF-Chem | MOSAIC[a] | Bin 1 | | Bin 2 | | Bin 3 | | Bin 4 | | |
| | | 0.039–0.156 μm | | 0.156–0.625 μm | | 0.625–2.5 μm | | 2.5–10.0 μm | | |
| | | BC, OC, sulfate, nitrate, sea salt[d] | | BC, OC, sulfate, nitrate, sea salt | | BC, OC, sulfate, nitrate, sea salt | | Duste, sea salt, OIN[g] | | |
| WRF-CHIMERE | SAM[b] | Bin 1 | Bin 2 | Bin 3 | Bin 4 | Bin 5 | Bin 6 | Bin 7 | Bin 8 | Bin 9 | Bin 10 |
| | | 0.039–0.078 μm | 0.078–0.156 μm | 0.156–0.312 μm | 0.312–0.625 μm | 0.625–1.25 μm | 1.25–2.5 μm | 2.5–5.0 μm | 5.0–10.0 μm | 10.0–20.0 μm | 20.0–40.0 μm |
| | | BC, OC, sulfate, PPM[c] | BC, OC, sulfate, PPM | BC, OC, sulfate, PPM | BC, OC, sulfate, PPM | BC, OC, sulfate, PPM | BC, OC, sulfate, dust, sea salt | Dust, sea salt | BC, OC, PPM, dust, sea salt | Dust, sea salt | Dust |

[a]MOSAIC is the Model for Simulating Aerosol Interactions and Chemistry, and the cbmz-mosaic emissions in "PNNL" format (emiss_inpt_opt==101) was used in WRF-Chem simulations.

[b]SAM is the sectional aerosol mechanism.

[c]PPM is the primary particulate matter.

[d]PMOTHR is the remaining unspeciated particulate matter in fine mode and more detailed information is at https://www.airqualitymodeling.org/index.php/CMAQv5.0_PM_emitted_species_list.

[e]PNCOM is the primary non-carbon organic matter in fine mode and more detailed information is at https://www.airqualitymodeling.org/index.php/CMAQv5.0_PM_emitted_species_list.

[f]PMC is the primary particulate matter in coarse mode and more detailed information is at https://www.airqualitymodeling.org/index.php/CMAQv5.0_PM_emitted_species_list.

[g]OIN is the other inorganic matter.

4. Table 1. Does photolysis account for clouds? How are clouds treated for radiative transfer calculations?

Response: Yes, all the three coupled models considered the effects of cloud on photolysis in the photolysis calculation. Even though the Fast-JX photolysis scheme was applied in the three coupled models, how the cloud effects were treated was different. For WRF-

CMAQ, the impacts of cloud cover and cloud optical properties on the radiative transfer and actinic flux are taken into account. Both cloud fraction (CF) from WRF and CF calculated using relative humidity (RH) and RH thresholds (set to 0.85 over ocean and 0.75 over land (Mocko and Cotton, 1995)) are utilized in the CMAQ version of Fast-JX (Sundqvist et al., 1989). The total column CF is determined by exponential-random overlapping. The optical properties of hydrometeors (cloud liquid water, rain, snow, graupel and ice) output from WRF are included in the computation of cloud optical properties in the CMAQ version of Fast-JX (Hu and Stamnes, 1993; Fu, 1996; Binkowski et al., 2007). In the WRF-Chem version of Fast-JX, CF is set to 1 when cloud liquid water content (CLWC) is greater than 0 and CF is set to 0 when CLWC = 0, and the calculation of cloud optical depth only considers CLWC from WRF. In WRF-CHIMERE, CF = 1 when CLWC or cloud ice content (CIC) is greater than $0.00001$ g m$^{-3}$ and CF = 0 if CLWC or CIC is 0. To compute cloud optical depth in the CHIMERE version of Fast-JX, both cloud liquid water and ice output from WRF are taken into account (Mailler et al., 2017).

These information is reflected in the revised Table 1, and we also added this sentence *"In the Fast-JX photolysis scheme used by the three coupled models, the impacts of clouds are included by considering cloud cover and cloud optical properties. However, the calculations of cloud cover and cloud optical properties are different in these models and all the relevant information is listed in Table S1."* in Lines 166-170 of the revised manuscript. Table S5 is in Supplement of the revised manuscript.

*Table S1. Summary of representations of cloud cover and cloud optical properties in the Fast-JX scheme for WRF-CMAQ, WRF-Chem and WRF-CHIMERE.*

| Model | Cloud clover | Cloud optical properties | | | |
| --- | --- | --- | --- | --- | --- |
| | | Optical properties | Effective Wavelength | Hydrometeor types | Method |
| WRF-CMAQ | 1. CF[a] from WRF and CF calculated using RH and RH thresholds
2. Exponential-random overlapping | Extinction, single scattering albedo and asymmetry factor | 294.6, 303.2, 310.0, 316.4, 333.1, 382.0 and 607.7 nm | Cloud liquid water, rain, snow, graupel and ice | The parameterizations proposed by Hu and Stamnes (1993) and Fu (1996) |
| WRF-Chem | 1. CF=0 if CLWC[b]=0
2. CF=1 if CIC[c]>0 | Cloud optical depth | 300, 400, 600 and 999 nm | Cloud liquid water | Based on the empirical functions of relative humidity and cloud liquid water content |
| WRF-CHIMERE | 1. CF=0 if CLWC or CIWC=0
2. CF=1 if CLWC or CIC>0 | Cloud optical depth | 200, 300, 400, 600, and 999 nm | Cloud liquid water and ice | Based on the functions of cloud effective radiuses and cloud liquid water/ice contents |

[a]CF is fraction. [b]CLWC is cloud liquid water content. [c]CIC is cloud ice content.

In this study, the RRTMG shortwave radiation (SWR) and longwave radiation (LWR) schemes were chosen for the three two-way coupled models. The considerations of cloud

effects on SWR and LWR in RRTMG are twofold, as listed below:

(1) Regarding the effects of cloud cover on radiation: the cloud fraction (CF) at grid scale is calculated using relative humidity and mixing ratio of all hydrometeors (Xu and Randall, 1996) and then the total column CF is determined by maximum-random overlapping (Iacono et al., 2008). The cumulus CF is only considered when the Kain-Fritsch cumulus scheme is chosen and computed as a function of the updraft mass flux in cloud (Alapaty et al., 2012). Therefore, the coupled WRF-CMAQ model with the Kain-Fritsch cumulus scheme included the cumulus CF impacts on RRTMG radiation but not the WRF-Chem and WRF-CHIMERE models with the Grell-Freitas cumulus scheme.

(2) Regarding the impacts of cloud optical properties on radiation: the treatments of cloud liquid water and ice optical properties as proposed by Hu and Stamnes (1993) and Fu (1996) are deployed in both RRTMG SWR and LWR schemes in all three coupled models.

Thus, we added these descriptions in Lines 160-163 of the revised manuscript as follows.

"*To consider the effects of clouds on radiative transfer calculations, the fractional cloud cover and cloud optical properties were included in the RRTMG shortwave/longwave radiation schemes used by all three coupled models (Xu and Randall, 1996; Iacono et al., 2008). The coupled WRF-CMAQ model with the Kain-Fritsch cumulus scheme included the cumulus cloud fraction impacts on RRTMG radiation (Alapaty et al., 2012), but not the WRF-Chem and WRF-CHIMERE models with the Grell-Freitas cumulus scheme.*"

5. Table 1. What height is the model top and how are model-top boundary conditions treated?

Response: The height of model top is about 16 km (100 hPa). For the meteorological model-top boundary conditions, WRF assumes zero flux at the model top. Regarding the chemical model-top boundary conditions, WRF-CMAQ and WRF-Chem models both take into account the impacts of stratosphere-troposphere $O_3$ exchange using the parameterization of $O_3$-potential vorticity (Safieddine et al., 2014; Xing et al., 2016). For WRF-CHIMERE, climatological data from the Laboratoire de Météorologie Dynamique (LMDz) coupling a global chemistry and aerosol model INteractions between Chemistry and Aerosols (INCA) were used for model-top boundary conditions (Mailler et al., 2017).

To distinguish lateral and model-top BCs used in this study, these sentences are edited

in the revised manuscript as follows:

"The meteorological ICs and BCs were derived from the National Center for Environmental Prediction Final Analysis (NCEP-FNL) datasets (http://rda.ucar.edu/datasets/ds083.2), with a horizontal resolution of 1° × 1° at 6-hour intervals for each of the three coupled models." was revised as "*The meteorological ICs and lateral BCs were derived from the National Center for Environmental Prediction Final Analysis (NCEP-FNL) datasets (http://rda.ucar.edu/datasets/ds083.2), with a horizontal resolution of 1° × 1° at 6-hour intervals for each of the three coupled models, and the flux at the model-top boundary is set to zero.*".

"The chemical ICs/BCs were downscaled from the Whole Atmosphere Community Climate Model (WACCM) for WRF-CMAQ and WRF-Chem via the mozart2camx and mozbc tools, respectively." was revised as "*The chemical ICs/lateral BCs were downscaled from the Whole Atmosphere Community Climate Model (WACCM) for WRF-CMAQ and WRF-Chem via the mozart2camx and mozbc tools, respectively. WRF-CHIMERE used the climatology data from a general circulation model developed at the Laboratoire de Météorologie Dynamique (LMDz) coupling a global chemistry and aerosol model INteractions between Chemistry and Aerosols (INCA) (Mailler et al., 2017). For chemical model-top BCs, WRF-CMAQ and WRF-Chem models both take into account the impacts of stratosphere-troposphere $O_3$ exchange using the parameterization of $O_3$-potential vorticity (Safieddine et al., 2014; Xing et al., 2016), and WRF-CHIMERE utilized the LMDz-INCA climatology data (Mailler et al., 2017).*"

6. The authors evaluate with RMSE, which is an absolute quantity for each variable. However, normalized gross error (absolute value of differences between model and data, divided by data, summed over all locations and normalized by the number of locations, is a more useful metric since it gives error relative to the data values rather than an absolute amount. It is similar to NMB, but with absolute values taken, since NMB cancels out large errors of the opposite sign. Also, it would be useful to see some time-series plots of model results versus data.

Response: We agree that it would be useful to add the normalized gross error (NGE) in our simulation assessment. We added NGE in Table 3, Table 4, Table 5 and Table S3 as well as descriptions in the revised manuscript.
Lines 226-227: "*normalized gross error (NGE)*"

Lines 565-568: "*All models with ARI feedbacks enabled resulted in slight decreases in annual and seasonal $O_3$ NMBs and NGEs, ranging from −3.02% to +0.85% (the only positive value of +0.85% was produced by WRF-CMAQ in summer) and from −1.42% to −0.75%, respectively.*"

Lines 568-570: "*Meanwhile, for ACI effects, WRF-Chem and WRF-CHIMERE had increased annual $O_3$ NMBs and NGEs of 0.12%−0.65% and 0.40%−0.55%, respectively.*"

We presented the time-series plots of simulated and observed hourly meteorology and air qualit over Eastern China during the year of 2017 in Figs. 3 and 6, respectively. The metrological variables involved surface shortwave radiation (SSR), temperature (T2), specific humidity (Q2), relative humidity (RH2) and wind speed (WS10). The air quality variables included $PM_{2.5}$ and $O_3$ concentrations. These two figures are put into the revised manuscript.

The related descriptions are added in the revised manuscripts as follows:

Lines 278-280: "*Looking at annual and seasonal T2, models tended to have a negative bias, and T2 underestimations in spring and winter were greater than those in summer and autumn (Figs. 3 and 4).*"

Lines 317-319: "*The R values for all three models ranged from 0.47 to 0.60; WRF-CMAQ and WRF-Chem overestimated wind speed by approximately 0.5 m s$^{-1}$, while WRF-CHIMERE overestimated it by approximately 1.0 m s$^{-1}$ (Table S3 and Figs. 3–4)).*"

Lines 531-535: "*As shown in Table 4 and Figs. 6–7, WRF-CMAQ underestimated annual and seasonal (except for autumn) $PM_{2.5}$ concentrations with NMBs ranging from −9.78% to −6.39% and −17.68% to +5.17%, respectively. WRF-Chem generated both overestimations and underestimations of $PM_{2.5}$ at the annual and seasonal scales, with related NMBs varying from −39.11% to +24.72%, respectively.*"

[revised manuscript text omitted]

7. A lot of comparisons are performed, but what are the most relevant comparisons with data? Ozone and PM$_{2.5}$ calculations? Please focus the discussion of results more. Right now the results section is crammed with lots of information that is not easy to determine from what is important and not important.

Response: We agreed that we need to be more focused while evaluating the simulation results from the three coupled models. At the same time, we believe the most relevant comparisons in this paper should look into the surface meteorological variables (SSR, T2, RH2, WS10) and air quality variables (PM$_{2.5}$ and O$_3$). The comparisons against satellite data should focus on SSR, SLR, PREC, cloud fraction, and cloud liquid water path. To improve the paper's readability, we rearranged some paragraphs and figures and added sentences in the revised manuscript, as listed below:

(1) The results and discussion about the comparisons of simulated Q2, PREC, PBLH00, PBLH12 against ground-based observations are moved to Section 1.1 of Supplement. In Lines 250-252 of the revised manuscript, we added "Here, we mainly focused on the comparisons of SSR, T2, RH2, and WS10, and the analysis of PREC, PBLH00, and PBLH12 are presented in Section 1.1 of Supplement."

(2) The comparisons of simulated TSR and TLR against satellite observations are moved to Section 1.2 of Supplement. We modified the sentences in Lines 373-380 of the revised manuscript as "To further evaluate the performance of WRF-CMAQ, WRF-Chem, and WRF-CHIMERE against satellite observations, we analyzed the annual and seasonal statistical metrics of short- and long-wave radiation at the surface, precipitation, cloud

cover, and liquid water path simulated by the three coupled models with and without aerosol feedbacks, via comparisons between simulations and satellite-borne observations (Table 3; Figures 5, S9, S12–S14). In addition, the evaluations of short- and long-wave radiation at top of the atmosphere (TOA) are presented in Section 1.2 of Supplement."

(3) The evaluation of simulated $NO_2$, $SO_2$ and CO against surface measurements is moved to Section 2 of Supplement. In Lines 525-527 of the revised manuscript, we added "The evaluations between surface measurements and simulations of $PM_{2.5}$ and $O_3$ are presented below, and the performance assessments of other gaseous pollutants are in Section 2 of Supplement."

(4) The original Figure 4 and Figure 7 are moved to Supplement as Figure S8 and Figure S20, respectively.

We added more discussions of in-depth analysis in the result part of revised manuscript as follows:

Lines 401-402: "*the representation differences for aerosol components, size distributions and mechanisms contributed to the diversity of seasonal MBs (Tables 1 and S2).*"

Lines 407-421: "*When ARI effects are enabled, the diversities of refractive indices of aerosol species groups lead to the discrepancies of online calculated aerosol optical properties in different shortwave and longwave (SW and LW) bands in the RRTMG SW/LW radiation schemes of WRF-CMAQ, WRF-Chem, and WRF-CHIMERE (Tables S5–S6). The online calculated cloud optical properties induced by aerosol absorption in the RRTMG radiation schemes are different in treatments of aerosol species groups in the three coupled models. With enabling ACI effects, the activation of cloud droplets from aerosols based on the Köhler theory is taken into account in WRF-Chem and WRF-CHIMERE, in comparison to simulations without aerosol feedbacks (Table S7). The treatments of prognostic ice nucleating particles (INP) formed via heterogeneous nucleation of dust particles (diameters > 0.5 µm) and homogeneous freezing of hygroscopic aerosols (diameters > 0.1 µm) are only considered in WRF-CHIMERE, but the prognostic ice nucleating particles are not included in WRF-CMAQ and WRF-Chem. These discrepancies eventually contribute to the differences of simulated radiation changes caused by aerosols.*"

Lines 485-495: "*This may be explained as the different parameterization treatments of cloud droplet number concentration (CDNC) simulated by the three coupled models with/without enabling ACI effects. The cloud condensation nuclei activated from aerosol*

*particles can increase CDNC and impact on LWP and CF. Without enabling any aerosol feedbacks or only enabling ARI, the CDNC is default prescribed as a constant value of 250 cm$^{-3}$ in the Morrison scheme of WRF-CMAQ and WRF-Chem and 300 cm$^{-3}$ in the Thompson scheme of WRF-CHIMERE. When only ACI or both ARI and ACI are enabled, the online calculating of prognostic CDNC is performed in WRF-Chem and WRF-CHIMERE by using the method of maximum supersaturation (Abdul-Razzak and Ghan, 2002; Chapman et al., 2009; Tuccella et al., 2019)."*

Lines 537-539: *"These biases could be related to different aerosol and gas phase mechanisms, dust and sea salt emission schemes, chemical ICs and BCs, and aerosol size distribution treatments applied in the three two-way coupled models."*

Lines 573-578: *"Such diversity in NMB and NGE variations can be explained by two aspect differences. For model-top boundary conditions, the WRF-CMAQ and WRF-Chem models employed the parameterization scheme of O$_3$-potential vorticity and WRF-CHIMERE used the climatological data from LMDz-INCA. For gas-phase chemistry mechanisms, three coupled models incorporate a variety of photolytic reactions, with a more comprehensive explanation provided in Section 4.2."*

Lines 675-685: *"More detailed interpretations were grouped into four aspects: (1) AODs are calculated via Mie theory using refractive indices of different numbers (5, 6 and 10) of aerosol species group in different coupled models (WRF-CMAQ, WRF-Chem and WRF-CHIMERE) (Tables S5–S6); (2) 7 (294.6, 303.2, 310.0, 316.4, 333.1, 382.0 and 607.7 nm), 4 (300, 400, 600 and 999 nm), and 5 (200, 300, 400, 600, and 999 nm) effective wavelengths are used in calculating actinic fluxes and photolysis rates in Fast-JX photolysis modules of WRF-CMAQ, WRF-Chem and WRF-CHIMERE, respectively;(3) Different calculating methods of aerosol and cloud optical properties exist in the Fast-JX schemes of three coupled models (Tables S1 and S5–S6); (4) 77, 52 and 40 gas-phase species involve 218, 132, 120 gas-phase reactions in CB6, CBMZ and MELCHIOR2 mechanisms, respectively."*

The added references were listed as follows.

*Jacobson, M. Z., Developing, coupling, and applying a gas, aerosol, transport, and radiation model to study urban and regional air pollution. Ph. D. Thesis, Dept. of Atmospheric Sciences, University of California, Los Angeles, 436 pp., 1994.*
*Jacobson, M. Z., Development and application of a new air pollution modeling system. Part III: Aerosol-phase simulations, Atmos. Environ., 31A, 587–608, 1997.*

Jacobson, M. Z., Studying the effects of aerosols on vertical photolysis rate coefficient and temperature profiles over an urban airshed, J. Geophys. Res., 103, 10,593-10,604, 1998.

Jacobson, M. Z., GATOR-GCMM: A global through urban scale air pollution and weather forecast model. 1. Model design and treatment of subgrid soil, vegetation, roads, rooftops, water, sea ice, and snow, J. Geophys. Res., 106, 5385-5401, 2001.

Binkowski F S, Roselle S J. Models-3 Community Multiscale Air Quality (CMAQ) model aerosol component 1. Model description[J]. Journal of geophysical research: Atmospheres, 2003, 108(D6).

Zaveri R A, Easter R C, Fast J D, et al. Model for simulating aerosol interactions and chemistry (MOSAIC)[J]. Journal of Geophysical Research: Atmospheres, 2008, 113(D13).

Menut L, Bessagnet B, Khvorostyanov D, et al. CHIMERE 2013: a model for regional atmospheric composition modelling[J]. Geoscientific model development, 2013, 6(4): 981-1028.

Bessagnet B, Hodzic A, Vautard R, et al. Aerosol modeling with CHIMERE—preliminary evaluation at the continental scale[J]. Atmospheric environment, 2004, 38(18): 2803-2817.

Appel K W, Napelenok S L, Foley K M, et al. Description and evaluation of the Community Multiscale Air Quality (CMAQ) modeling system version 5.1[J]. Geoscientific model development, 2017, 10(4): 1703-1732.

Chapman E G, Gustafson Jr W I, Easter R C, et al. Coupling aerosol-cloud-radiative processes in the WRF-Chem model: Investigating the radiative impact of elevated point sources[J]. Atmospheric Chemistry and Physics, 2009, 9(3): 945-964.

Mailler S, Menut L, Khvorostyanov D, et al. CHIMERE-2017: From urban to hemispheric chemistry-transport modeling[J]. Geoscientific Model Development, 2017, 10(6): 2397-2423.

Xing J, Mathur R, Pleim J, et al. Representing the effects of stratosphere–troposphere exchange on 3-D $O_3$ distributions in chemistry transport models using a potential vorticity-based parameterization[J]. Atmospheric Chemistry and Physics, 2016, 16(17): 10865-10877.

Grell G A, Peckham S E, Schmitz R, et al. Fully coupled "online" chemistry within the WRF model[J]. Atmospheric environment, 2005, 39(37): 6957-6975.

Iacono M J, Delamere J S, Mlawer E J, et al. Radiative forcing by long-lived greenhouse gases: Calculations with the AER radiative transfer models[J]. Journal of Geophysical Research: Atmospheres, 2008, 113(D13).

---

## Author Comment (AC3)

We would like to thank the reviewer for the constructive comments really helping to improve the paper. Below, we address each comment in full detail. Following the Reviewers' comments in black, please find our point-to-point responses in blue. Hereafter, all new added or modified sentences are marked in blue and italic in this response.

**Anonymous Referee #2**

I have some major concerns of using FDDA for a study like this and try to objectively learn from model performance. When using FDDA, nudging is forcing the model to the same observations that are being used for evaluation (NCEP reanalysis includes probably all observations that are using for evaluation). Additionally, FDDA makes conclusions for feedback studies very problematic, since the physics parameterizations react in very different ways. Furthermore, you are using different physics and chemistry routines in all models. This makes this an "apples to oranges" comparison. Do you know what happens in Morrison microphysics when the mix-activation routine is not called (no chemistry)? My first thought was to reject the paper, however, the authors have done an immense amount of work and present some useful results that can be used by some of the developers. In turn I will propose accepting but with major revisions. These major revisions should be focused on the interpretation of the results. Abstract and conclusions should clearly say that this work is NOT to decide which model is better or worse since employed setups are very different and furthermore, it is not clear how FDDA runs influences feedback studies. Also, the authors need to be clear on what is used by Radiation (R) and MicroPhysics (MP) if feedback is off versus on. Are you just using a constant droplet number? A climatology? WRF-Chem has a lot of options, how come you decided to use different physics than in WRF-CMAQ? Chimere is way behind in the WRF version used, which makes that one even harder to compare. I am not asking you to rerun this monster simulation, but you will need to rephrase some of your abstract, conclusion, and results description. Since this reviewer is not asking for additional runs, this should not be a major effort. I really appreciate the work you folks put into this paper!

It would be interesting, maybe in a later additional paper, to compare the feedbacks in the different models for a shorter run that does not use FDDA. Maybe picking one or several interesting 5 day periods from the long runs that you used for this paper.

To answer the major concerned questions brought by the reviewer about FDDA in the coupled models, we put our response into three parts:

Enabling FDDA can improve the accuracy of simulated meteorology, e.g., temperature, wind speed and precipitation (Otte et al., 2012; Sommerfeld et al., 2019; Wang et al., 2016; Huang et al., 2021), and air quality, e.g., $PM_{2.5}$, $PM_{10}$, and $O_3$ (Barna and Lamb, 2000; Otte, 2008a, 2008b; Jeon et al., 2015; Tran et al., 2018). Enabling FDDA in WRF is

to decrease the error accumulations and avoid the significant deviations between simulation and observation, it is particularly beneficial in dynamic analyses of long-term simulations of meteorology and air quality (Otte et al., 2008). In this study, the 6-hourly NCEP-FNL data were used in the FDDA nudging, and hourly evaluations were conducted. We set the nudging coefficients for u/v components, temperature and water vapor mixing ratio above the planetary boundary layer (PBL) as 0.0001, 0.0001 and 0.00001 $s^{-1}$, respectively. The nudging coefficients for surface u/v components, temperature and water vapor mixing ratio are set to 0. Since we mainly focused on surface variables in the most evaluations of this study, the considerations of FDDA have relative limited impacts on our evaluation results.

Until now, the impacts of enabling FDDA on aerosol feedback effects are still under debate:

(1) Previous studies pointed out that enabling FDDA can reduce the simulated effects of aerosol feedbacks. The impact of aerosol feedbacks is diminished when comparing two-way coupled models of WRF-Chem and WRF-CMAQ with enabling FDDA to those without enabling FDDA (Forkel et al., 2012; Hogrefe et al., 2015; Zhang et al., 2016), and suggested that future works should be achieved the optimal balance between enabling options of FFDA and aerosol feedbacks. In the perspective part of Wong et al. (2012), the authors also pointed that aerosol-radiation interaction (ARI) effects could be attenuated by enabling FDDA, depending on the strength of the nudging coefficients employed. Hogrefe et al. (2015) emphasized that it is difficult to identify the extent to which nudging may have diminished the impact of simulated aerosol feedback effects. Referring to the model setting experiences of long-term aerosol feedback simulations in Gan et al. (2015) and Xing et al. (2015), the nudging coefficients for u/v wind, temperature, and water vapor mixing ratio were set to 0.00005 $s^{-1}$, 0.00005 $s^{-1}$, and 0.00001 $s^{-1}$, respectively.

(2) Several researches have used the coupled models with FFDA nudging technology to investigate aerosol feedback effects at the regional scales. All nudging coefficients of u/v components, temperature and water vapor mixing ratio are set to 0.0003 $s^{-1}$ above PBL (Sekiguchi et al., 2018). Nguyen et al., (2019a) adopted the nudging coefficients of u/v components and water vapor mixing ratio for 0.0001 $s^{-1}$ in all layers, and nudging coefficients of temperature is set to 0.0001 $s^{-1}$ above PBL. Another study only considered the nudging coefficients of u/v components in all vertical layers with 0.0001 $s^{-1}$ and 0.0003 $s^{-1}$ for domains D01 and D02, respectively (Nguyen et al., 2019b). The FDDA nudging technology was applied to better represent the realistic atmosphere.

In future, we will choose several 5-day heavy pollution episodes in our long-run simulations and conduct sensitive simulations by turning off FDDA. We further evaluate and quantify the difference of impacts of FDDA on aerosol feedbacks among different coupled models in another research paper.

To make this point clear to readers, we have added relevant information in the Section 2.1 and conclusion section as follows:

Lines 137-142: *"Turing on FDDA in two-way coupled models could dampen the simulated aerosol feedbacks (Wong et al., 2012; Forkel et al., 2012; Hogrefe et al., 2015; Zhang et al., 2016). To reduce the effects of enabling FDDA on aerosol feedbacks in long-term simulations, here the nudging coefficients for u/v wind, temperature, and water vapor mixing ratio above the planetary boundary layer were set to 0.0001 $s^{-1}$, 0.0001 $s^{-1}$, and 0.00001 $s^{-1}$, respectively."*

Lines 759-762: "*In addition, FDDA nudging technique can attenuate the ARI effects during severe air polluted episodes, and optimal nudging coefficients among different regions need to be determined.*"

2. You are using different physics and chemistry routines in all models. This makes this an "apples to oranges" comparison.

Response: As we have not clearly described the selection principles of physics and chemistry routines in the methodology part, which made the reviewer and readers to have the sense of "apples to oranges" comparison. To solve it, more explanations were added in Lines 228-230 of the revised manuscript as follows:

"*To compare simulations by three coupled models, the respective model configurations of physics and chemistry routines are set as consistent as possible.*"

In the result part, we further rewrote related subtitles and sentences on multi-model evaluation results, as follows:

Line 239: "Meteorological evaluations and intercomparisons" was revised as "Multi-model meteorological evaluations".

Lines 257-259: "The overall model performances of WRF-CMAQ and WRF-Chem were better than that of WRF-CHIMERE, while all simulated results were overestimated at both annual and seasonal scales (MBs in spring and summer were larger than those in autumn and winter)." was revised as "All simulated results were overestimated at both annual and seasonal scales (MBs in spring and summer were larger than those in autumn and winter)."

Line 516: "Air quality evaluations and intercomparisons" was revised as "Multi-model air quality evaluations".

Line 559-560: "For $O_3$, WRF-CHIMERE (R = 0.62) exhibited the best model performance, followed by WRF-CMAQ (R = 0.55), and WRF-Chem (R = 0.45) (Table 4 and Figure S15)." was revised as "For $O_3$, WRF-CHIMERE (R = 0.62) exhibited the

highest correlation, followed by WRF-CMAQ (R = 0.55), and WRF-Chem (R = 0.45) (Table 4 and Fig. S16).”

Line 647-649: “The simulation accuracies of $NO_2$ columns via WRF-CHIMERE were significantly better than those using WRF-CMAQ or WRF-Chem in all seasons except for winter (Figure S20)” was revised as “The seasonal $NO_2$ columns were generally underestimated in WRF-CMAQ (-0.68 to -0.16 DU), WRF-Chem (-1.40 to -0.44 DU), WRF-CHIMERE (-1.31 to -0.19 DU) (Fig. S22).”.

3. Do you know what happens in Morrison microphysics when the mix-activation routine is not called (no chemistry)?

Response: Whether aerosol feedbacks are enabled in WRF-CMAQ and WRF-Chem or not, Morrison microphysics scheme calculate the number concentrations and mixing ratios of five hydrometeor species (cloud water, rain, ice, snow and graupel) including 26 microphysical processes as listed in Table R1 (Morrison et al., 2008). If the mix-activation routine is not called, cloud water mixing ratio is predicted and cloud droplet number concentration (CDNC) is prescribed as the constant value of 250 $cm^{-3}$ (Yang et al., 2011). Then, cloud water and constant cloud droplet effective radius from Morrison scheme are used to drive RRTMG shortwave and longwave radiation schemes in coupled models.

With considering aerosol-cloud interactions, prognostic CDNC were online calculated in Morrison microphysics scheme based on Köhler theory. CDNC further alter cloud droplet effective radius ($r_c$), which is calculated in the RRTMG shortwave and longwave radiation schemes as follows.

$$r_c = \left(\frac{3L_c}{4\pi\rho_w N_c}\right)^{1/3} \tag{1}$$

where $\rho_w$ denotes the cloud liquid water density, $L_c$ and $N_c$ are the cloud water mixing ratio and cloud droplet number concentration, respectively.

Once the $r_c$ changes, the corresponding cloud optical parameters (cloud extinction coefficient ($\beta_c$), single scattering albedo ($\omega_c$), and asymmetry factor ($g_c$)) also vary, and the empirical formulas are expressed as:

$$\beta_c = L_c(a_1 r_c^{b_1} + c_1) \tag{2}$$
$$\omega_c = 1-(a_2 r_c^{b_2} + c_2) \tag{3}$$
$$g_c = a_3 r_c^{b_3} + c_3 \tag{4}$$

where $L_c$ stands for the cloud water mixing ratio, and $a_i$, $b_i$ and $c_i$ are the functions of wavelength (Hu and Stamnes, 1993).

Regarding the precipitation, the cloud-to-rain autoconversion rate (P) are calculated in the Morrison cloud microphysics scheme (Khairoutdinov and Kogan, 2000):

$$P=1350 N_c^{-1.79} L_c^{2.47} \tag{5}$$

For the Thompson microphysics scheme in WRF-CHIMERE, the CDNC is set as a constant value of 300 cm$^{-3}$ without considering aerosol feedbacks or only enabling ARI. With only enabling ACI or both ARI and ACI, the method for calculating prognostic CDNC in the Thompson scheme is the same as that in the Morrison scheme, but the discrepancies of representations for cloud droplet effective radius (Eqs. 1 and 6) and cloud-to-rain autoconversion rate (Eqs. 5 and 8) exist in these two schemes.

$$r_c = \left( \frac{3 + (\frac{1000}{N_c} + 2)}{2\lambda_c} \right) \tag{6}$$

$$\lambda_c = \begin{cases} \left( \frac{N_c \pi \times 1000 \times 4896}{6L_c} \right)^{1/3} & \text{for } N_c \leq 100, \\[2ex] \left( \frac{N_c \pi \times 1000 \times 60}{6L_c} \right)^{1/3} & \text{for } N_c \geq 10^{10}, \\[2ex] \left( \frac{N_c \pi \times 1000 \times g\_ratio[\min(15, (\frac{1000}{N_c} + 2))]}{6L_c} \right)^{1/3} & \text{otherwise.} \end{cases} \tag{7}$$

where g_ratio is an array with values of 24, 60, 120, 210, 336, 504, 720, 990, 1320, 1716, 2184, 2730, 3360, 4080 and 4896.

Compared to the Morrison scheme, Thompson scheme has the capability to calculate the number concentration ($N_i$) of ice nucleating particles (INP), and detailed information is presented in Table S6 of Supplement of the revised manuscript. Similar to cloud droplet, $N_i$ also has the impacts on cloud ice effective radius and further their optical properties ($\beta_i$, $\omega_i$ and $g_i$) as follows.

$$r_i = \left( \frac{3}{2\lambda_i} \right) \tag{8}$$

$$\lambda_i = \left( \frac{N_i \pi \times 890 \times \Gamma(4)}{6L_i \times \Gamma(1)} \right)^{1/3} \tag{9}$$

$$\beta_i = I_c \left( a_0 + \frac{a_1}{r_i} \right) \tag{10}$$

$$\omega_i = 1 - (b_0 + b_1 r_e + b_2 r_e^2 + b_3 r_e^3) \tag{11}$$

$$g_i = c_0 + c_1 r_e + c_2 r_e^2 + c_3 r_e^3 \tag{12}$$

where $\beta_i$, $\omega_i$ and $g_i$ are the ice extinction coefficient, single scattering albedo and asymmetry factor, respectively. $I_c$ stands for the ice water content, $a_i$, $b_i$ and $c_i$ are the coefficient functions of wavelengths, and their detailed information are provided in the Tables 3a-d of Fu (1996).

The physical formulation for calculating cloud-to-rain autoconversion rate is adopted from Berry and Reinhardt (1974):

$$P = \frac{0.027 \rho L_c (\frac{1}{16} \times 10^{20} D_b^3 D_f - 0.4)}{\frac{3.72}{\rho L_c} (\frac{1}{2} \times 10^6 D_b - 7.5)^{-1}} \tag{13}$$

$$D_f = \left( \frac{6\rho L_c}{\pi \rho_w N_c} \right)^{1/3} \tag{14}$$

$$D_g = \frac{[\frac{\Gamma(\mu_c+7)}{\Gamma(\mu_c+4)}]^{1/3}}{\lambda_c} \tag{15}$$

$$D_b = (D_f^3 D_g^3 - D_f^6)^{1/6} \tag{16}$$

where $\rho$ is the moist air density, $\rho_w$ is the cloud liquid water density, $D_b$, $D_g$ and $D_f$ are the characteristic diameters, $\mu_c$ and $\lambda_c$ are the shape and slope parameters of gramma size distribution ($\Gamma$), respectively.

Table R1 Summary of microphysical process in the Morrison scheme

| No. | Process name |
| --- | --- |
| 1 | Ice nucleation from freezing of aerosol |
| 2 | Droplet activation from aerosol |
| 3 | Ice multiplication |
| 4 | Autoconversion of droplets to from rain |
| 5 | Autoconversion of ice to form snow |
| 6 | Accretion of droplets by rain |
| 7 | Accretion of droplets by snow |
| 8 | Accretion of rain by snow |
| 9 | Accretion of cloud ice by snow |
| 10 | Heterogeneous freezing of droplets to form cloud ice |
| 11 | Heterogeneous freezing of rain to form snow |
| 12 | Melting of snow to form rain |
| 13 | Melting of cloud ice to form droplets |
| 14 | Self-collection of droplets |
| 15 | Self-collection of cloud ice |
| 16 | Self-collection of snow |
| 17 | Self-collection of rain |
| 18 | Loss of N due to sublimation of cloud ice |
| 19 | Loss of N due to evaporation of rain |
| 20 | Loss of N due to sublimation of snow |
| 21 | Deposition/sublimation of cloud ice |
| 22 | Condensation/evaporation of droplets |
| 23 | Condensation/evaporation of rain |
| 24 | Deposition/sublimation of snow |
| 25 | Homogeneous freezing of droplets to form cloud ice |
| 26 | Homogeneous freezing of rain to form snow |

4. The authors need to be clear on what is used by Radiation (R) and MicroPhysics (MP) if feedback is off versus on.

Response: In the two-way coupled models, for the radiation calculation processes, numerous combinations of radiation and microphysics schemes are presented, and our selections in this study are presented in Figure S26.

When aerosol feedback is turned on, for radiation, the spectral optical properties of different aerosol species groups and heating rates (ttenld) are online calculated, and then inter/extrapolated into specific shortwave (SW) and longwave (LW) bands in the RRTMG SW/LW schemes (Tables S5–S6). For microphysics, when the ACI feedback is turned on, the prognostic cloud droplet number concentration (CDNC) or/and ice nucleating particle concentration (INP), optical properties of cloud liquid water and ice is taken into account in the Morrison scheme of WRF-Chem and Thompson scheme of WRF-CHIMERE (Table S7).

When aerosol feedback is turned off, for radiation, the option of aer_opt is set to 0 (no aerosols) as baseline scenario in our study, which result in no calculations of aerosol optical properties in RRTMG, as shown in Table S6. Although aer_opt=1 or 2 can be set when feedback is off (the climatology data or empirical formulas of aerosol optical properties were utilized to calculate aerosol radiative effects), the corresponding simulated results can not be used as baseline scenario to quantify the ARI effects in our study. For microphysics, prescribed CDNC are set to 250 and 300 cm$^{-3}$ in the Morrison scheme of WRF-Chem and Thompson scheme of WRF-CHIMERE WRF-Chem and WRF-CHIMERE, respectively. The prescribed INP are calculated by the empirical formula of temperature in these three coupled models (Rasmussen et al., 2002; DeMott et al., 2015; Thompson and Eidhammer, 2014).

To reflect the above information, we compiled the Tables S6-S8 and plotted the Figure S24 and put them into the Supplement of the revised manuscript. We also added the sentences in the revised manuscript as follows.

Lines 408-422: "*When ARI effects are enabled, the diversities of refractive indices of aerosol species groups lead to the discrepancies of online calculated aerosol optical properties in different shortwave and longwave (SW and LW) bands in the RRTMG SW/LW radiation schemes of WRF-CMAQ, WRF-Chem, and WRF-CHIMERE (Tables S5–S6). The online calculated cloud optical properties induced by aerosol absorption in the RRTMG radiation schemes are different in treatments of aerosol species groups in the three coupled models. With enabling ACI effects, the activation of cloud droplets from aerosols based on the Köhler theory is taken into account in WRF-Chem and WRF-CHIMERE, in comparison to simulations without aerosol feedbacks (Table S7). The treatments of prognostic ice nucleating particles (INP) formed via heterogeneous nucleation of dust particles (diameters > 0.5 µm) and homogeneous freezing of hygroscopic aerosols (diameters > 0.1 µm) are only considered in WRF-CHIMERE, but the prognostic ice nucleating particles are not included in WRF-CMAQ and WRF-Chem. These discrepancies eventually contribute to the differences of simulated radiation changes caused by aerosols.*"

[Figure]

*Figure S26. Summary of the selected options of radiation and microphysics schemes in coupled WRF-CMAQ, WRF-Chem and WRF-CHIMERE in this study.*

*Table S5. Radiation variables used in the two-way coupled WRF-CMAQ, WRF-Chem and WRF-CHIMERE models with only enabling ARI compared to without aerosol feedbacks.*

| Model | SW/LW radiation schemes | Turning off feedback | Turning on ARI feedback | |
|---|---|---|---|---|
| | | | Direct effects | Semi-direct effects |
| WRF-CMAQ | RRTMG/RRTMG | Aerosol optical properties are not calculated | Aerosol extinction, single scattering albedo ($\omega_o$), and asymmetry factor (g) 14 shortwave bands and 5 longwave bands (Wong et al., 2012) | 1. Solar uv and ir fluxes 2. Radiative heating rate for the tten1d variable |
| WRF-Chem | RRTMG/RRTMG | Aerosol optical properties are not calculated | $\omega_o$ (300 nm, 400 nm, 600 nm, 999 nm), g (300 nm, 400 nm, 600 nm, 999 nm), AOD ($\tau$) (300 nm, 400 nm, 600 nm, 999 nm, 16 bands 3400 nm to 55600 nm) (Zhao et al., 2011) | 1. Solar uv and ir fluxes 2. Radiative heating rate for the tten1d variable |
| WRF-CHIMERE | RRTMG/RRTMG | Aerosol optical properties are not calculated | $\omega_o$ (400 nm, 600 nm), g (400 nm, 600 nm), AOD (300 nm, 400 nm, 999 nm, 16 bands 3400 nm to 55600 nm) (Briant et al., 2017) | 1. Solar uv and ir fluxes 2. Radiative heating rate for the tten1d variable |

*Table S6. Description of refractive indices and radiation schemes used in the WRF-CMAQ, WRF-Chem and WRF-CHIMERE models.*

| Model | Refractive indices of aerosol species groups | |
|---|---|---|
| | SW | LW |
| WRF-CMAQ | 1. Water (1.408+1.420×10$^{-2}$i, 1.324+1.577×10$^{-1}$i, 1.277+1.516×10$^{-3}$i, 1.302+1.159×10$^{-3}$i, 1.312+2.360×10$^{-4}$i, 1.321+1.713×10$^{-4}$i, 1.323+2.425×10$^{-5}$i, 1.327+3.125×10$^{-6}$i, 1.331+3.405×10$^{-8}$i, 1.334+1.639×10$^{-9}$i, 1.340+2.955×10$^{-9}$i, 1.349+1.635×10$^{-8}$i, 1.362+3.350×10$^{-8}$i, 1.260+6.220×10$^{-2}$i)  2. Water-soluble (1.443+5.718×10$^{-3}$i, 1.420+1.777×10$^{-2}$i, 1.420+1.060×10$^{-2}$i, 1.420+8.368×10$^{-3}$i, 1.463+1.621×10$^{-2}$i, 1.510+2.198×10$^{-2}$i, 1.510+1.929×10$^{-2}$i, 1.520+1.564×10$^{-2}$i, 1.530+7.000×10$^{-3}$i, 1.530+5.666×10$^{-3}$i, 1.530+5.000×10$^{-3}$i, 1.530+8.440×10$^{-3}$i, 1.530+3.000×10$^{-2}$i, 1.710+1.100×10$^{-1}$i)  3. BC (2.089+1.070i, 2.014+0.939i, 1.962+0.843i, 1.950+0.784i, 1.940+0.760i, 1.930+0.749i, 1.905+0.737i, 1.870+0.726i, 1.850+0.710i, 1.850+0.710i, 1.850+0.710i, 1.850+0.710i, 1.850+0.710i, 2.589+1.771i)  4. Insoluble (1.272+1.165×10$^{-2}$i, 1.168+1.073×10$^{-2}$i, 1.208+8.650×10$^{-3}$i, 1.253+8.092×10$^{-3}$i, 1.329+8.000×10$^{-3}$i, 1.418+8.000×10$^{-3}$i, 1.456+8.000×10$^{-3}$i, 1.518+8.000×10$^{-3}$i, 1.530+8.000×10$^{-3}$i, 1.530+8.000×10$^{-3}$i, 1.530+8.000×10$^{-3}$i, 1.530+8.440×10$^{-3}$i, 1.530+3.000×10$^{-2}$i, 1.470+9.000×10$^{-3}$i)  5. Sea-salt (1.480+1.758×10$^{-3}$i, 1.534+7.462×10$^{-3}$i, 1.437+2.950×10$^{-3}$i, 1.448+1.276×10$^{-3}$i, 1.450+7.944×10$^{-4}$i, 1.462+5.382×10$^{-4}$i, 1.469+3.754×10$^{-4}$i, 1.470+1.498×10$^{-4}$i, 1.490+2.050×10$^{-7}$i, 1.500+1.184×10$^{-8}$i, 1.502+9.938×10$^{-8}$i, 1.510+2.060×10$^{-6}$i, 1.510+5.000×10$^{-6}$i, 1.510+1.000×10$^{-2}$i) in term of 14 wavelengths at 3.4615, 2.7885, 2.325, 2.046, 1.784, 1.4625, 1.2705, 1.0101, 0.7016, 0.53325, 0.38815, 0.299, 0.2316, 8.24 $\mu$m | 1. Water (1.160+0.321i, 1.140+0.117i, 1.232+0.047i, 1.266+0.038i, 1.300+0.034i)  2. Water-soluble (1.570+0.069i, 1.700+0.055i, 1.890+0.128i, 2.233+0.334i, 1.220+0.066i)  3. BC (1.570+2.200i, 1.700+2.200i, 1.890+2.200i, 2.233+2.200i, 1.220+2.200i)  4. Insoluble (1.482+0.096i, 1.600+0.107i, 1.739+0.162i, 1.508+0.117i, 1.175+0.042i)  5. Sea-salt (1.410+0.019i, 1.490+0.014i, 1.560+0.017i, 1.600+0.029i, 1.402+0.012i) in term of 5 thermal windows at 13.240, 11.20, 9.73, 8.870, 7.830 $\mu$m |
| WRF-Chem | 1. Water (1.35+1.524×10$^{-8}$i, 1.34+2.494×10$^{-9}$i, 1.33+1.638×10$^{-9}$i, 1.33+3.128×10$^{-6}$i)  2. Dust (1.55+0.003i, 1.550+0.003i, 1.550+0.003i, 1.550+0.003i)  3. BC (1.95+0.79i, 1.95+0.79i, 1.95+0.79i, 1.95+0.79i)  4. OC (1.45+0i, 1.45+0i, 1.45+0i, 1.45+0i)  5. Sea salt (1.51+8.66×10$^{-7}$i, 1.5+7.019×10$^{-8}$i, 1.5+1.184×10$^{-5}$i, 1.47+1.5×10$^{-4}$i)  6. Sulfate (1.52+1.00×10$^{-8}$i, 1.52+1.00×10$^{-9}$i, 1.52+1.00×10$^{-9}$i, 1.52+1.75×10$^{-4}$i) in term of 4 spectral intervals in 0.25-0.35, 0.35-0.45, 0.55-0.65, 0.998-1.000 $\mu$m | 1. Water (1.532+0.336i, 1.524+0.360i, 1.420+0.426i, 1.274+0.403i, 1.161+0.321i, 1.142+0.115i, 1.232+0.047i, 1.266+0.039i, 1.296+0.034i, 1.321+0.0344i, 1.342+0.092i, 1.315+0.012i, 1.330+0.013i, 1.339+0.01i, 1.350+0.0049i, 1.408+0.0142i)  2. Dust (2.34+0.7i, 2.904+0.857i, 1.748+0.462i, 1.508+0.263i, 1.91+0.319i, 1.822+0.26i, 2.917+0.65i, 1.557+0.373i, 1.242+0.093i, 1.447+0.105i, 1.432+0.061i, 1.473+0.0245i, 1.495+0.011i, 1.5+0.008i)  3. BC (1.95+0.79i, 1.95+0.79i, 1.95+0.79i, 1.95+0.79i, 1.95+0.79i, 1.95+0.79i, 1.95+0.79i, 1.95+0.79i, 1.95+0.79i, 1.95+0.79i, 1.95+0.79i, 1.95+0.79i, 1.95+0.79i,)  4. OC (1.86+0.5i, 1.91+0.268i, 1.988+0.185i, 1.439+0.198i, 1.606+0.059i, 1.7+0.0488i, 1.888+0.11i, 2.489+0.3345i, 1.219+0.065i, 1.419+0.058i, 1.426+0.0261i, 1.446+0.0142i, 1.457+0.013i, 1.458+0.01i)  5. Sea salt (1.74+0.1978i, 1.76+0.1978i, 1.78+0.129i, 1.456+0.038i, 1.41+0.019i, 1.48+0.014i, 1.56+0.016i, 1.63+0.03i, 1.4+0.012i, 1.43+0.0064i, 1.56+0.0196i, 1.45+0.0029i, 1.485+0.0017i, 1.486+0.0014i)  6. Sulfate (1.89+0.22i, 1.91+0.152i, 1.93+0.0846i, 1.586+0.2225i, 1.678+0.195i, 1.758+0.441i, 1.855+0.696i, 1.597+0.695i, 1.15+0.459i, 1.26+0.161i, 1.42+0.172i, 1.35+0.14i, 1.379+0.12i, 1.385+0.122i) in term of 16 spectral intervals in 10-350, 350-500, 500-630, 630-700, 700-820, |

[Figure]

WRF-CHIMERE
1. Water (1.35+2.0×10⁻⁹i, 1.34+2.0×10⁻⁹i, 1.34+1.8×10⁻⁸i, 1.33+3.4×10⁻⁸i, 1.33+3.9×10⁻⁷i)
2. Dust (1.53+0.0055i, 1.53+0.0055i, 1.53+0.0024i, 1.53+8.9-4i, 1.53+7.6-4i)
3. BC (1.95+0.79i, 1.95+0.79i, 1.95+0.79i, 1.95+0.79i, 1.95+0.79i)
4. OC (1.53+0.09i, 1.53+0.008i, 1.53+0.005i, 1.53+0.0063i, 1.52+0.016i)
5. Sea salt (1.38+8.7×10⁻⁷i, 1.38+3.5×10⁻⁷i, 1.37+6.6×10⁻⁸i, 1.36+1.2×10⁻⁸i, 1.35+2.6×10⁻⁵i)
6. PPM (1.53+0.008i, 1.52+0.008i, 1.52+0.008i, 1.51+0.008i, 1.5+0.008i)
7. SOA (1.56+0.003i, 1.56+0.003i, 1.56+0.003i, 1.56+0.003i, 1.56+0.003i)
8. H₂SO₄ (1.5+1.0×10⁻⁸i, 1.47+1.0×10⁻⁸i, 1.44+1.0×10⁻⁸i, 1.43+1.3×10⁻⁸i, 1.42+1.2×10⁻⁶i)
9. HNO₃ (1.53+0.006i, 1.53+0.006i, 1.53+0.006i, 1.53+0.006i, 1.53+0.006i)
10. NH₃ (1.53+0.0005i, 1.52+0.0005i, 1.52+0.0005i, 1.52+0.0005i, 1.52+0.0005i) in term of 5 wavelengths at 0.2, 0.3, 0.4, 0.6, 0.999 μm

820-980, 980-1080, 1080-1180, 1180-1390, 1390-1480, 1480-1800, 1800-2080, 2080-2250, 2250-2390, 2390-2600, 2600-3250 cm⁻¹
1. Water (1.42+0.02i, 1.35+0.0047i, 1.34+0.0085i, 1.33+0.015i, 1.32+0.01i, 1.32+0.13i, 1.32+0.032i, 1.3+0.034i, 1.27+0.039i, 1.23+0.047i, 1.15+0.1i, 1.16+0.32i, 1.27+0.4i, 1.41+0.43i, 1.52+0.37i, 1.65+0.55i)
2. BC (1.95+0.79i, 1.95+0.79i, 1.95+0.79i, 1.95+0.79i, 1.95+0.79i, 1.95+0.79i, 1.95+0.79i, 1.95+0.79i, 1.95+0.79i, 1.95+0.79i, 1.95+0.79i, 1.95+0.79i, 1.95+0.79i, 1.95+0.79i, 1.95+0.79i, 1.95+0.79i)
3. OC (1.43+1.42i, 1.46+1.43i, 1.46+1.25i, 1.46+2.67i, 1.45+1.89i, 1.42+1.71i, 1.43+1.71i, 1.25+0.007i, 2.67+0.005i, 1.89+0.01i, 1.71+0.013i, 1.43+0.014i, 1.46+0.025i, 1.46+0.062i, 1.46+0.064i, 1.45+0.031i)
4. Salt (1.43+0.019i, 1.37+0.0043i, 1.36+0.0084i, 1.36+0.011i, 1.34+0.01i, 1.35+0.083i, 1.34+0.029i, 1.31+0.03i, 1.33+0.037i, 1.29+0.042i, 1.2+0.09i, 1.2+0.27i, 1.3+0.34i, 1.47+0.37i, 1.56+0.03i, 1.51+0.09i)
5. PPM (1.45+0.01i, 1.45+0.01i, 1.45+0.01i, 1.45+0.01i, 1.45+0.01i, 1.45+0.01i, 1.45+0.05i, 1+0.5i, 1+0.2i, 2.6+0.2i, 1.7+0.2i, 1.7+0.2i, 1.7+0.2i, 1.7+0.2i, 1.7+0.2i)
6. SOA (1.56+0.003i, 1.56+0.003i, 1.56+0.003i, 1.56+0.003i, 1.56+0.003i, 1.56+0.003i, 1.56+0.003i, 1.56+0.003i, 1.56+0.003i, 1.56+0.003i, 1.56+0.003i, 1.56+0.003i, 1.56+0.003i, 1.56+0.003i, 1.56+0.003i)
7. H₂SO₄ (1.35+0.16i, 1.4+0.13i, 1.39+0.12i, 1.38+0.12i, 1.35+0.15i, 1.42+0.18i, 1.26+0.16i, 1.15+0.44i, 1.57+0.73i, 1.83+0.7i, 1.71+0.46i, 1.68+0.2i, 1.59+0.21i, 1.87+0.48i, 1.89+0.27i, 1.86+0.31i)
8. HNO₃ (1.45+0.01i, 1.45+0.01i, 1.45+0.01i, 1.45+0.01i, 1.45+0.01i, 1.45+0.01i, 1.45+0.05i, 1+0.5i, 1+0.2i, 2.6+0.2i, 1.7+0.2i, 1.7+0.2i, 1.7+0.2i, 1.7+0.2i, 1.7+0.2i)
9. NH₃ (1.45+0.01i, 1.45+0.01i, 1.45+0.01i, 1.45+0.01i, 1.45+0.01i, 1.45+0.01i, 1.45+0.05i, 1+0.5i, 1+0.2i, 2.6+0.2i, 1.7+0.2i, 1.7+0.2i, 1.7+0.2i, 1.7+0.2i, 1.7+0.2i) in term of 16 wavelengths at 3.4, 4, 4.3, 4.6, 5.2, 6.1, 7.0, 7.8, 8.8, 9.7, 11.1, 13.2, 15.0, 17.7, 23.5, 55.6 μm

*Table S7. Microphysics variables used in the two-way coupled WRF-CMAQ, WRF-Chem and WRF-CHIMERE models with enabling ACI effects compared to without aerosol feedbacks.*

| Model | Microphysics scheme | Turning off feedback | Turning on ACI feedback | |
| --- | --- | --- | --- | --- |
| | | | *First indirect effects* | *Second indirect effects* |
| WRF-CMAQ | Morrison | 1. Prescribed constant CDNC value of 250 cm⁻³ | None | None |
| WRF-Chem | Morrison | 1. Prescribed constant CDNC value of 250 cm⁻³
2. Constant cloud droplet effective radius with 10 μm
3. Cloud droplet extinction coefficient, single scattering albedo, and asymmetry factor based on Eqs. (2)-(4)
4. Prescribed ice nucleating particle (INP) concentration based on empirical formula (Rasmussen et al., 2002) | 1. Hygroscopicity
2. Prognostic CDNC based on Köhler theory
3. Prognostic cloud droplet effective radius
4. Prognostic cloud droplet extinction coefficient, single scattering albedo, and asymmetry factor
5. Prescribed INP | 1. Prognostic cloud-to-rain autoconversion rate |
| WRF-CHIMERE | Thompson | 1. Prescribed constant CDNC values of 300 cm⁻³
2. Prescribed INP from heterogeneous ice nucleation using climatical dust concentration (dimeters > 0.5μm) (DeMott et al., 2015) and homogeneous freezing (Thompson and Eidhammer, 2014) with climatological hygroscopic aerosol concentrations (dimeters > 0.1μm) generated by QNWFA_QNIFA_Monthly_GFS file
3. Prescribed cloud droplet and ice effective radius
4. Prescribed extinction coefficient, single scattering albedo, and asymmetry factor of cloud droplet and ice | 1. Hygroscopicity
2. Prognostic CDNC based on Köhler theory
3. Prognostic INP from heterogeneous ice nucleation based on online dust calculation (dimeters > 0.5 μm) and homogeneous freezing with prognostic hygroscopic aerosol concentrations (dimeters > 0.1μm) (Tuccella et al., 2019)
4. Prognostic cloud droplet and ice effective radius
5. Prognostic extinction coefficient, single scattering albedo, and asymmetry factor of cloud droplet and ice | 1. Prognostic cloud-to-rain autoconversion rate |

5. Are you just using a constant droplet number? A climatology?

Response: It depends on the situation. Without turning on aerosol feedbacks or only considering ARI, cloud droplet number concentration (CDNC) is prescribed as a constant value of 250 cm⁻³ in the Morrison scheme of WRF-CMAQ and WRF-Chem and 300 cm⁻³ in the Thompson scheme of WRF-CHIMERE. With enabling ACI or both ARI and ACI, the online calculation of (CDNC) is performed in WRF-Chem and WRF-CHIMERE by utilizing the aerosol activation parameterization (Abdul-Razzak and Ghan, 2002; Chapman et al., 2009; Tuccella et al., 2019).

The above information is added in Lines 486-496 of the revised manuscript as follows:

*"This may be explained as the different parameterization treatments of cloud droplet number concentration (CDNC) simulated by the three coupled models with/without enabling ACI effects. The cloud condensation nuclei activated from aerosol particles can increase CDNC and impact on LWP and CF. Without enabling any aerosol feedbacks or only enabling ARI, the CDNC is default prescribed as a constant value of 250 cm$^{-3}$ in the Morrison scheme of WRF-CMAQ and WRF-Chem and 300 cm$^{-3}$ in the Thompson scheme of WRF-CHIMERE. When only ACI or both ARI and ACI are enabled, the online calculating of prognostic CDNC is performed in WRF-Chem and WRF-CHIMERE by using the method of maximum supersaturation (Abdul-Razzak and Ghan, 2002; Chapman et al., 2009; Tuccella et al., 2019)."*

6. WRF-Chem has a lot of options, how come you decided to use different physics than in WRF-CMAQ?

Response: The options of commonly used physics schemes for the two-way coupled WRF-CMAQ and WRF-Chem models are summarized in the Table S1 of Gao et al. (2022). To keep the consistency of physical schemes, the same RRTMG shortwave and longwave radiation schemes and the Morrison microphysics scheme were adopted both in WRF-Chem and WRF-CMAQ. It should be noted that the microphysics processes only support the Thompson scheme in coupled WRF-CHIMERE. As possible as we can, the different cumulus, surface, and land surface schemes in WRF-CMAQ and WRF-Chem were chosen according to widely used options outlined in Table S1 of Gao et al. (2022). Related information is added in Section 2.1 of the revised manuscript as follows.

*"To keep the consistency of physical schemes, the same RRTMG shortwave and longwave radiation schemes and Morrison microphysics schemes are adopted in both WRF-Chem and WRF-CMAQ. WRF-CHIMERE applied the same radiation schemes and Thompson microphysics scheme. The different other schemes (cumulus, surface, and land surface) in WRF-CMAQ and WRF-Chem were chosen according to widely used options outlined in Table S1 of Gao et al. (2022). The other schemes used in WRF-CHIMERE are the same as with WRF-Chem."*

Reference:

Gao C, Xiu A, Zhang X, et al. Two-way coupled meteorology and air quality models in Asia: a systematic review and meta-analysis of impacts of aerosol feedbacks on meteorology and air quality[J]. Atmospheric Chemistry and Physics, 2022, 22(8): 5265-5329.

7. Chimere is way behind in the WRF version used, which makes that one even harder to compare?

Response: Yes, it really makes the comparison to be harder. Even under the same setting of scientific options in namelist, different versions of WRF have notable impacts on the simulated results (Appel et al., 2021). Until now, the coupled WRF-CHIMERE only support the version 3.7.1 of WRF (Briant et al., 2017; Tuccella et al., 2019). In order to include this new developed coupled model into our inter-comparisons, we have to accept the lower version of WRF. To clarify this discrepancy, we added a new sentence in Lines 116-118 of the methodology part of the revised manuscript as follows.

*"Compared to WRF v4.1.1-CMAQ v5.3.1 and WRF-Chem v4.1.1, the coupled WRF-CHIMERE only support the version 3.7.1 of WRF (Briant et al., 2017; Tuccella et al., 2019)."*

8. I am not asking you to rerun this monster simulation, but you will need to rephrase some of your abstract, conclusion, and results description.

Response: Thanks for your suggestions, we have rewritten and modified some parts of the abstract, results and conclusion, and relevant revisions are as follows:

*Abstract:*

*In the eastern China region, two-way coupled meteorology and air quality models have been applied aiming to more realistically simulate meteorology and air quality by accounting for the aerosol–radiation–cloud interactions. There have been numerous related studies being conducted, but the performances of multiple two-way coupled models simulating meteorology and air quality have not been compared in this region. In this study, we systematically evaluated annual and seasonal meteorological and air quality variables simulated by three open-source and widely used two-way coupled models (i.e., WRF-CMAQ, WRF-Chem, and WRF-CHIMERE) by validating the model results with surface and satellite observations for eastern China during 2017. Note that although we have done our best to keep the same configurations, this study is not aiming to screen which model is better or worse since different setups are still presented in simulations. Our evaluation results showed that all three two-way coupled models reasonably well simulated the annual spatiotemporal distributions of meteorological and air quality variables. The impacts of aerosol-cloud interaction (ACI) on model performances' improvements were limited compared to aerosol-radiation interaction (ARI), and several possible improvements on ACI representations in two-way coupled models are further discussed and proposed. When sufficient computational resources become available, two-way coupled models should be applied for more accurate air quality forecast and timely warning of heavy air pollution events in atmospheric environmental management. The potential improvements of two-way coupled models are proposed in future research perspectives.*

*Conclusions:*

*Applications of two-way coupled meteorology and air quality models have been performed in eastern China in recent years, but no research focused on the comprehensive assessments of multiple coupled models in this region. To the best of our knowledge, this is the first time to conduct comprehensive inter-comparisons among the open-sourced two-way coupled meteorology and air quality models (WRF-CMAQ, WRF-Chem, and WRF-CHIMERE). This study systemically evaluated the hindcast simulations for 2017 and explored the impacts of ARI and/or ACI on model and computational performances in eastern China.*

*After detailed comparisons with ground-based and satellite-borne observations, the evaluation results showed that three coupled models perform well for meteorology and air quality, especially for surface temperature (with R up to 0.97) and $PM_{2.5}$ concentrations (with R up to 0.68). The effects of aerosol feedbacks on model performances varied depending on the two-way coupled models, variables, and time scales. There were around 20%–70% increase of computational time when these two-way coupled models enabled aerosol feedbacks against simulations without aerosol-radiation-cloud interactions. It is noteworthy that all three coupled models could well reproduce the spatiotemporal distributions of satellite-retrieved CO column concentrations but not for ground-observed CO concentrations.*

*With inter-comparisons, some uncertainty sources can be ascertained in evaluating aerosol feedback effects. As numerous schemes can be combined in configurations of different coupled models, here we only evaluated simulations with specific settings. Future comparison works with considering more combinations of multiple schemes within the same or different coupled models need to be conducted. Among the three coupled models, the numerical representations for specific variable in same scheme are diverse, e.g., treatments of cloud cover and cloud optical properties in the Fast-JX photolysis scheme. More accurate representations of photolysis processes should be taken into account to reduce the evaluation uncertainties. In addition, FDDA nudging technique can attenuate the ARI effects during severe air polluted episodes, and optimal nudging coefficients among different regions need to be determined. Last but not least, the actual mechanisms underlying ACI effects are still unclear, and the new advances in the measurements and parameterizations of CCN/IN activations and precipitation need to be timely incorporated in coupled models.*

Revisions of the result descriptions are as follows.

Lines 408-411: "*that the differences of aerosol representations contributed to the diversity of seasonal MBs. For example, 3 modes AERO6, 4 bins sectional MOSAIC and 10 bins SAM were implemented in WRF-CMAQ, WRF-Chem and WRF-CHIMERE, respectively (Table 2).*" is added.

Lines 415-429: "*The refractive indices of aerosol species groups show discrepancies for calculating aerosol optical depth or extinction coefficients, single scattering albedo and asymmetry factor in different shortwave and longwave (SW and LW) bands in RRTMG SW/LW radiation schemes of WRF-CMAQ, WRF-Chem, and WRF-CHIMERE when ARI effects are enabled (Tables S4 and S5). Representations of cloud optical depth induced by influence of various aerosol absorption of sunlight for the same RRTMG radiation schemes have the different attributing to treatments of aerosol in the three coupled models. The activation of cloud droplets from aerosols based on the Köhler theory is taken into account in the WRF-Chem and WRF-CHIMERE models with enabled ACI effects, in comparison to simulations without aerosol feedbacks (Table S6). However, parametrizations for ice nucleating (INP) formed via heterogeneous nucleation of (diameters > 0.5 μm) and homogeneous freezing of hygroscopic aerosols (diameters > 0.1 μm) are only implemented in WRF-CHIMERE. The descriptions of all radiation and cloud microphysics variables used in the three coupled models with enabling ARI or ACI effects compared to without enabling aerosol feedbacks are presented in Figure S23. These discrepancies result in a variety of simulated radiation changes caused by aerosol.*" is added.

Lines 498-507: "*This may be caused by the different treatments of cloud droplet number concentration (CDNC) resulting from the enabling or disabling ACI effects in coupled models. The cloud condensation nuclei activated from aerosol can increase CDNC and affect LWP and CF. The CDNC is prescribed as a constant value of 250 $cm^{-3}$ in Morrison scheme for WRF-CMAQ and WRF-Chem or 300 $cm^{-3}$ in Thompson scheme for WRF-CHIMERE without enabling aerosol feedbacks or ARI. The prognostic CDNC calculation is performed in WRF-Chem and WRF-CHIMERE with enabling ACI or both ARI and ACI by utilizing the maximum supersaturation (Abdul-Razzak and Ghan, 2002; Chapman et al., 2009; Tuccella et al., 2019).*" is added.

Lines 583-594: "*Such diversity in NMB variation can be explained by two aspect differences. First, model-top boundary conditions, for the WRF-CMAQ model, the potential impacts of stratosphere-troposphere $O_3$ exchange are considered via the parameterization of $O_3$-potential vorticity correlations (Xing et al., 2016) and used in our study, but not been in WRF-Chem (Grell et al., 2005). For the WRF-CHIMERE, climatological data or concentrations of coarse simulation can be used to represent model-top boundary conditions, and we applied the climatology from a general circulation model developed at the Laboratoire de Météorologie Dynamique (LMDz) coupling a global chemistry and aerosol model INteractions between Chemistry and Aerosols (INCA) (Mailler et al., 2017). Secondly, for gas-phase chemistry mechanisms, three coupled models incorporate a variety of photolytic reactions, with a more comprehensive explanation provided in Section 4.2.*" is added.

Line 691-699: "*More detailed interpretations were grouped into four aspects: (1) AODs are calculated via Mie theory using refractive indices of different numbers (5, 6 and 10) of aerosol species group in different coupled models (WRF-CMAQ, WRF-Chem and WRF-CHIMERE) (Tables S4 and S5); (2) 7 (294.6, 303.2, 310.0, 316.4, 333.1, 382.0 and 607.7 nm), 4 (300, 400, 600 and 999 nm), and 5 (200, 300, 400, 600, and 999 nm) effective wavelengths are used in calculating actinic fluxes and photolysis rates in Fast-JX photolysis modules of WRF-CMAQ, WRF-Chem and WRF-CHIMERE, respectively;(3) Different calculating methods of aerosol and cloud optical properties exist in the Fast-JX schemes of three coupled models (Tables S4-S6); (4) 77, 52 and 40 gas-phase species involve 218, 132, 120 gas-phase reactions in CB6, CBMZ and MELCHIOR2 mechanisms, respectively.*" is added.

We exchanged sequence of Sections 2.1 and 2.2 in the revised manuscript to improve the readability. Also, the comparisons of seasonal simulations and satellite observations in Section 4.2 are merge a paragraph and interpretations of mode diversities regarding simulated column variables are rephase a paragraph.

The added references are as follows.

*Alapaty K, Herwehe J A, Otte T L, et al. Introducing subgrid-scale cloud feedbacks to radiation for regional meteorological and climate modeling[J]. Geophysical Research Letters, 2012, 39(24).*

*Archer-Nicholls S, Lowe D, Utembe S, et al. Gaseous chemistry and aerosol mechanism developments for version 3.5.1 of the online regional model, WRF-Chem[J]. Geoscientific Model Development, 2014, 7(6): 2557-2579.*

*Binkowski F S, Arunachalam S, Adelman Z, et al. Examining photolysis rates with a prototype online photolysis module in CMAQ[J]. Journal of Applied Meteorology and Climatology, 2007, 46(8): 1252-1256.*

*Binkowski F S, Roselle S J. Models-3 Community Multiscale Air Quality (CMAQ) model aerosol component 1. Model description[J]. Journal of geophysical research: Atmospheres, 2003, 108(D6).*

*Chapman E G, Gustafson Jr W I, Easter R C, et al. Coupling aerosol-cloud-radiative processes in the WRF-Chem model: Investigating the radiative impact of elevated point sources[J]. Atmospheric Chemistry and Physics, 2009, 9(3): 945-964.*

*Fu Q. An accurate parameterization of the solar radiative properties of cirrus clouds for climate models[J]. Journal of climate, 1996, 9(9): 2058-2082.*

*Grell G A, Freitas S R. A scale and aerosol aware stochastic convective parameterization for weather and air quality modeling[J]. Atmospheric Chemistry and Physics, 2014, 14(10): 5233-5250.*

*Heymsfield A J, Matrosov S, Baum B. Ice water path–optical depth relationships for cirrus and deep stratiform ice cloud layers[J]. Journal of Applied Meteorology and Climatology, 2003, 42(10): 1369-1390.*

*Hong S Y, Juang H M H, Zhao Q. Implementation of prognostic cloud scheme for a regional spectral model[J]. Monthly weather review, 1998, 126(10): 2621-2639.*

Hu Y X, Stamnes K. An accurate parameterization of the radiative properties of water clouds suitable for use in climate models[J]. Journal of climate, 1993, 6(4): 728-742.

Iacono M J, Delamere J S, Mlawer E J, et al. Radiative forcing by long‑lived greenhouse gases: Calculations with the AER radiative transfer models[J]. Journal of Geophysical Research: Atmospheres, 2008, 113(D13).

Mailler S, Menut L, Khvorostyanov D, et al. CHIMERE-2017: From urban to hemispheric chemistry-transport modeling[J]. Geoscientific Model Development, 2017, 10(6): 2397-2423.

Menut L, Bessagnet B, Khvorostyanov D, et al. CHIMERE 2013: a model for regional atmospheric composition modelling[J]. Geoscientific model development, 2013, 6(4): 981-1028.

Menut L, Siour G, Mailler S, et al. Observations and regional modeling of aerosol optical properties, speciation and size distribution over Northern Africa and western Europe[J]. Atmospheric Chemistry and Physics, 2016, 16(20): 12961-12982.

Mocko D M, Cotton W R. Evaluation of fractional cloudiness parameterizations for use in a mesoscale model[J]. Journal of Atmospheric Sciences, 1995, 52(16): 2884-2901.

Pincus R, Barker H W, Morcrette J J. A fast, flexible, approximate technique for computing radiative transfer in inhomogeneous cloud fields[J]. Journal of Geophysical Research: Atmospheres, 2003, 108(D13).

Safieddine S, Boynard A, Coheur P F, et al. Summertime tropospheric ozone assessment over the Mediterranean region using the thermal infrared IASI/MetOp sounder and the WRF-Chem model[J]. Atmospheric chemistry and physics, 2014, 14(18): 10119-10131.

Wild O, Zhu X I N, Prather M J. Fast-J: Accurate simulation of in-and below-cloud photolysis in tropospheric chemical models[J]. Journal of Atmospheric Chemistry, 2000, 37: 245-282.

Xing J, Mathur R, Pleim J, et al. Representing the effects of stratosphere–troposphere exchange on 3-D $O_3$ distributions in chemistry transport models using a potential vorticity-based parameterization[J]. Atmospheric Chemistry and Physics, 2016, 16(17): 10865-10877.

Zaveri R A, Easter R C, Fast J D, et al. Model for simulating aerosol interactions and chemistry (MOSAIC)[J]. Journal of Geophysical Research: Atmospheres, 2008, 113(D13).

---

## Author Response (AR2)

Dear Dr. **Xiaohong Liu**

Thank you very much for your feedback and bringing the typos and grammar errors in our manuscript to my attention. I appreciate your thorough review and commitment to maintaining the high standard of GMD. I have addressed all the mentioned issues and revised the manuscript to assure the quality and readability. All the revisions are listed as follows:

1. Some grammar errors reguarding the manuscript and supplementary materials.

Response: We corrected all the errors accordingly and posted the manuscript and the Supplement with Check Changes turned on.

2. "radiuses" should be "radii" in Table S1.

Response: We have fixed the typo.

3. "Duste" should be "Dust" in Table S2.

Response: We have fixed the typo.

4. "BCs" is boundary conditions or black carbon?

Response: "BCs" is boundary conditions. To avoid confusion, we add the full spelling of BC (black carbon) in Table S2.

In addition to above typos, we have paid close attention to our manuscript and the Supplement and further corrected other typos and grammar errors (in blue and italic) as follows:

Manuscript part:

1. Abstract has be rewritten as "*Two-way coupled meteorology and air quality models, which account for aerosol–radiation–cloud interactions, have been employed to simulate meteorology and air quality more realistically. Although numerous related studies have been conducted, none compared the performances of multiple two-way coupled models in simulating meteorology and air quality over eastern China. Thus, we systematically evaluated annual and seasonal meteorological and air quality variables simulated by three open-sourced, widely utilized two-way coupled models (Weather Research and Forecasting (WRF)–Community Multiscale Air Quality (WRF–CMAQ), WRF coupled with chemistry (WRF–Chem), and WRF coupled with a regional chemistry-transport model named CHIMERE (WRF–CHIMERE)) by validating their results with surface and satellite observations for eastern China in 2017. Although we have made every effort to evaluate these three coupled models under configurations as*

*consistent as possible, there are still unavoidable differences in the treatments of physical and chemical processes in them. Our thorough evaluations revealed that all three two-way coupled models reasonably captured the annual and seasonal spatiotemporal characteristics of meteorology and air quality. Notably, the roles of aerosol–cloud interaction (ACI) in improving the models' performances were limited compared to those of aerosol–radiation interaction (ARI). The sources of uncertainties and bias among the different ACI schemes in the two-way coupled models were identified. With sufficient computational resources, these models can provide more accurate air quality forecasting to support atmospheric environment management and deliver timely warning of heavy air pollution events. Finally, we proposed potential improvements of two-way coupled models for future research.*".

2. Lines 109-114: "One-year long-term simulations in eastern China were examined using the two-way coupled WRF-CMAQ, WRF-Chem, and WRF-CHIMERE models, with and without enabling ARI and/or ACI, and with 27-km horizontal grid spacing (there were 110, 120, and 120 grid cells in the east–west direction, and 150, 160, and 170 in the north–south direction for WRF-CMAQ, WRF-Chem, and WRF-CHIMERE, respectively)." has been rewritten as "*One-year simulations of meteorology and air quality in eastern China were examined using the two-way coupled WRF–CMAQ, WRF–Chem, and WRF–CHIMERE models with and without enabling ARI and/or ACI, as well as with a 27 km horizontal grid resolution (the east–west direction comprised 110, 120, and 120 grid cells, and the north–south direction 150, 160, and 170 grid cells for the WRF–CMAQ, WRF–Chem, and WRF–CHIMERE models, respectively).*".

3. Line 118: "verision 1.5 (FINN v1.5)" has been revised as "*version 1.5 (FINN v1.5)*".

4. Line 137: "0.0001 s$^{-1}$, 0.0001 s$^{-1}$, and 0.00001 s$^{-1}$, respectively" has been revised as "*0.0001, 0.0001, and 0.00001 s$^{-1}$, respectively*".

5. Caption for Table 2: "Table 2. Summary of scenarios setting in three coupled models." has been revised as "*Table 2. Summary of scenario settings in the three coupled models.*".

6. Line 233: "Multi-model" has been revised as "*Multimodel*".

7. Line 332: "LT 08:00 and 20:00" has been revised as "*08:00 and 20:00 LT*".

8. Line 337: "5th, 25th, 75th, and 95th percentiles" has been revised as "*the 5$^{th}$, 25$^{th}$, 75$^{th}$, and 95$^{th}$ percentiles*".

9. Line 372: "Table 3; Figures 5, S9, S12–S14" has been rewritten as "*Table 3 and Figs. 5, S9, and S12–S14*".

10. Line 374: "relative poor" has been revised as "*relatively poor*".

11. Line 462: "conversion of liquid to ice" has been rewritten as "*conversion of liquid water to ice*".

12. Lines 481-485: "When only ACI or ARI and ACI are enabled, the online calculation of prognostic CDNC is performed in WRF–Chem and WRF–CHIMERE using the maximum supersaturation (Abdul-Razzak and Ghan, 2002; Chapman et al., 2009; Tuccella et al., 2019)" has been revised as "*With enabling only ACI or both ARI and ACI effects, prognostic CDNC is online calculated in the two-way coupled WRF–Chem and WRF–CHIMERE models when cloud maximum supersaturation is greater than aerosol critical supersaturation (Abdul-Razzak and Ghan, 2002; Chapman et al., 2009; Tuccella et al., 2019).*".

13. Lines 485-488: "Although we have obtained preliminary quantitative results of the ACI effects on regional precipitation, CF, and LWP, it should be kept in mind that several limitations in representing ACI effects still exist in state-of-the-art two-way coupled models;" has been rewritten as "*Although we have obtained preliminary quantitative results of the ACI effects on regional PREC, CF, and LWP, we acknowledge that several limitations still exist regarding the representation of the ACI effects in state-of-the-art two-way coupled models.*".

14. Line 488-489: "a lack of consideration of the responses" has been revised as "*a lack of consideration for the responses*".

15. Line 501: "Multi-model" has been revised as "*Multimodel*".

16. Lines 517-519: "As shown in Table 4 and Figs. 6–7, WRF-CMAQ underestimated annual and seasonal (except for autumn) PM$_{2.5}$ concentrations with NMBs ranging from −9.78% to −6.39% and −17.68% to +5.17%, respectively." has been rewritten as "*Table 4 and Figs. 6–7 reveal that WRF–CMAQ underestimated the annual and seasonal (except for autumn) PM$_{2.5}$ concentrations, with NMBs of −9.78% to −6.39% and −17.68% to +5.17%, respectively.*".

17. Line 649: "sections 4.1 and 4.2" has been revised as "*Sections 4.1 and 4.2*".

18. Line 661: "aerosol species group" has been revised as "*aerosol species groups*".

19. Line 662-664: "7 (294.6, 303.2, 310.0, 316.4, 333.1, 382.0 and 607.7 nm), 4 (300, 400, 600 and 999 nm), and 5 (200, 300, 400, 600, and 999 nm) effective wavelengths" has been revised as "*seven* (294.6, 303.2, 310.0, 316.4, 333.1, 382.0, and 607.7 nm), *four* (300, 400, 600 and 999 nm), and *five* (200, 300, 400, 600, and 999 nm) effective wavelengths".

20. Lines 702, 704 and 706: "hour per day" has been revised as "*h per day*".

21. In Table 5: "hour" has been revised as "*h*".

Supplement part:

1. Lines 6-8: "Most models tended to underestimate annual and seasonal Q2 ($-0.57$ to $-0.18$ g kg$^{-1}$ and $-1.16$ to $+0.20$ g kg$^{-1}$, respectively), and the underestimations were most significant in summer." has been rewritten as "*Most models exhibited a tendency to underestimate annual and seasonal Q2, with MBs ranging from $-0.57$ to $-0.18$ g kg$^{-1}$ and $-1.16$ to $+0.20$ g kg$^{-1}$ in WRF-Chem and WRF-CHIMERE, respectively.*".

2. Lines 11-14: "Compared with simulations that did not have aerosol feedbacks enabled, WRF-CMAQ_ARI and WRF-CHIMERE_ARI increased the negative biases of annual and seasonal Q2, with the former being more significant (Fig. 3 and Table S3)." has been rewritten as "*In contrast to simulations without enabling aerosol feedbacks, the negative biases in annual and seasonal Q2 simulated by WRF-CMAQ_ARI and WRF-CHIMERE_ARI were amplified, and the WRF-CMAQ_ARI simulations exhibited bigger negative biases (see Fig. 3 and Table S3).*".

3. Lines 19-22: "All simulated results had the highest correlations in winter and the lowest in summer, because the convective activity was enhanced in summer and the models struggle to effectively capture this." has been rewritten as "*All simulated results presented the highest correlations in winter and the lowest in summer and the possible reasons are due to much more convective activities in summertime, which are not accurately captured in all coupled models.*".

4. Lines 55-57: "the entrainment layer was further considered in the ACM2 scheme for PBLH calculations (Xie et al., 2012)." has been revised as "*different to the YSU scheme, ACM2 considers the entrainment layer in the PBLH calculation (Xie et al., 2012).*".

5. Lines 58-62: "Meanwhile, all correlations of the three models at 20:00 LT (R = 0.3–0.4) were better than those at 8:00 LT (R = 0.1–0.2), because the gradient of the rapid increase in PBLH in the morning was larger than that of the gradual decrease in PBLH at night, and hence more difficult to accurately simulate." has been rewritten as "*Meanwhile, all correlations of PBLH simulated by the three coupled models at 20:00 LT (R = 0.3–0.4) were better than those at 08:00 LT (R = 0.1–0.2), which indicated that the PBL schemes in these models were able to calculate PBLH after PBL collapsing slightly better than before PBL developing and more obervations with better spatiotemporal resolutions are needed to further evaluate the models' performance.*"

6. Lines 66-69: "As shown in Fig. 3 and Table S3, the effects of aerosol feedbacks on MB and RMSE were larger than that on R. Considering that the MBs of PBLH are important for the simulation of air quality, the MBs were further analyzed here." has been rewritten as "*As shown in Fig. 3 and Table S3, the effects of aerosol feedbacks on MB and RMSE of PBLH were greater than on R. Considering that the MBs of PBLH are important for accurately simulating regional air quality, the MBs were further analyzed here.*".

7. Lines 82-83: the three coupled models performed well for the shortwave radiation at *the* top of the atmosphere (*SRTOA*) and longwave radiation at *the* top of the atmosphere (*LRTOA*).

8. Lines 108-109: "These gaps had been filled by Zhang et al. (2021) in CMAQ v5.3 but not incorporated into the official released versions." has been rewriiten as "*Recently, Zhang et al. (2021) addressed these gaps in CMAQ version 5.3 but related modules had not been integrated into the latest officially released version (version 5.4).*".

9. Line 129: Gao et al. (2018) also showed that all two-way coupled models, except the WRF-Chem version from the University of Iowa *modelling* group, tended to underestimate $SO_2$ ($-54.77$ to $4.50$ $\mu g\ m^{-3}$) over the North China Plain during January, 2013.

10. Line 155: "*normalized gross error (NGE)*" has been added.

11. Title of Table S3: "*NGE*" has been added. "at LT 08:00 and 20:00" has been changed to "*at 08:00 and 20:00 LT*".

12. Title of Table S4: "evaluation" has revised as "*statistical*".

13. In Table S4: "*N/A*" has been added.

14. Footnote of Table S2: "unspeciated particulate matter in" has been revised as "*particulate matter that can not be speciated into*".

15. In Table S6: "in term of" has been revised as "*in terms of*".

16. Title of Table S7: Microphysics variables used in the two-way coupled WRF-CMAQ, WRF-Chem and WRF-CHIMERE models with enabling ACI effects compared to *those* without aerosol feedbacks.

Thank you again for your valuable suggestions and we really appreciate your help! Please let me know if we need to make any further revisions to improve the quality of our paper.

Sincerely yours,

Chao Gao, PhD

Assistant Professor, Key Laboratory of Wetland Ecology and Environment

Northeast Institute of Geography and Agroecology, Chinese Academy of Sciences

We would like to thank the reviewer for the constructive comment really helping to improve this manuscript. Below, we address the comment in full detail. Following the Reviewers' comment in black, please find our point-to-point responses in blue. Hereafter, all new added or modified sentences are marked in blue and italic in this response.

**Anonymous Referee #2**

In the abstract, this sentence is not clear to this reviewer, please clarify:

"The impacts of aerosol-cloud interaction (ACI) on model performances' improvements were limited compared to aerosol-radiation interaction (ARI), and several possible improvements on ACI representations in two-way coupled models are further discussed and proposed."

Response: We rewrote this sentence as follows.

*Notably, the roles of aerosol–cloud interaction (ACI) in improving the models' performances were limited compared with those of aerosol–radiation interaction (ARI).* The sources of uncertainties for ACI schemes in the two-way coupled models were further pointed out.

*The sources of uncertainties and bias in meteorology and air quality simulated by the two-way coupled models with considering ARI and ACI effects were further pointed out.*

Furthermore, as commented by the editor, all the typos had been fixed. The language quality of both the manuscript and supplementary materials has been improved.

---

## Author Response (AR3)

Dear Polina Shvedko,

We would like to thank you for the constructive comment really helping to improve this manuscript. Below, we address the comment in full detail. Following your comment in black, please find our point-to-point responses in blue. Hereafter, all new added or modified sentences are marked in blue and italic in this response.

Regarding figures 3, 6: please ensure that the colour schemes used in your maps and charts allow readers with colour vision deficiencies to correctly interpret your findings. Please check your figures using the Coblis – Color Blindness Simulator (https://www.color-blindness.com/coblis-color-blindness-simulator/) and revise the colour schemes accordingly.

Response: We revised colour schemes in figures 3 and 6 as follows.

[Figure]

Figure 3. Time series of the observed and simulated hourly SSR, T2, RH2, and WS10 by coupled WRF–CMAQ, WRF–Chem, and WRF–CHIMERE with/without aerosol feedbacks over ECR in 2017.

[Figure]

Figure 6. Time series of the observed and simulated hourly PM$_{2.5}$ and O$_3$ concentrations by WRF–CMAQ, WRF–Chem, and WRF–CHIMERE with/without aerosol feedbacks over ECR in 2017.